# Technical Note: Considerations on using uncertain proxies in the analogue method for spatiotemporal reconstructions of millennial-scale climate

Oliver Bothe[1] and Eduardo Zorita[1]

[1]Helmholtz Zentrum Geesthacht, Institute of Coastal Research, 21502 Geesthacht, Germany

**Correspondence:** Oliver Bothe (ol.bothe@gmail.com)

**Abstract.** Inferences about climate states and climate variability of the Holocene and the deglaciation rely on sparse paleo-observational proxy data. Combining these proxies with output from climate simulations is a means for increasing the understanding of the climate throughout the last tens of thousands of years. The analogue method is one approach to do this. The method takes a number of sparse proxy records and then searches within a pool of more complete information (e.g., model simulations) for analogues according to a similarity criterion. The analogue method is non-linear and allows considering the spatial covariance among proxy records.

Beyond the last two millennia, we have to rely on proxies that are not only sparse in space but also irregular in time and with considerably uncertain dating. This poses additional challenges for the analogue method, which have seldom been addressed previously. The method has to address the uncertainty of the proxy-inferred variables as well as the uncertain dating. It has to cope with the irregular and non-synchronous sampling of different proxies.

Here, we describe an implementation of the analogue method including a specific way of addressing these obstacles. We include the uncertainty in our proxy estimates by using *ellipses of tolerance* for tuples of individual proxy values and dates. These ellipses are central to our approach. They describe a region in the plane spanned by proxy dimension and time dimension for which a model analogue is considered to be acceptable. They allow us to consider the dating as well as the data uncertainty. They therefore form the basic criterion for selecting valid analogues.

We discuss the benefits and limitations of this approach. The results highlight the potential of the analogue method to reconstruct the climate from the deglaciation up to the late Holocene. However, in the present case, the reconstructions show little variability of their central estimates but large uncertainty ranges. The reconstruction by analogue provides not only a regional average record but also allows assessing the spatial climate field compliant with the used proxy predictors. These fields reveal that uncertainties are also locally large. Our results emphasize the ambiguity of reconstructions from spatially sparse and temporally uncertain, irregularly sampled proxies.

## 1 Introduction

It is a pervasive idea in environmental and climate sciences that past states provide us with information about the future (Schmidt et al., 2014a; Kageyama et al., 2018). Therefore, paleoclimatology aims to understand past spatial and temporal

climate variability, preferentially using a dynamical understanding of the climate processes. To achieve this, we need spatial and temporal information about past climate states and past climate evolutions. Our understanding of the past, however, relies on spatially and temporally sparse paleo-information. Data assimilation methods and data-science approaches are ways to provide estimates for the gaps in time and space. One simple approach is the analogue method or so called proxy surrogate reconstructions (Gómez-Navarro et al., 2017; Jensen et al., 2018). This method is similar to k-nearest neighbour classification algorithms in machine learning applications. The present manuscript discusses an implementation of the analogue method for reconstructing surface temperature over timescales including the Holocene and the last deglaciation.

If we want to use the analogue method beyond approximately the last two millennia, we have to tackle additional challenges, which usually can be evaded for the Common Era. For example, our proxy records are not only spatially sparse but they also have a coarse temporal resolution on these timescales. Furthermore, the sampling generally is irregular for each individual proxy. Indeed, sample dates differ between proxies on these timescales, and these dates are also uncertain. Recently, Jensen et al. (2018) use the approach to reconstruct the climate at the Marine Isotope Stage 3 (MIS3, 24,000 to 59,000 years before present; 24–59kyr BP) addressing such challenges. Including part of a deglacial period, as we do here, further complicates applications as we consider a climate trajectory with strong trends.

The basic idea of the analogue method is simple. An analogy tries to explain an item based on the item's resemblance or equivalence to something else. In the analogue method, one uses a set of sparse proxies, i.e. predictors, and searches for analogues for them in a pool of candidates that are spatially more complete. In paleoclimatology, the predictors can be local proxy records and the candidate analogues can be fields from climate model simulations. One assesses the similarity of the simulation output and the proxy records at the proxy locations to find valid analogues. The reconstructed field is then the complete field given by the analogue.

It is important to note that comparable approaches suffer from a trade-off between accuracy and reliability of reconstructions as shown by Annan and Hargreaves (2012) for a particle filter method. This depends on quality and quantity of the available proxy records. This drawback also affects the analogue method as shown by Franke et al. (2010) and Gómez-Navarro et al. (2015), who find that the skill accumulates at the predictor locations. Similarly, Talento et al. (2019) highlight that the analogue method may perform badly in regions with little proxy coverage.

Most paleoclimate applications of the analogue method focussed on the Common Era of the last 2,000 years (e.g., Franke et al., 2010; Trouet et al., 2009; Gómez-Navarro et al., 2015, 2017; Talento et al., 2019; Neukom et al., 2019). In this context, Graham et al. (2007) call the results of the analogue method a "proxy surrogate reconstruction". Gómez-Navarro et al. (2017) provide a comparison of the analogue approach to more complex common data assimilation-techniques. Applications often only consider the single best analogue, which may not necessarily be appropriate especially for predictors affected by uncertainty. Paleo-applications of the analogue method generally try to upscale the local proxy information but the analogue method was also applied for downscaling of large-scale information (e.g., Zorita and von Storch, 1999).

Here, we describe another approach to obtain reconstructions by analogue over millennial timescales based on spatially and temporally sparse and uncertain proxies. It differs in some aspects from the approach so far applied to shorter and more recent periods. Our approach tries to explicitly consider not only age uncertainties (compare with Jensen et al., 2018) but also

the uncertainties of the proxy values or, similarly, of the temperature reconstructions inferred from these proxies. We make specific assumptions on the uncertainty of the data and the dates of the proxy predictors. We further account for the temporal irregularity of the sampling of different predictors. As explained in more detail later, our approach considers an analogue candidate simulation field as valid analogue if it complies with our assumptions on the uncertainty of the proxy predictors. We apply the method over time periods encompassing parts of the last deglaciation until the late 20th century of the Common Era (CE). That is, we try to apply the analogue method over a period when the climate cannot validly be described as stationary at local, regional, and global spatial scales.

Beyond the mentioned challenges for analogue reconstructions on millennial timescales, the method is also constrained by the pool of available analogue fields. Van den Dool (1994) considers how likely it is to observe two atmospheric flows over the northern Hemisphere that resemble each other within the observational uncertainty. The study finds that a pool would have to include a nonillion, i.e., $10^{30}$ of potential analogues to achieve this. Obviously, we aim for less accuracy in paleoclimatology due to larger uncertainties. However, there are still only few climate simulations for relevant timescales, and these simulations also cover only parts of the time periods of interest. Furthermore, these simulations stem from different climate models whose reliability on these timescales may not have been shown yet (Weitzel et al., 2018; Kageyama et al., 2018).

The next section first summarizes again the main characteristics of analogue searches for paleo-reconstructions. Afterwards, we present our way of dealing with uncertain tuples of data and date, that is with describing ranges of tolerance for which we choose analogues. Simulation fields are considered analogues if they fall within these tolerance ranges at all considered proxy locations. We also describe how we consider the fact that different proxies are sampled at different times. The section also presents our selection of a simulation pool. We present results for a multimillennial period for a pseudoproxy setup (compare Smerdon, 2012) and a realistic setup for the European-North-Atlantic sector. We also shortly describe results for alternative proxy setups. Finally, we discuss our assumptions and results. We aim to emphasize the opportunities of the analogue method while also highlighting its challenges.

## 2   Analogue method, assumptions, and data

### 2.1   General Method

In an analogue search one tries to complement incomplete information from one dataset by data from other more complete datasets. One ranks the more complete data by its similarity to the available information in the first data set. In paleoclimatology this usually means that one uses a set of spatially sparse proxy records and wants to find fields from simulations or reanalyses that are most analogous to the proxy records at their locations. The pool of candidate fields depends on the available simulations and reanalyses.

If, for example, one uses proxies for temperature, such a ranking may simply provide the simulated temperature field that has the smallest Euclidean distance to the sparse proxy information at their locations. Alternatively, one can consider not just one but a small number of good analogues with small distances (Franke et al., 2010; Gómez-Navarro et al., 2015, 2017; Talento et al., 2019). However, it is also possible to define a range of tolerable deviations from the proxy predictor values and consider

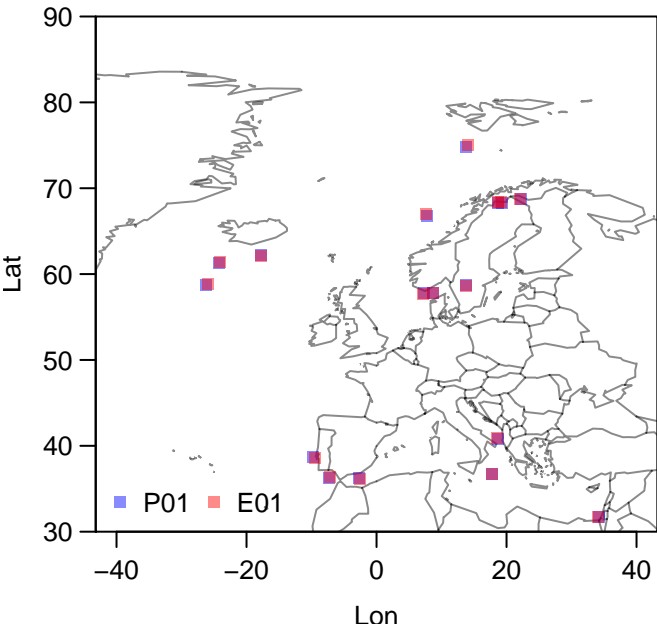

**Figure 1.** Map of the reconstruction domain and the proxy predictors: for the pseudoproxy setup (blue), experiment P01, and for the main proxy setup (red), experiment E01. Please note, the small offset between the proxy locations and their pseudoproxy counterparts on the discrete model grid.

all analogue-candidates that are within this range as valid analogues (compare Bothe and Zorita, 2020). Matulla et al. (2008) discuss the effect of the choice of similarity measures for a different application.

An important aspect of a paleoclimate reconstructions is the uncertainty of the reconstructed data. To our knowledge, only Jensen et al. (2018) and Neukom et al. (2019) consider the uncertainty of the final reconstruction among earlier paleo-applications of the analogue method, and only Jensen et al. use proxies with prominent age uncertainties in their work on MIS3. They perform multiple reconstructions to obtain reconstruction uncertainties by shifting the dates of their proxies within the stated age uncertainties. Uncertainty information is particularly relevant for applications like the one of Jensen et al. where one
has to deal with predictors that are sparse, irregular, and uncertainly dated.

## 2.2   Present application of the analogue method

We use spatially and temporally sparse proxies, affected by uncertainties in their values and their dating for analogue searches on millennial timescales. Next, we detail our simplifying assumptions about what the data represents, its uncertainties, and the dating uncertainties. We also describe how we choose the dates for which we perform the climate reconstruction.

### 2.2.1 Variable of interest

Our interest is in temperature. Specifically, we concentrate on means of seasonal or annual temperature at the surface. We consider proxies for which the literature previously reports a sensitivity to temperature in form of a calibration relation. We search for analogues within fields of simulated surface temperature. To do the comparison, we consider the model variable "surface temperature" over the European-North-Atlantic domain shown in Figure 1. The reconstruction also uses these fields.

Theoretically, the variable or variables to be reconstructed can be different from the variable or multiple variables represented by the paleo-observational predictors. Indeed, we here assume that it is possible to reconstruct annual temperatures from proxy records with diverse seasonal attributions.

Using temperature in a multi-proxy comparison requires a number of assumptions. First, we assume that the proxy recorders indeed were temperature-sensitive. More importantly here, we assume that all the different recorders, aquatic or otherwise, represent temperature at the surface. This is an assumption of convenience in view of potential habitat biases of the proxy records (Telford et al., 2013; Tierney and Tingley, 2015; Jonkers and Kučera, 2017; Jonkers and Kučera, 2019; Rebotim et al., 2017; Tierney and Tingley, 2018; Dolman and Laepple, 2018; Reschke et al., 2019; Kretschmer et al., 2016, 2018; Malevich et al., 2019; Tierney et al., 2019).

### 2.2.2 Data handling and use of model simulation pool

Section 2.3.1 gives details on our selected proxies. In short, we choose 17 proxies at locations in the European-North Atlantic sector (Figure 1) from the compilation of Marcott et al. (2013). These are from a variety of different proxy systems. We take these as published by either Marcott et al. (2013) or the original publications. Therefore, calibrations and uncertainty estimates have diverse origins. Considering the proxy ages and their uncertainties, we adopt those as published.

Optimally one would aim for maximal consistency in the comparison. Consistency among parameters and calibration ensures a relation among the proxy predictors, which, one can assume, increases the chance that the proxy records lead to a selection of physically meaningful analogues. In this case, the proxies can effectively anchor the analogue selection. We here assume that all chosen proxy-types reliably represent the target of interest and a multi-proxy approach is viable.

The analogue method allows searching for analogues at dates when there is information. One can pool the predictor dates into consistent intervals of, for example, 500 years, and search for analogues for these 500-year pools. One can follow the example of Jensen et al. (2018) and interpolate the proxy records to consistent time steps using the age models for the individual records. We choose here a different approach; we identify all the years for which at least one of the chosen proxy records includes a dated value. We perform analogue searches for these dates according to our considerations on uncertainty, which we describe in the following subsection 2.2.3.

Each data point of a proxy series potentially represents a time-interval of a specific length and the comparison should consider this temporal resolution. That is, if one data point represents a 50-year accumulation and another data point represents a 500-year accumulation, the procedure ideally accounts for these differences. We decide to use typical resolutions instead of individual resolution estimates to simplify the procedure and allow a computationally more efficient analogue search for data

and time uncertain proxies. Indeed, it is not necessarily the case that a proxy-record publication includes the information to estimate the pointwise temporal resolution. Considering information provided by Marcott et al. (2013) on their proxies, we conclude for our chosen subset of these (compare Table 1) that the proxies have at best centennial average resolutions. While there are proxies with higher and lower resolutions, a reasonable estimate for the overall average resolution is centennial.

Therefore, we decide to compare the proxy estimates to 101-year averages of the model simulation output. That is, we compare them to 101-year mean values, which we obtain by using a 101-year moving mean on the simulation output time series that is closest to the proxy location.

In one test case, we do not preprocess the simulation output but use the annually resolved values of the output for the comparison. For this specific test, we also include the simulation data from the FAMOUS-HadCM3 simulations for the QUEST-project (compare Smith and Gregory, 2012, and section 2.3.2) for which output data is only representative for decadal forcing conditions. We refer to these as QUEST FAMOUS simulations. We do two more tests with differing resolutions that use 51- and 501-year averages respectively.

We test the whole approach by using pseudoproxies. We construct the pseudoproxies following the ensemble approach of Bothe et al. (2019a). Their approach takes simulated grid-point data and transforms it in multiple steps into a pseudoproxy record. The steps follow the framework of a proxy system model including a sensor model, an archive model, and an observation model (see Evans et al., 2013). Bothe et al. (2019a) first add a noise estimate for environmental non-temperature influences at the sensor stage. This stage also includes adding a bias term due to changing insolation. Next, the archive stage primarily represents a smoothing of this record, which is meant to reflect effects of, e.g., bioturbation. The measurement stage adds another noise term. After sampling this record at a specific number of dates, the procedure, finally, also adds an error term for the proxy data reflecting effective dating uncertainties. We set this term to zero in the script of Bothe et al. (2019a) because we do not aim to transfer dating uncertainty to the data uncertainty. Pseudoproxy locations are simulation data grid-points close to the real proxy locations. Figure 1 shows the 17 pseudoproxy and proxy locations and allows to identify their slight offsets due to the discrete character of the simulation data. The pseudoproxy generation smooths each record to mimic the temporal filtering effects of the real environmental archive. The smoothing length is randomly chosen but temporally uniform for each record. The search for analogues again uses 101-year mean estimates from the simulation pool (compare the paragraph above).

Simulations potentially differ in their modern day climate mean (compare, e.g., Zanchettin et al., 2014). Using anomalies can circumvent this issue. However, there is not a clear path in our application towards computing them equivalently in simulations and proxy data. If such a path exists, one can consider simulation output as anomalies to the climatology over the 20th century or over the full simulation period or over the longest period common to all simulations. For example, Jensen et al. (2018) construct the anomaly record for a data series by subtracting the temporal mean calculated over the full period of the record of interest. Their period of interest backs this decision. The proxy records of Jensen et al. (2018) suggest an overall rather stable climate in the North Atlantic during Marine Isotope Stage 3 although a number of Dansgaard-Oeschger (DO) events occurred during this period. We presume that using anomalies allows to include a wider range of simulations and analogue candidates for each date.

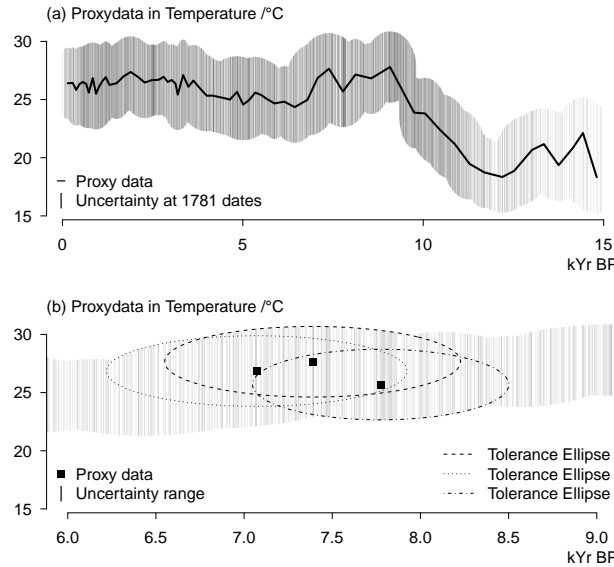

**Figure 2.** Considerations on uncertainty and constructing tolerance envelopes: **(a)** Example proxy data (line) and assumed data uncertainty at all dates when we reconstruct values. Number of dates are all dates when any of the proxies included has a dated data point. **(b)** Proxy data at three example dates and tolerance ellipses for these dates using data uncertainty and age/date uncertainty.

However, in the present case, the period of interest includes mainly the last 15kyr. Thus, it spans part of the deglaciation from the Last Glacial Maximum to the Holocene optimum. Our selection of simulations can only piecewise cover that period of interest, which complicates the construction of a surface temperature candidate pool. Indeed, the most recent dates differ among the proxy records, and, thus, there is no simple procedure to provide anomalies relative to a consistent modern climate.

Additionally, using anomalies may introduce climatic inconsistencies if we are interested in climate variables other than temperature. For these reasons, we decide that we cannot reasonably use anomalies. Instead, we try to find analogues for the local proxy reconstructions in their absolute temperature units without subtracting any climatology.

### 2.2.3 Proxy uncertainty

We are interested in millennial timescales from the last deglaciation until the recent past. On these timescales, uncertainty
affects our proxy predictors in two ways. First, we have to consider the age or dating uncertainty. Second, the measured proxy data and the temperatures inferred from them are affected by various sources of uncertainty (compare, e.g., Dolman and Laepple, 2018; Reschke et al., 2019; Jensen et al., 2018, and their references).

Previous applications of the analogue method usually did not consider proxies with considerable age uncertainties except for Jensen et al. (2018). Jensen et al. consider the age uncertainty by shifting the date of each proxy by $\pm 500$ years. Thereby,
they obtain an ensemble of $2^{14}$ reconstructions from which they calculate confidence intervals for their final reconstruction. They do not separately consider the uncertainty of the proxy/reconstruction value. For details, see Jensen et al. (2018).

Uncertainty of proxies in time and date is commonly expressed as central value and a given uncertainty range. These ranges may be given as plus and minus standard deviations around the central value, e.g., $\sim \pm 1.64$ or $\sim \pm 2.58$ standard deviation ranges, or as percentage confidence intervals, e.g., 90% or 99% intervals. Such usage of a percentage view can be extended to pairwise expressions of the uncertainty. This is central to our approach.

That is, we choose a different approach (Figure 2) compared to, e.g., Jensen et al. (2018). We interpret each data point in a proxy series together with its dating as a data point in the two dimensional space spanned by temperature and time. Each proxy data point is located on this two dimensional temperature-time plane and each point is surrounded by uncertainty ranges along both dimensions. We can utilize the uncertainty ranges in the two dimensions as our *area of tolerance*, in which the analogue candidate simulation fields should be located to be considered as good analogues. That is the area of tolerance defines our criterion for selection of good and valid analogues. If a candidate field is within the tolerance areas at all considered proxy locations, it is thought to be a valid analogue. We can define tolerance ranges for different levels of pairwise proxy-time uncertainty. In equivalence to common expressions of uncertainty or confidence intervals these can be formulated as pairwise 90% or 99% intervals. These choices of intervals yield increasingly larger areas of tolerance. For the real proxies, the data uncertainties that enter the computation of the tolerance area follow our simplifying assumptions detailed in 2.3.1, while the dating uncertainties are taken from the compilation published by Marcott et al. (2013). For the pseudoproxy setup, the pseudoproxy algorithm of Bothe et al. (2019a) provides estimates for data and dating uncertainty.

To define these areas of tolerance we still have to define their shape. Our interest is in finding analogues that agree with the proxy data but also account for these uncertainties. Then, we could take the uncertainty estimates of temperature and time to construct a two-dimensional uniform estimate in form of a rectangle of tolerance. Analogue candidates would be valid analogues if they fall locally within these rectangles. If they fall outside of the rectangle they would not be considered valid analogues. Although the uncertainties in temperature and time are commonly taken to be Gaussian, the rectangular approach is the best one if we consider the uncertainties of date and temperature isolated from each other. Then, our tolerance for the temperature data has the same structure at the border of our temporal tolerance range as it has at the central estimate for the date. However, in our application, we do not see both tolerance ranges in isolation. We assume that our tolerance range is a two-dimensional pairwise construct in time and temperature. Then, our tolerance construct takes the shape of a two-dimensional Gaussian. This implies that our tolerance areas are ellipses. Such ellipses can be computed dependent on an assumed pairwise confidence level or coverage or in our interpretation tolerance range. We refer to these as percentage levels.

According to our view of tolerance ranges as tolerance ellipses, we accept fewer analogues for dates far away from the median proxy age estimate. For these dates, analogue candidates need to be numerically very close to the proxy. In contrast, we accept more analogues close to the central age estimate of the proxy, and tolerate that they may more strongly differ from the numerical central estimate of the proxy. We acknowledge that it may seem counterintuitive that we reduce the range in data uncertainty at dates far away from our central best estimate of temperature and date. This originates from our assumption that the pair of data and date stems from a two-dimensional distribution that is centred on our best estimate. Thereby, the likelihood of a valid pair of data and date reduces further away from our best estimate according to the assumptions on the distribution.

As we have estimates of the uncertainties of the data point, we can construct and visualise the ellipses of tolerance around each data point under the assumption of two-dimensional Gaussian tolerance areas. We use the R (R Core Team, 2019) package ellipse (Murdoch and Chow, 2018) whose default ellipse-function follows Murdoch and Chow (1996) by implementing the ellipse equation as

$$55 \quad (x,y) = (\alpha \cdot \sigma_x \cdot cos(\theta + d/2) + \mu_x, \alpha \cdot \sigma_y \cdot cos(\theta - d/2) + \mu_y), \quad (1)$$

where $x$ is our time-dimension and $y$ is our temperature dimension. Furthermore, $\alpha$ is the tolerance level of interest (i.e., the percentage levels mentioned above) transformed to a t-test statistic as implemented by Murdoch and Chow (2018), $\sigma_i$ are the one standard deviation levels of the uncertainties in x- and y-direction, $\mu_i$ are the best estimates of the values in x- and y-direction, i.e. date and data, and $\theta \in [0, 2\pi]$. $cos(d) = \rho$ is the correlation between temperature and time uncertainties.

However, we do not consider potential non-zero covariances between dating uncertainty and proxy uncertainty. For simplicity, we also do not take account of the likely correlations between subsequent tuples of data and date.

A two-dimensional tolerance ellipse represents tolerance levels for two-dimensional normal distributed data. However, as in the simple case of a tolerance rectangle, our interest is only in the ellipse as a binary decision criterion to consider the data included in the ellipse and to neglect the data outside of the ellipse. That is, we use the ellipse as an area of tolerance to identify

valid analogues from our analogue candidate simulation field pool. The ellipses provide the maximal acceptable distance for a simulated data to be considered as an analogue (Figure 2b). That is, the ellipses are not meant to represent the uncertainty ranges in the value of the proxies. They are rather meant to define a limit beyond which an analogue candidate is not considered any more. Essentially the ellipses define a weighting scheme (although with binary weights) based on the proxy and dating uncertainties.

The ellipses are defined from points in the proxy-time space (see Figure 2b). We construct ellipses for those data points for which a published record provides ages. Our tolerance range for a specific date as well as the tolerance envelope for the full proxy record follows from the superposition of the tolerance ellipses from successive data points (see panels of Figure 2 and later Figures 5 and 8). This envelope generally provides for each date upper and lower limits of values that the analogue candidates need to fall between. However, the envelope may also result in the impossibility to define analogues for specific

locations or even for all locations for specific dates.

That is, the superposition of ellipses constructs a tolerance envelope (Figure 2a), which we use to identify valid analogues from our candidate pool. The ellipses around the data points mark the limit of their pointwise two-dimensional area of influence in our search for spatially resolved analogues. Their superposition is essential for identifying those simulated data to be considered as analogues. If the tolerance ranges for multiple data points in a record overlap for a given year, we simply take their maximal ranges. Simulated data that fall outside the tolerance ranges are rejected. Thus, for a selected date candidate

5 analaogues have to fall within the tolerance range at all considered locations to be valid analogues.

Because we provide reconstructions only for those years for which one of the chosen proxy records includes a dated value, and because our tolerance estimates are essentially pointwise, the envelope may not be one continuous envelope over the full

period of interest. Furthermore, because we use the envelopes as decision criterion, it can happen that the method fails to find any valid analogues for given years.

10    Our pointwise estimates are compliant with the initial uncertainty of the proxies and our final reconstruction uncertainties are an expression of this initial confidence in the local data. This is in contrast to Jensen et al. (2018), who provide an ensemble of reconstructions. Their uncertainty estimate measures the uncertainty of the initial reconstruction relative to shifted ages. That is, the two different applications of the analogue method consider different things in their uncertainty estimates. The reconstruction uncertainty in our approach originates from the selected analogues.

### 2.2.4    Analogue search

The ellipses of tolerance allow in theory to produce reconstructions for each year included in the dating uncertainty. That is, if a proxy series has a value dated to the year 500 BP with a dating uncertainty of $\sigma = 50$yr, and if we decide to consider dates within $\pm 2\sigma$ then we can search for analogues from 600 to 400 BP. However, we decide to only reconstruct values at those dates at which at least one proxy is dated. That is, if only this hypothetical proxy has a dated value between 600 and 400 BP and it only has this one dated value, we perform the reconstruction only for the year 500 BP. Our assumption is that this maximises the link between the reconstruction and the underlying proxies. Thus, if we increase the width of the tolerance envelope, we usually do not obtain reconstructed values at more dates but only increase the probability to find a valid analogue at a given date. As we show later, there are exceptions, when the wider tolerance envelopes lead to the inclusion of more proxies at specific dates so that the search becomes more constrained and finding an analogue becomes less likely.

In other applications of the analogue method, the choice of a valid analogue usually relies on a distance metric. This is commonly the Euclidean distance (compare Franke et al., 2010; Gómez-Navarro et al., 2017; Talento et al., 2019), although Jensen et al. (2018) use an unweighted root mean square error (RMSE) as distance metric between their proxies and the analogue candidates from their simulation pool. Based on such a distance, one can select the best fit, a small number of good fits, e.g., the ten analogues with the smallest distance, or a composite or interpolation of a small number of good fits.

Here, we deviate from this and decide neither on a fixed number of analogues nor on a defined metric. Candidates in our pool are valid analogues if they are within the tolerance range (compare section 2.2.3) at all considered locations for a selected date. That is, as described above, we have an envelope of tolerance values for specific years and each proxy record. For our standard approach, a candidate is a valid analogue for a date if it falls within the ellipse of tolerance for all proxies. We also mention tests where an analogue is valid if it is outside the ellipses at one location, at two locations or at 25% of the locations. We consider only a small set of potential ellipses. These use 90% and 99.9% percentage levels for the pseudoproxy approach, and either 99% or 99.99% percentage levels for the various proxy setups.

We additionally show one instance of a reconstruction using just one best analogue. For this test, we choose the analogue with the smallest Euclidean distance to our proxy values. As we deal with proxy records that are irregularly spaced in time, we have to find a way to select dates for which to do a single best analogue reconstruction and get the proxy values for these dates. To do so, we consider the proxy values valid at all dates within a given range around their dating. We identify the range of these values, and take the mid-point of that range as the proxy value for this date. We consider values within a 90% or $\sim 1.64$

standard deviation dating uncertainty around the dating. We compute these based on one standard deviations inferred from the originally published dating uncertainty.

In short, our reconstruction is based on the following workflow. We have a set of sparse proxy predictors and a pool of simulated fields. As our proxies are not only sparse in space and uncertain in their values but also irregular and uncertain in time, we have to decide, (a) when to compare them, (b) in which resolution to compare them, and (c) how to consider the uncertainties in time and value. Therefore, we decide to (i) compare the proxies and simulated data for all dates when one proxy is dated, to (ii) compare the proxies to 101-moving means of the simulated data, and (iii) to take the proxy data values as valid within an ellipse of tolerance around the dated value in time and temperature space. Then analogue candidate simulation fields are valid analogues if they are within these tolerance ranges around all proxy records included in the search.

## 2.3 Data

### 2.3.1 Proxies

We concentrate on a European-North-Atlantic domain (Figure 1). There, we choose 17 locations with proxy-inferred temperature records from the collection of Marcott et al. (2013, see also Tables 1 and A1). Nine of these series use alkenone $U_{37}^{K'}$ but the set also includes temperatures derived from foraminifera Mg/Ca (2 records), pollen (2), chironomids (2), TEX86 (1), and foraminiferal assemblages (1) (compare Table 1 and Figure 3). For the various proxy types see, e.g., Rosell-Melé et al. (2001, and their references) or Tierney and Tingley (2018, and their references) for $U_{37}^{K'}$, Anand et al. (2003, and their references) or Tierney et al. (2019, and their references) for foraminiferal Mg/Ca, Kim et al. (2008, and their references) or Tierney and Tingley (2015, and their references) for TEX86, Seppä and Birks (2001) and Seppä et al. (2005) for the specific pollen records, Larocque and Hall (2004) for the specific chironomid records, and Sarnthein et al. (2003a) for the specific record using foraminiferal assemblages.

We do not include all records from Marcott et al. (2013) within the chosen domain. We do not consider additional seasonal attributions for the foraminifera assemblage data of Sarnthein et al. (2003a, compare also Marcott et al., 2013; Sarnthein et al., 2003b). We further excluded the alkenone unsaturation ratios of Bendle and Rosell-Melé (2007, see also Marcott et al., 2013) as well after initial tests due to concerns about the potential influence of sea-ice in simulations. Indeed, we find (not shown) that including this record puts very strong constraints on the analogue candidates and can reduce the chance of finding valid analogues. We exclude two more records because they are co-located with other proxies. That is, we do not use the stacked radiolaria assemblage records of Dolven et al. (2002, see also Marcott et al., 2013) because the upper part of the record is from the same upper core as the $U_{37}^{K'}$ data of Calvo et al. (2002, see also Marcott et al., 2013). Similarly, we ad hoc decide to use the $U_{37}^{K'}$ data of Cacho et al. (2001, see also Marcott et al., 2013) instead of that of Kim et al. (2004a, see also Marcott et al., 2013), which are basically co-located. We use the data of Kim et al. (2004a) in one alternative proxy setup. Table 1 and Figure 3 provide details on our different proxy setups. All in all, we consider nine different setups of proxy networks, which we name E01 to E09. In a pseudoproxy-setup, we use a network of locations equivalent to E01 and therefore name this pseudoproxy-setup P01.

**Table 1.** Information about the considered proxy records: IDs, geographical location, seasonal attribution according to Marcott et al. (2013), proxy type, seasonal attribution used here, and analogue search setups that use the record. All proxy data are from the supplement of Marcott et al. (2013). Table A1 provides references to original publications and data sets. Proxy setups refer to those analogue search tests where this proxy is included (compare also Figure 3).

| Proxy ID | Lat | Lon | Season in Marcott et al. (2013) | Proxy type | Season used | Proxy setups |
|---|---|---|---|---|---|---|
| MD95-2043 | 36.1 | -2.6 | Annual | $U_{37}^{K'}$ | Annual | 1-3, 4-6 |
| M39-008 | 36.4 | -7.1 | Annual | $U_{37}^{K'}$ | Annual | 1-2, 4-7, 9 |
| MD95-2011 | 67 | 7.6 | Summer | $U_{37}^{K'}$ | Summer | 1-4 |
| ODP984 | 61.4 | -24.1 | Winter | Mg/Ca (*N. pachyderma d.*) | Winter | 1, 7-9 |
| GeoB 7702-3 | 31.7 | 34.1 | Summer | TEX86 | Summer | 1, 5-9 |
| IOW225517 | 57.7 | 7.1 | Spring to Winter | $U_{37}^{K'}$ | Annual | 1-4, 6 |
| IOW225514 | 57.8 | 8.7 | Spring to Winter | $U_{37}^{K'}$ | Annual | 1-4 |
| M25/4-KL11 | 36.7 | 17.7 | Spring to Winter | $U_{37}^{K'}$ | Annual | 1-7 |
| AD91-17 | 40.9 | 18.6 | Annual (seasonal bias likely) | $U_{37}^{K'}$ | Annual | 1-6 |
| Lake 850 | 68.4 | 19.2 | Summer | Chironomid transfer function | Summer | 1, 7-8 |
| Lake Nujulla | 68.4 | 18.7 | Summer | Chironomid transfer function | Summer | 1, 7-8 |
| MD95-2015 | 42 58.8 | -26 | Annual | $U_{37}^{K'}$ | Annual | 1-4 |
| D13882 | 38.6 | -9.5 | Summer | $U_{37}^{K'}$ | Summer | 1-6,8 |
| GIK23258-2 | 75 | 14 | Summer | Foram transfer function | Summer | 1, 4-9 |
| Flarken Lake | 58.6 | 13.7 | Annual | Pollen MAT | Annual | 1, 7-9 |
| Tsuolbmajavri Lake | 68.7 | 22.1 | Summer | Pollen MAT | Summer | 1, 5-9 |
| RAPID-12-1K | 62.1 | -17.8 | Late Spring to early Summer | Mg/Ca (*G. bulloides*) | Summer | 1, 6-9 |
| GeoB 5901-2 | 36.4 | -7.1 | Annual | $U_{37}^{K'}$ | Annual | 3 |

We consider the seasonal attributions of individual proxy records in our search for analogues. We generally take the attributions and the calibrations for the records as published by Marcott et al. (2013) but also check the references provided by them. Seasonal attributions are diverse for the various proxy records. The majority is either for summer season (7) or annual (8) according to Marcott et al. (2013). We compare the proxies to the simulation output season that is close to the seasonal attribution as given by Marcott et al. (2013) or the original publication. For simplicity's sake, we only consider the modern meteorological seasons DJF (December to February), MAM (March to May), JJA (June to August), and SON (September to November) as well as the calendar annual simulation means (compare Table 1). We do ignore possible calendar effects (e.g., Bartlein and Shafer, 2019; Kageyama et al., 2018).

Regarding proxy uncertainty, we decided to assume an uncertainty of $\sigma = 1K$ for all proxies as we were not able to infer full uncertainties for every temperature reconstruction either from Marcott et al. (2013) or from the original publications. This

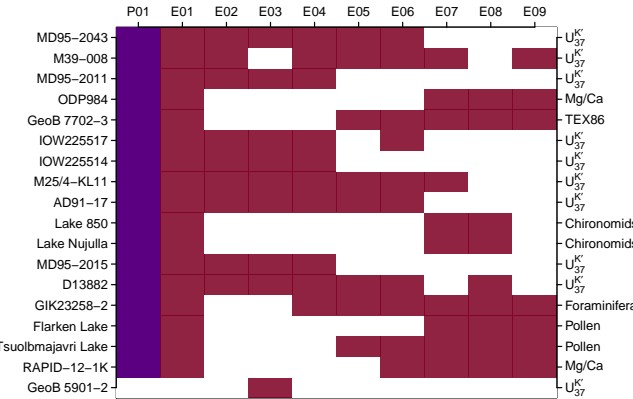

**Figure 3.** Information about the different proxy setups: Matrix of proxy records against proxy setup (P01, indigo, and E01 to E09, burgundy red). For more information see Table 1. White-out means that the relevant proxy is not included in the respective proxy setup. Note that the pseudoproxy setup P01 does not distinguish between proxy types and uses only approximate locations due to the discrete simulation output.

reduces the uncertainty for some records and potentially increases the uncertainty for others. We regard this to be a reasonable simplification.

We performed reconstruction exercises for various proxy setups. We concentrate on the full set of proxies mentioned above (see Figure 1 and first 17 lines of Table 1). Figure 4b visualizes how many of these 17 proxies are available for the dates for which we aim to reconstruct temperature. The Figure shows this for two different assumptions on uncertainty (red and grey lines, see section 2.2.3).

Figure 3 and Table 1 give a first impression of setups for additional reconstructions. We shortly describe the results for these alternative setups in our results section below. Most notably among these alternative tests are setups that use only $U_{37}^{K'}$ proxies (Figure 3). The difference between the two $U_{37}^{K'}$ setups is that E03 uses record GeoB 5901-2 instead of record M39-008 (compare Table 1 and Figure 3).

### 2.3.2 Pseudoproxies

We use pseudoproxies calculated following Bothe et al. (2019a) to test our approach. Bothe et al. provide pseudoproxies based on simulated annual mean temperature and for a global selection of grid points from the TraCE-21ka simulation (He, 2011; Liu et al., 2009). Here, we calculate the pseudoproxies for annual average data and for the chosen European-North-Atlantic domain only. The approach also provides randomly chosen pseudo age uncertainties. Following Bothe et al. (2019a) and their repository (Bothe et al., 2019b), these base on assumptions on the smoothing of the pseudoproxies and a Gaussian term.

Here, the pseudoproxy computation uses QUEST FAMOUS simulation data (Smith and Gregory, 2012). Specifically, we use the simulation ALL-5G (see tables A2 and A5). For details on this and the other QUEST FAMOUS simulations, please see Smith and Gregory (2012). The FAMOUS-HadCM3 simulations for QUEST use accelerated forcings (compare Smith and Gregory, 2012). That is, the last glacial cycle of approximately 120,000 years of climate forcing was simulated in approximately

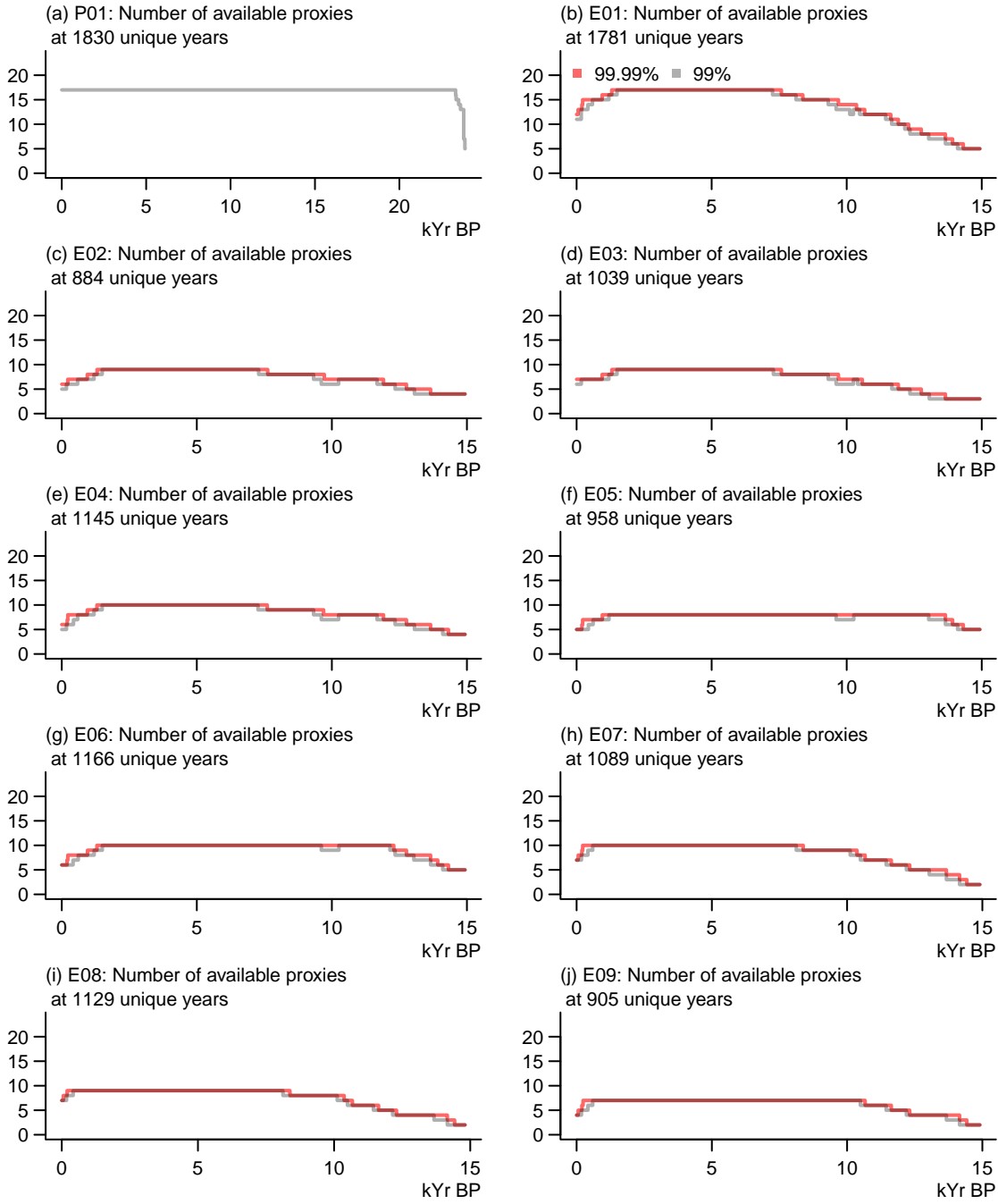

**Figure 4.** Information about the number of available proxies for the dates to be reconstructed: **(a)** the pseudoproxy setup, **(b)** to **(j)** the various proxy setups according to Figure 3 and Table 1. In **(b)** to **(j)** we show results for two different assumptions on the uncertainties, a 99% envelope and a 99.99% envelope (compare section 2.2.3).

105    12,000 simulation years. Thus, the annual simulation data is only representative of ten years of climate evolution. The data is available in monthly resolution for the full simulation period for air temperature in 1.5 meter height, and as snapshots every ten simulation years for surface temperature. We use the simulation year annual means of the air temperature data for the construction of the pseudoproxies. The FAMOUS-HadCM3 simulations use a very low resolution atmospheric model with a 5 degree latitude by 7.5 degree longitude grid. Therefore, we use the cdo application from the Max Planck Institute for Meteorology (https://code.mpimet.mpg.de/projects/cdo/, last access: 18 August 2020) to remap the data to a 0.5 by 0.5 degree

grid and use this for the pseudoproxy calculations. In this remapped data we follow Bothe et al. (2019a) and use grid point data close to proxy locations used in the realistic setup.

We modify the pseudoproxy script of Bothe et al. (2019a) to account for the reduced temporal resolution of the available QUEST FAMOUS data. This primarily means considering the default parameter settings that are given in time units. It also includes ad-hoc scaling the randomly chosen dating uncertainty to approximate the distribution of the observed dating uncer-

tainties. The latter modification also avoids that individual data points extend their influence too far along the time dimension.

The 17 pseudoproxy locations are close to the realistic proxy locations (compare Figure 1). Figure 4a visualizes the number of pseudoproxy locations with data against the dates at which we try to reconstruct values. The pseudoproxy records are shown in Figure 5. The figure also visualizes our assumptions on the uncertainty of the pseudoproxies in terms of the tolerance envelope (compare section 2.2.3).

**2.3.3   Simulations**

Table 2 provides a general overview of the various simulations in our pool of candidates. Supplementary Tables A2 to A5 give additional information. We only consider previously published simulations. These stem from a variety of projects and were performed with a variety of models. The projects are TraCE-21ka "Simulation of Transient Climate Evolution over the last 21,000 years" (Liu et al., 2009), the Paleoclimate Modelling Intercomparison Project Phase III (PMIP3, Braconnot et al.,

2011, 2012), the CESM Last Millennium Ensemble Project (Otto-Bliesner et al., 2015), the Max Planck Institute Community Simulations of the last Millennium (Jungclaus et al., 2010), and Quaternary QUEST (e.g. Smith and Gregory, 2012). We include the QUEST FAMOUS simulations only for a test case and exclude them for the main discussions due to their specific characteristics (compare section 2.3.2).

We use simulations for various different time periods to increase the candidate pool. We assume that simulation climatolo-

gies can differ over a relatively wide range (e.g., Zanchettin et al., 2014). Simulations from the TraCE-21ka and the QUEST projects are transient over periods covering the last approximately 22kyr and the last glacial cycle respectively. Otherwise, the simulations are transient over the last millennium, or time-slices for the Mid-Holocene and the Last Glacial Maximum. Additionally we also include pre-industrial control simulations. Such a multi-model and multi-time-period candidate pool effectively follows suggestions of Steiger et al. (2014). We note that considering simulations for the last millennium as candidate

for the Last Glacial Maximum can introduce climatological inconsistencies if the method identifies these fields as analogues.

We remap all simulation output to a 0.5 by 0.5 degree grid for the construction of pseudoproxies and for the search for analogues. The motivation is that thereby fewer proxies are close to the same grid point. However, resulting differences between

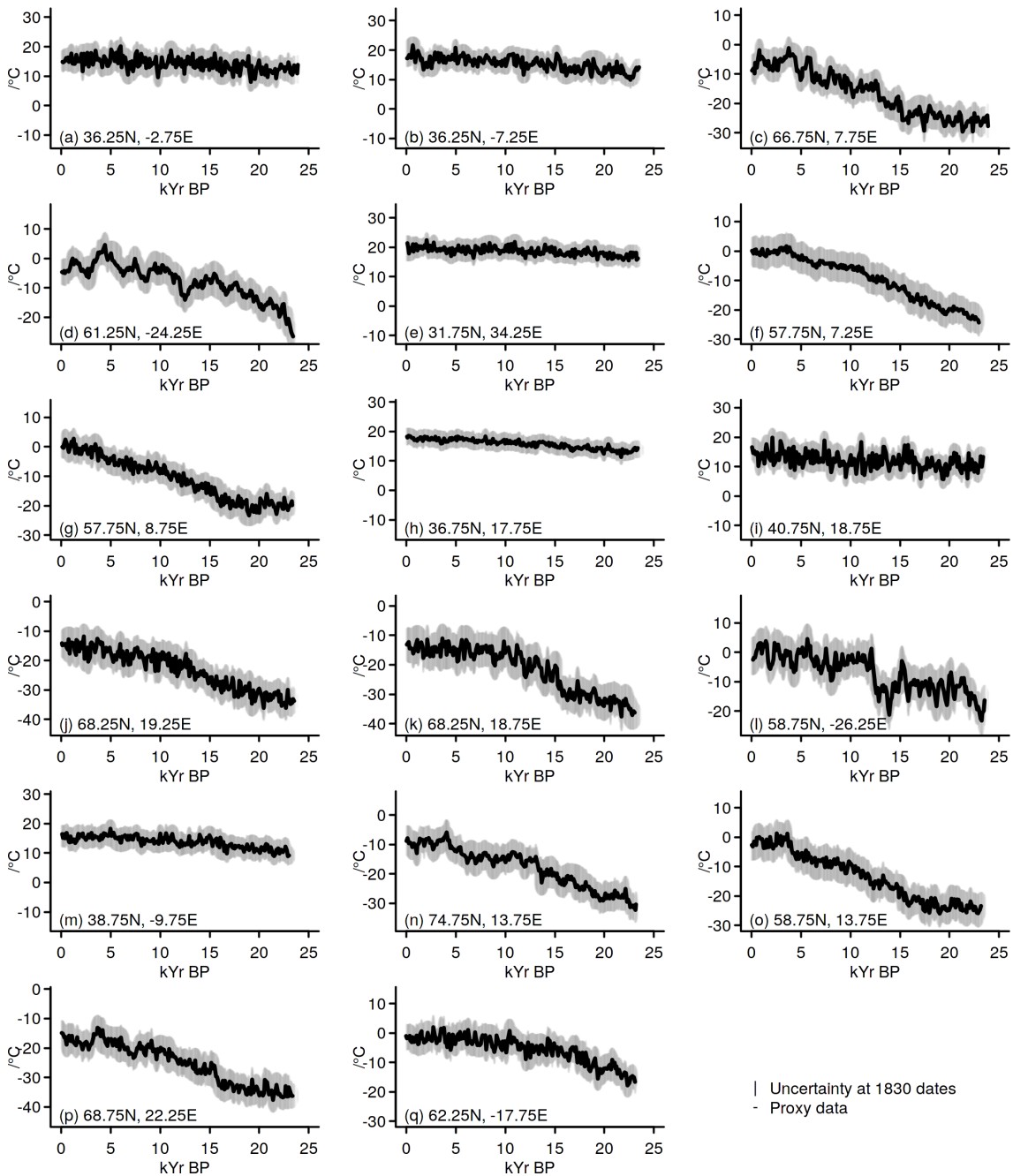

**Figure 5.** Pseudoproxy data and assumed uncertainties for the 17 locations in our pseudoproxy application.

**Table 2.** Information about the pool of simulation data: Model name, the project for which the simulations were performed, the simulated periods from this model output, the number of total years. All simulation data are remapped to 0.5 by 0.5 degree grids. References and data locations are provided in Appendix Table A2. The Appendix also lists all individual simulations used in tables A3 to A5. Note FAMOUS-HadCM3 uses accelerated forcings. We, thus, chose to exclude this simulation for most cases.

| Model | Project | Periods | Total years |
|---|---|---|---|
| CNRM-CM5 | PMIP3 | LGM, MidHolocene | 400 |
| COSMOS-ASO | PMIP3 | LGM | 600 |
| CSIRO-Mk3L-1-2 | PMIP3 | LGM | 500 |
| GISS-E2-R | PMIP3 | LGM, MidHolocene, past1000 | 9309 |
| HadCM3 | PMIP3 | past1000 | 1001 |
| HadGEM2-CC | PMIP3 | MidHolocene | 35 |
| HadGEM2-ES | PMIP3 | MidHolocene | 102 |
| IPSL-CM5A-LR | PMIP3 | LGM, MidHolocene, past1000 | 1701 |
| MPI-ESM-P | PMIP3 | LGM, MidHolocene, past1000 | 1400 |
| CESM1 | Last Millennium ensemble | past1000, pre-industrial control, industrial | 33156 |
| CCSM3 | TraCE-21ka | LGM to present | 22040 |
| MPI-ESM-Cosmos | MILLENNIUM COSMOS | past1000, pre-industrial control, industrial, projection | 5909 |
| FAMOUS-HadCM3 | Quaternary QUEST | Last Glacial Cycle | 6014 |

grid points are likely small. We use the original resolution for the final regional average reconstructions and the evaluation of field data. Local grid point evaluations are done against the remapped files.

## 3  Results

### 3.1  Pseudoproxy application

The pseudoproxy application allows highlighting the possibilities of our implementation of the analogue method. It further already provides a glimpse at potential problems.

We recapture our approach briefly. Our analogue method searches for analogues within the full pool of simulation fields but excluding the FAMOUS-HadCM3 output from the QUEST project. Pseudoproxies are derived from this latter simulation. We compare the pseudoproxy predictors to 101-year moving averages of the simulation output. We concentrate on 90% tolerance ellipses in the pseudoproxy application of the analogue search but also include results for 99.9% tolerance ellipses. Valid analogues are those simulation fields that are within the resultant tolerance envelopes for all pseudoproxy locations available for a date.

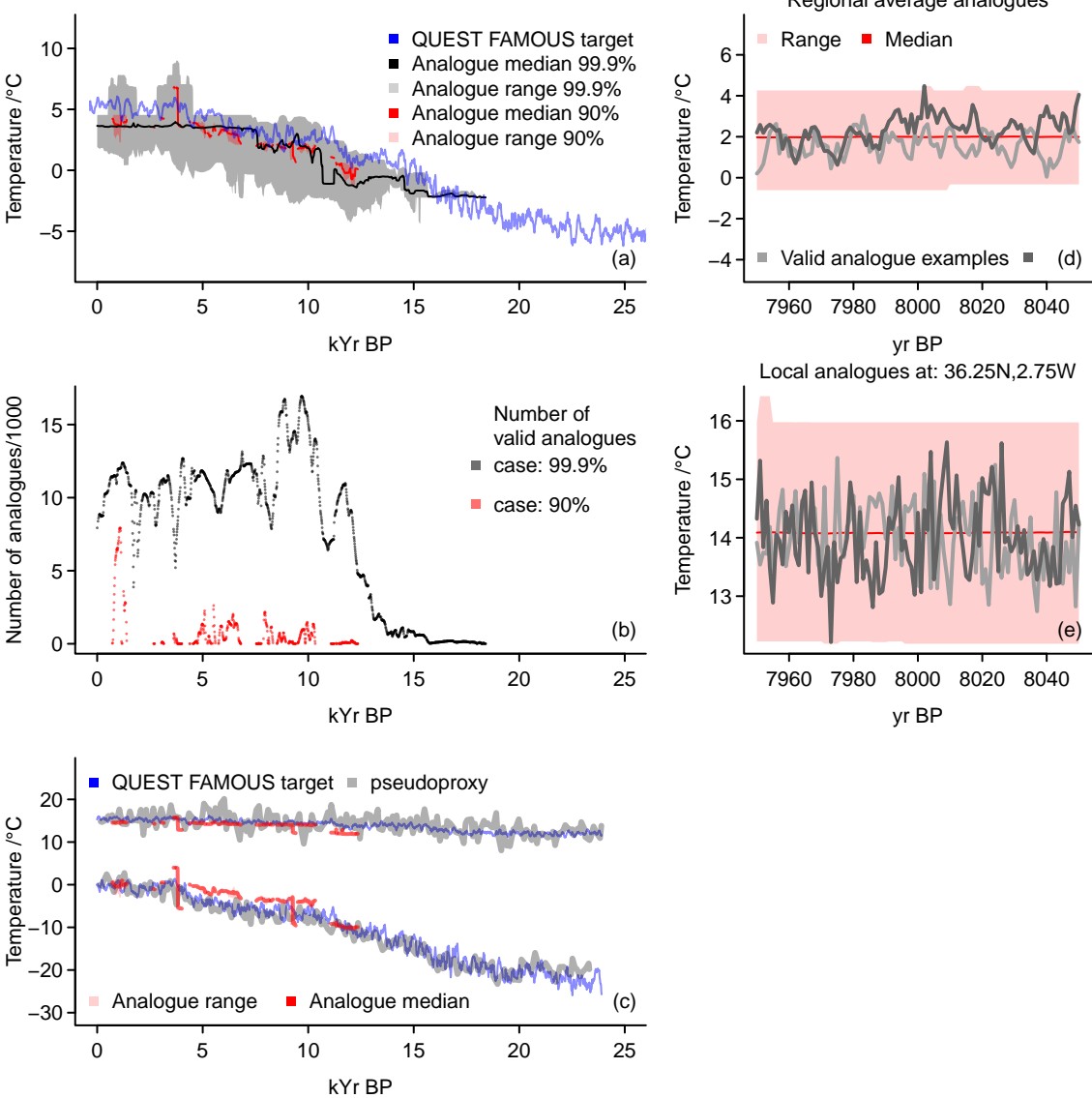

**Figure 6.** Reconstruction results for the pseudoproxy application of the analogue method: **(a)** regional averages for 101-year moving averages for two different tolerance envelope levels (90%, reds, 99.9%, greys). Lines show the 101-year moving average regional target in the TraCE-21ka simulation (blue), the median of all analogues (90%, red, 99.9%, black), and the range of all analogues for the respective tolerance ranges (colored shading). **(b)** shows the number of analogues found for each of the dates considered for both setups (90%, red, 99.9%, black). **(c)** adds 101-year moving averages of local pseudoproxy data (grey), local target data (blue), the range of all local analogue values (light red), and the local median of the analogues (red) for two locations (warmer case: 36.25N, 2.75W; colder case: 57.75N, 8.75E) and for the 90% tolerance envelopes only. **(d,e)** provide expansions of regional **(d)** and local **(e)** 101-year moving average analogues into 101-year long time-series for the 90% tolerance envelopes only. The panels show the median (red), the range (light red), and two valid analogue examples of the expansions. Due to the coarse resolution for the QUEST FAMOUS data, panels **(c)** and **(e)** use the remapped data of the simulation.

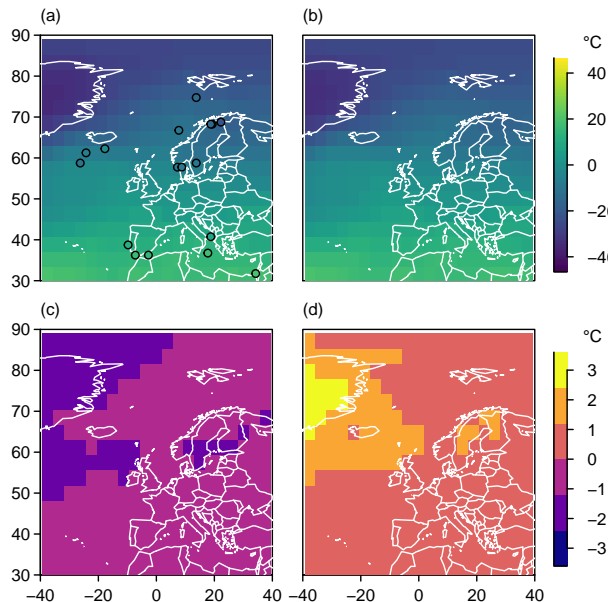

**Figure 7.** Temperature field reconstructions in °C for the pseudoproxy approach: Example of a 101-year mean annual temperature analogue reconstruction for the European-North-Atlantic sector in °C centered in the year 8,000 before 1950. **(a)** One example analogue, **(b)** local median of all analogues, **(c)** local minimum of all analogues, and **(d)** local maximum of all analogues. Panels **(c)** and **(d)** show differences to the median in °C.

Temperatures are reconstructed for the full domain of the European-North-Atlantic sector including the Arctic (Figure 1) and over a multimillennial period leading up to the late 20th century CE. Figure 4a highlights that most pseudoproxies are defined at all dates. That is, the chosen sample dates of the pseudoproxies are close to each other and, thereby, the generated dating uncertainties result in relatively large overlaps. Figure 5 presents the pseudoproxies including their tolerance envelopes.

    In this setting, the analogue search tries to identify analogues for 1,830 dates. Our implementation finds between 1 and 50  7,919 analogues at 531 dates (Figure 6b); it fails to find analogues for dates during the deglaciation and the glacial maximum. Analogues stem mainly from the Trace-21ka simulation. Occasionally, output from the PMIP3 past1000 simulation with IPSL-CM5A-LR is also classified as valid analogues.

    Results change if we consider a wider tolerance envelope. For an 99.9% tolerance envelope instead of a 90% one, we are able to find between 1 and 16,944 valid analogues at 1,438 of 1,830 dates (Figure 6b). Analogues stem from four additional 55  simulations compared to the smaller tolerance envelope. These are the PMIP3 midHolocene and lgm setups of IPSL-CM5A-LR, the COSMOS-ASO lgm setup, and the GISS-E2-R past1000 ensemble member r1i1p122.

    Figures 6 and 7 provide information on which type of reconstructions we obtain from our analogue method. Panel (a) of Figure 6 is for the area mean reconstructions for two different tolerance envelopes. It shows the resulting reconstruction medians in black and red for 99.9% and 90% tolerance assumptions respectively. The blue line in the panel is the 101-year

moving average regional temperature from the simulation, i.e. the reconstruction target. Shading in the panel shows the full range of potential analogues.

The results are encouraging but problems are obvious. We are able to find valid analogues for both tolerance ranges.

Analogues are regularly relatively close to the target for the narrow tolerance range. However, their number is often small and there are periods without any valid analogues. The range does seldom include the target. Further, the reconstruction with a

65 narrow tolerance assumption does not provide valid analogues earlier than approximately year 13,500 BP.

On the other hand, the range of potential analogues is only weakly constrained for the wider tolerance range. For example, the analogue search may regard more than 17,000 records of the TraCE-21ka simulation as valid analogues around the year 10,000 BP (compare Figure 6b). This wide range often includes the target. However, the target is mostly above the median estimate. The reconstruction gives a rather constant estimate from a small number of analogues for the period earlier than 16

thousand years before present.

The pseudoproxies, together with their uncertainties, are a weak constraint during most of the period of interest if we assume a wider tolerance but they fail to capture the target if we assume a stronger knowledge about their value. In addition, the reconstruction envelopes and medians show rather little variability and often give nearly constant values over long periods. That is, the set of valid analogues has a notable overlap for these periods. The lacking variability among analogues together

with the potentially wide range of analogues is reflected in the small variability in the reconstruction median.

Besides the regional average, the results allow to extract the local representations. Figure 6c shows two examples for the narrow 90% tolerance assumption. These are for the pseudoproxies at 36.25N,2.75W and 57.75N, 8.75E. We refer to those as the warmer southern and colder northern locations respectively. The panel plots again the target simulation output in blue, the full analogue range in light red, and the analogue median in red. We also add the pseudoproxy in grey.

At both locations, the range is very small for the narrow tolerance range. At the southern location, the reconstruction median is generally below the target and the range is hardly identifiable and does not include the target. This is comparable to the northern location, where, however, the median is generally above the target. Even for the wide tolerance range the target is more often outside than within the full analogue range at the southern location while at the northern location the range includes the target regularly (not shown). Thus, the range of potential analogue cases is still relatively narrow at the southern location but can be already quite wide at the northern location. Also locally, analogue range and median show little variability. In the northern case, the analogue medians fail for both tolerance assumptions to capture the average characteristics of the pseudoproxy except for approximately the most recent 3kyr.

5 The pseudo-reconstruction results suggest that the approach can provide local information in addition to the regional average. Relatively wide tolerance appears to be necessary to capture the local characteristics at the two chosen locations. This is more successful for some periods but success always varies regionally.

Since we search analogues among temporal moving window averages, the analogue search provides one more result of interest. Any analogue state represents a temporal average. Since we also know the period that has been averaged, we can provide

10 the climatic time-varying sequence. This informs us about the time-variations underlying the analogue average climate state. That is, we obtain climate evolutions that comply with our proxy-constraints. This, for example, allows to get an impression of

how temperature changed on sub-centennial, e.g. interannual timescales, or to obtain an estimate of the interannual variability. Panels (d) and (e) of Figure 6 provide such expansions of 101-year average states into 101-year time-series. They do so for a narrow tolerance assumption. The panels show the range and the median of 101-year series for all found analogues for one specific year. They also add two examples of 101-year time-series. Panel (d) is for the regional average, and panel (e) for the grid point at 36.25N, 2.75W. Both show 101-year expansions around the average centred in the year 8,000 BP.

Although we consider a narrow tolerance range, which results in very narrow ranges around the mean analogue state, the expanded range of potential analogues is still notably wide. The two examples of valid analogues highlight how much two climates may differ over the period although both are valid analogues considering the proxy uncertainty. Wider tolerance ranges give larger ranges of reconstructions and result in larger differences between the 101-year time-series.

Finally, our reconstruction approach allows considering the spatial fields of valid analogues. Figure 7 adds an example for 101-year mean annual temperature. It shows one valid analogue field in panel (a) and the local median, minimum, and maximum values of all analogues in panels (b) to (d) respectively. The chosen date is the year 8,000 BP for the narrow tolerance range. Panel (a) also adds the values for the pseudoproxies that enter the analogue search. The example analogue and the pseudoproxies agree to some extent but disagreement is notable south of Iceland. There are more than 1,000 analogues for this year. Their local range at no point exceeds 4 degree Celsius for the narrow tolerance setup. Local positive deviations from the median may differ most strongly over Greenland and in Scandinavia. In the latter region, proxies should constrain our search for analogues. Local negative deviations may become largest over comparable domains. We do not show the equivalent Figure for the wide tolerance assumption but note that in this case the local range of results may exceed 20 degree Celsius and that largest positive excursions occur southwest of Svalbard, where a proxy constrains our search. The largest negative excursions are located at the eastern border of our domain in the Barents Sea, where our search is effectively unconstrained.

The pseudoproxy application of our implementation of an analogue search shows the viability of such approaches for reconstructing past climates from spatially sparse proxies with temporally sparse, irregular, and uncertain ages. The pseudoproxy tests also show that the results depend on our assumptions on how tolerant we are with respect to our confidence in the proxy-input. Overall, the pseudoproxies are only weak constraints on the potential climate.

## 3.2    Application to real proxies

Already the pseudoproxy test highlights the potential but also the associated problems in using the analogue method for the type of proxies we are interested in, together with a limited pool of candidate fields. The analogue reconstruction is able to capture the target data but the search may provide either a very wide or a too narrow uncertainty range relative to the target. Wide ranges occur mostly due to the large number of valid analogues while narrow ranges signal that there are only few analogues fitting the proxy data under the made assumptions on the fidelity of the proxies. The method may overall fail to provide valid analogues.

Our focus here is on a multi-archive and multi-proxy reconstruction using 17 proxies (compare section 2.3.1) for the European-North Atlantic sector for approximately the last 15kyr. Preliminary tests showed that using a 90% tolerance level leads to ranges that are too narrow to find any suitable analogues (not shown). We only show the results for using 99% and

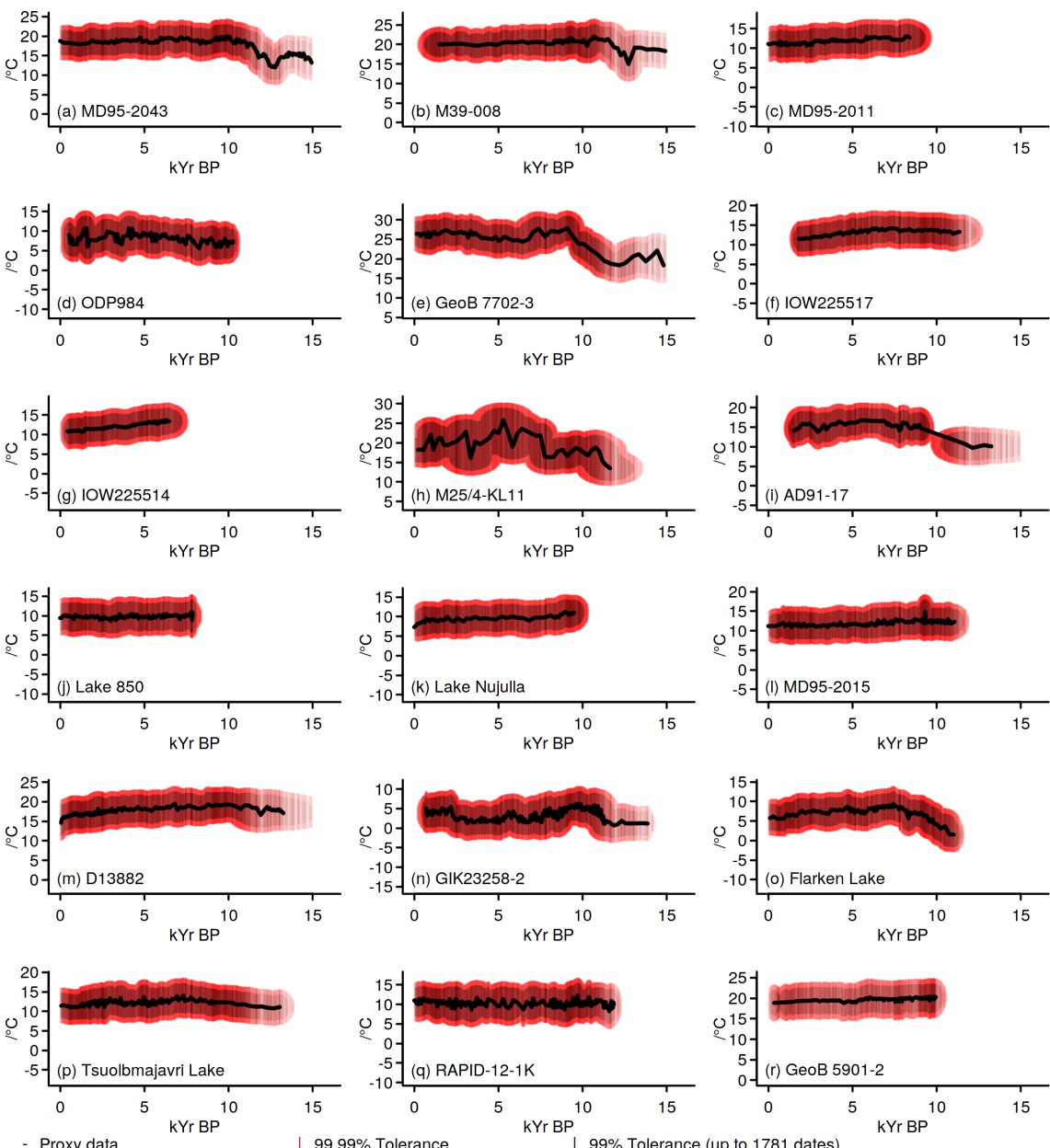

**Figure 8.** Proxy data and assumed uncertainties for all proxy record locations in our analogue search under two different tolerance envelopes.

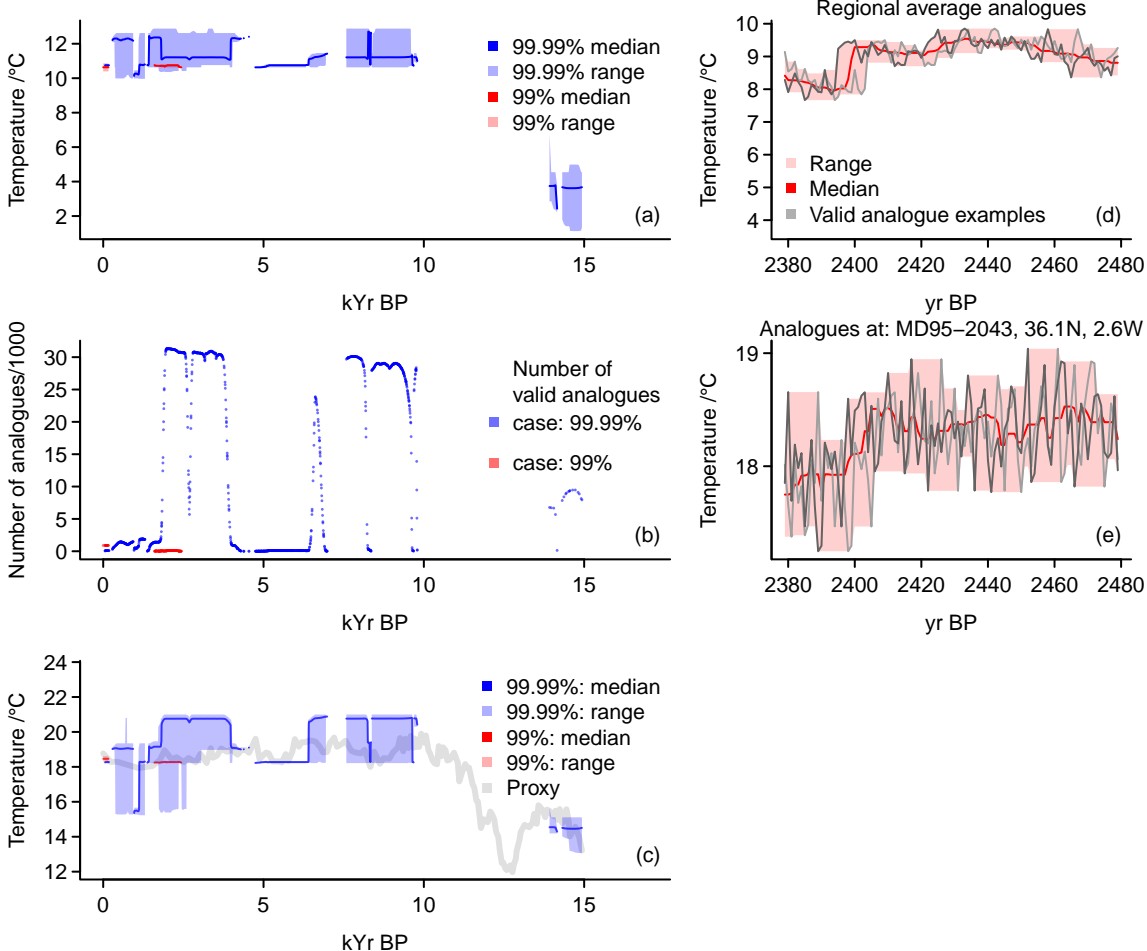

**Figure 9.** Reconstruction results for the analogue method under two different tolerance assumptions: **(a)** shows median and range of all analogues of regional averages for 101-year moving averages for an assumed 99% tolerance envelope (red) and a 99.99% envelope (blue). **(b)** provides the number of analogues found for each of the dates considered; red: 99% envelope, blue: 99.99% envelope. **(c)** adds for the location of the record MD95-2043 the proxy data (grey), and the range and median of all local analogue values for a 99% envelope (red) and a 99.99% envelope (blue). **(d,e)** gives expansions of regional **(d)** and local (36.1N,2.6W, MD95-2043) **(e)** 101-year moving averages in 101-year series ranges. They give the median (red), the range (light red), and two valid analogue examples of the expansions. Both panels only show results for the 99% tolerance envelope.

99.99% tolerance levels in the estimation of our tolerance envelopes around proxy records. For the meaning of these levels see the descriptions for equation 1.

Figure 8 shows the proxies and their constructed tolerance envelopes for the locations in Figure 1. The panels highlight that the real proxy values are less equally distributed through time, are generally smoother, and differ more in their lengths compared to the pseudoproxy setup. Figure 4b already showed how the number of available proxies increases from 11 to 17 but then again decreases until only 5 proxies are available for the earliest dates. Below we compare the full 17-proxy setup to different sets of proxies. Table 1 and Figure 3 to 4 give details for the different sets.

In the case of the main set of 17 proxies, our implementation tries to find analogues for 1,781 dates. There are between 1 and 900 analogues for 141 dates for 99% tolerance envelopes (see Figure 9b). Analogues come from two different simulations. It is obvious that the method often fails to provide a valid analogue.

For the 99.99% envelope, these basic results change. The method identifies 1 to 31,304 analogues at 1,288 dates (see Figure 9b). These come from 42 different simulations. There are no valid analogues between ~10kyr BP and ~14kyr BP. Otherwise, there are extended periods with very many analogues and other periods with few analogues.

For the narrower tolerance assumption, the method finds valid analogues only for the recent past millennia (Figure 9). Even then, it is only successful for few periods (Figure 9b). In this case, the range of the area average reconstruction (Figure 9a) and at the local proxy location (Figure 9c) is very narrow. There is very little regional or local temporal variability in the analogues. However, the reconstruction may reflect well the average state of the local proxy series (Figure 9c). As for the pseudoproxy test, we can expand the analogues, i.e., the 101-year moving means, to show the underlying time-variations (Figure 9d and e). These again provide an impression of interannual variability that is compliant with our proxy constraints on the centennial average. These panels emphasize the very narrow range of potential analogues for the regional average but also for the local series. For the chosen year, there is only a small number of analogues, which form a sequence of consecutive simulated years from one simulation. Therefore, the two examples in panels (d) and (e) are simply time shifted sequences.

For the wider tolerance envelope, the method identifies valid analogues for more dates (Figure 9) and, generally, there are more valid analogues for these dates (Figure 9b). However, there are more proxies available for some dates (compare Figure 4) and this increases the number of constraints on the analogue candidates for these dates. Thus, there are dates when the range of the regional average reconstruction for a 99.99% tolerance envelope does not necessarily include the 99% envelope data.

The range of the reconstruction may be regionally or locally wide for the 99.99% envelope, but this does not ensure that it locally includes the proxy values (Figure 9c). There is little temporal variability in the reconstructed data. This is mainly because of the large number of analogues and the relatively low temporal variation in the set of valid analogues (Figure 9b). Further, the reconstruction is rather constant.

Figure 10 plots examples of a field and of the local minima, median, and maxima of potential analogues for the two different tolerance envelopes. The upper row uses the 99% envelope reconstruction for the year 2,429 BP and the lower row uses the 99.9% envelope reconstruction for the year 14,105 BP. For both dates, all valid analogues are from only one simulation each. The examples in panels (a) and (e) of Fig. 10 also include as dots the proxy values available for the respective dates. These highlight that, for the late Holocene date, the found analogues capture the proxies rather well though with exceptions over

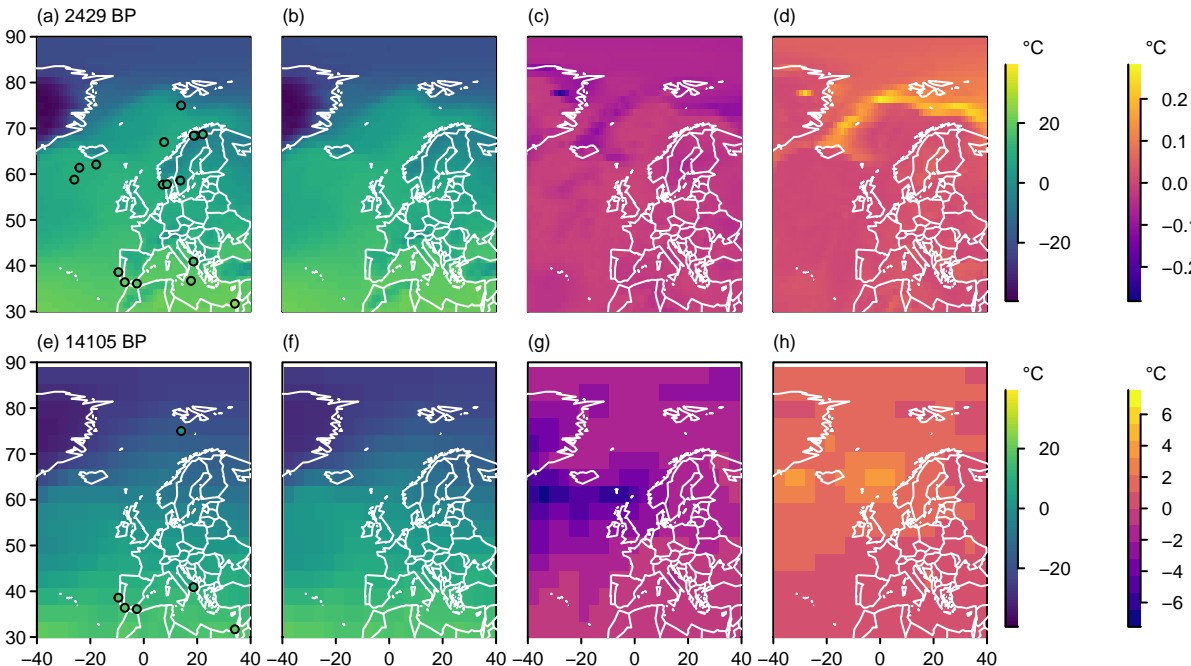

**Figure 10.** Field information for the analogue search: Two examples of 101-year mean annual temperature analogue reconstructions for the European-North-Atlantic sector in °C. **(a,e)** One example analogue, **(b,f)** local median of all analogues, **(c,g)** difference of local minimum to local median of all analogues, and **(d,h)** difference of local maximum to local median of all analogues. **(a)** to **(d)** are for the 99% tolerance envelope and the year 2,429 BP, **(e)** to **(h)** for the 99.99% tolerance envelope and the year 14,105 BP.

Scandinavia. However, the analogues for ~14kyr BP strongly disagree with the one proxy at high northern latitudes. The range of analogues is very narrow for the late Holocene example from the narrow tolerance case. Differences become largest over Greenland and along the sea-ice edge. For the deglacial example and the wider tolerance case, differences become largest east and west of Iceland.

### 3.2.1 Results for different proxy setups

Table 1 introduces a number of additional proxy setups (E02 to E09). These use different sub-selections of proxies from our initial selection. Further, most of them test sparser sets of locations around central Europe (compare Table 1). Figure 3 provides additional information about which records are included in the different setups and their proxy types. Here, we shortly present the results.

Experiment E01 is our main setup. It was described in the previous section. It uses the 17 chosen proxy locations, which we also use for the pseudoproxy setup. Setups E02 and E03 are based only on alkenone $U_{37}^{K'}$ records and E03 replaces M39-008 by GeoB 5901-2, as both are co-located. E04 to E09 include different numbers of other proxy types instead of $U_{37}^{K'}$. Figure 4

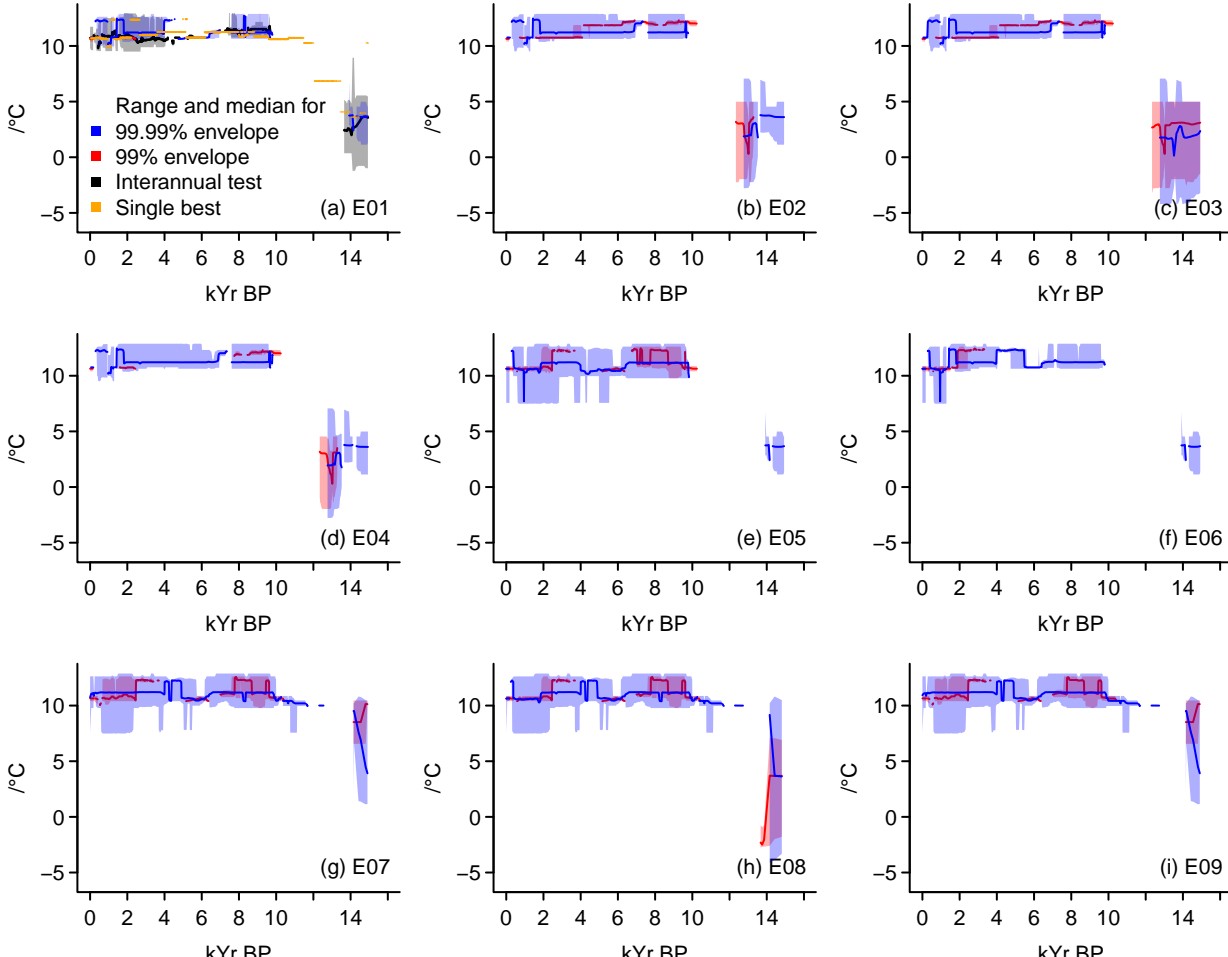

**Figure 11.** Visualising the reconstructions for the various proxy setups: **(a)** E01, **(b)** E02, **(c)** E03, **(d)** E04, **(e)** E05, **(f)** E06, **(g)** E07, **(h)** E08, **(i)** E09. All panels include the median and the full range for the reconstructions under a 99% tolerance envelope (red) and a 99.99% envelope (blue). Panel **(a)** additionally includes a setup in black where we do not consider 101-year moving averages of simulation data but all simulation output as provided including the FAMOUS-HadCM3 simulations for QUEST. Orange points in panel **(a)** are for a test considering only the single best analogues for each date.

shows the availability of proxies for the different setups. Figure 8 presents the proxy data and assumed uncertainties including record GeoB 5901-2. For more information see Table 1 and Figure 3.

95      Figure 11 shows the reconstruction results for the proxy setups E01 to E09. All panels plot the reconstructions using the 99% and the 99.99% tolerance envelopes. Panel (a) adds for our main setup a reconstruction where we consider interannual data for the simulations and include the QUEST FAMOUS-HadCM3 simulations. The panel also includes the results of testing an analogue approach where only the single best analogue is considered at each date.

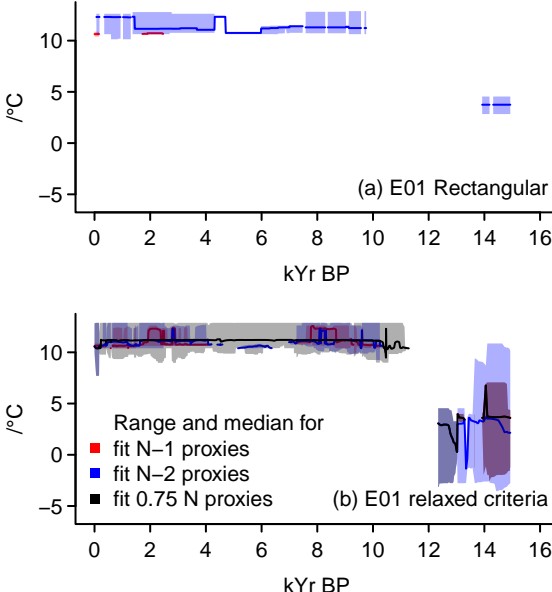

**Figure 12.** Visualising alternative reconstructions: **(a)** E01 with rectangular tolerance range, **(b)** E01 with relaxed criteria for analogue selection. Panel **(a)** includes the median and the full range for the reconstructions under a 99% tolerance envelope (red) and a 99.99% envelope (blue). Panel **(b)** shows range and median for setups where the analogue candidates are valid even if they fail at one (red) or two (blue) locations or at 25% of all locations (black).

The panels of Figure 11 highlight that loosening the tolerance constraint and thereby widening the tolerance envelope leads to valid analogues for notably more dates as well as a wider range of valid analogues. We also obtain analogues at more dates if we keep the tolerance envelope at the lower level but do not preprocess the simulation output to 101-year moving means (Figure 11a black lines). This inclusion of interannual data increases the number of analogues throughout the reconstruction period. This variation of the experiment also uses more simulation data by including the QUEST FAMOUS data, but this only affects the reconstruction success in the 15th millennium BP in this setup. We performed further tests with different averaging periods of 51 and 501 years, respectively, while keeping the narrow tolerance envelope (not shown). Increasing the averaging period to 501 years reduces the number of valid analogues and the number of dates with any valid analogues. Reducing the averaging period to 51 years allows to find a few valid analogues in the 15th and 16th millennium BP. In this setting, the approach also finds more valid analogues in recent millennia.

Generally, the method appears to provide more complete reconstructions among our proxy setups for those that only include $U_{37}^{K'}$ records (Figure 11b,c). That is, such consistent sets of proxies provide a more continuous reconstruction for both local tolerance assumptions. Nevertheless, we fail to obtain valid analogues, i.e. reconstructed values at the end of the deglaciation. While results are quite similar over much of the period between both reconstruction attempts E02 and E03, the second setup allows a wider and potentially colder range in the period before ~12kyr BP.

Further panels of Figure 11 add different setups. Panel (d) complements the $U_{37}^{K'}$ proxies by one foraminiferal assemblage record. Panels (e) and (f) also test different setups dominated by $U_{37}^{K'}$ but including other proxies. Panels (g) to (i) use different small setups of proxies around the European area.

Multi-archive setups with fewer proxies give generally wider ranges of possible analogues. Otherwise all setups tend to be in a comparable range regarding their median and their range considering the last 10 millennia. Differences between all setups are largest in the 14th millennium BP due to a larger range for some reconstructions.

Both multi-proxy setups in panels (e) and (f) fail to provide analogues before the deglaciation for the narrower tolerance assumption. The setups in panels (g) and (i) are notably warmer in the 14th millennium BP compared to results in panel (h) but also compared to other setups. This holds for both tolerance envelopes. A common difference is the inclusion of M39-008 while excluding the $U_{37}^{K'}$ record D13882 (compare Table 1 and Figure 3). The latter record is thought to represent summer temperatures off the west coast of Portugal while the former is meant to represent annual temperatures in the Gulf of Cadiz. We note that panels (e) and (f) also are warmer compared to other setups in the 14th millennium BP for the wider tolerance range. These also include M39-008 and exclude D13882. Please note, the supplement of Marcott et al. (2013) refers to D13882 as D13822.

Generally, we find that the reconstructions from different setups differ in their ability to reconstruct climate for specific periods. Indeed, different setups may provide notably different climates, particularly for the early part of the time period of interest. Particular proxies appear to shift the results for the earlier part of our reconstruction between a warmer and a colder deglacial estimate. It is beyond the scope of this paper to disentangle the reasons for this. All setups provide rather constant reconstruction ranges.

As noted, Figure 11a adds a single best analogue reconstruction, where the reconstructed value for a given date is the analogue candidate with the smallest Euclidean distance to the proxy values for that date. During the past approximately four millennia as well as during the period from eight to 10 thousand years BP, the single best estimate is included in the ranges of our other reconstruction efforts. Indeed it is close to the test with interannual data throughout the Common Era of the last 2,000 years.

In the period between four and eight thousand years BP, when other approaches give very narrow ranges due to few valid analogues, there are cases when the result from the single best analogue setup differs notably from the other efforts. However, it is still within the range of results from the other experiments for earlier and later periods. Such deviations from the tolerance area approach are reasonable since our construction of the proxy values for the single best analogue search can provide a notably different proxy state compared to the tolerance envelopes constructed for our standard approach. Another potential explanation is that the analogue that minimises the overall distance may be outside of one or even multiple tolerance ranges. Finally, we already mentioned that changing a tolerance level may change the number of proxy locations included in a search. For example widening a tolerance level, may result in inclusion of more proxy locations for specific dates. The construction of the proxy values for the single best search similarly changes the underlying multi-dimensional proxy vector. Indeed an inspection of the data indicates that in our test case, the found analogue does fall outside the tolerance ranges at least at one location.

We also note that the single best analogue approach allows to obtain estimates when the other approaches fail between 10 to 14 thousand years BP. Comparably to our other reconstruction attempts, the single best analogue reconstruction shows only little variability. Noteworthy are the reconstructed values in the 15th millennium BP where the single best analogue represents a Holocene level warm climate and not a deglacial climate.

We consider two more modifications of our approach. Figure 12 shows, first, results using a rectangular tolerance region, and, secondly, reconstructions for tests where an analogue candidate is valid although it falls outside the tolerance region at one, two, or 25% of the locations. The rectangular setup has minimal influence on the reconstruction but gives more homogeneous ranges of valid analogues and succeeds at slightly more dates in finding valid analogues. Relaxing the tolerance criteria results in very wide ranges in the early part of the study period. Due to few available proxies in that period, the criterion to fit 75% of locations is stricter than the criterion that allows to fail at two locations. The resulting reconstructions still have little variability. They also either give a wide and nearly constant range of potential values or a very narrow range.

## 4   Discussions

Our implementation of an analogue search method for reconstructing surface temperature over multimillennial timescales relies on a number of decisions, which are uncommon compared to other paleo-reconstruction efforts on multimillennial timescales. Central to our assumptions is that taking account of the uncertainty in our underlying data is indispensable in analogue approaches for paleoclimatology and, particularly, if one uses spatially and temporally sparse as well as data and age uncertain proxies. There is one prime motivation behind our specific handling of uncertainty in terms of tolerance ranges and our selection of reconstruction dates: The analogue search for a chosen date should use as much information about this date as possible, including the uncertainty of other data points whose age uncertainties include the currently given date of interest.

This leads to the use of tolerance ellipses. Assumptions here are that, firstly, data and date are inseparable; secondly, that this assumption also holds for the tuple and its two-dimensional uncertainty; and, thirdly, that a reconstruction exercise has to consider both parts of the uncertainty to sufficiently estimate the range of reconstructed values. Admittedly, our procedure is a simplified approach to incorporating these assumptions. More correctly, one would calculate the multivariate joint distribution, and use a measure of likelihood to select the analogues. As a side note, the highly dimensional space for all proxies also follows a multivariate distribution, which one could then employ in more sophisticated data science approaches.

We trust that considering both parts of the uncertainty enables better and more reliable reconstruction estimates. We concede that this procedure may exaggerate the range of potential climates and thereby may reduce the precision of the reconstruction (compare also Annan and Hargreaves, 2012). We postulate that this, however, is only partly due to the assumptions on uncertainty, which may transfer uncertainty to too many records. We think it is also because the simulation pool is not fully consistent with all the proxies simultaneously. It is beyond the scope of the present study to investigate whether this, in turn, is because of unreliable simulations, lacking overlap between reconstructed and simulated climates, or lacking reliability of the proxy records, that is their errors.

With respect to the lacking precision of the reconstructions, Annan and Hargreaves (2012) already identified a similar issue in their particle filter data assimilation approach. Annan and Hargreaves note that in a setup where one has only few and highly uncertain proxy predictors the reconstruction tends to lack accuracy. We think that for the analogue method one could remedy this by weighing the valid analogues by a distance measure relative to the pattern of proxy predictors or by their agreement with each individual predictor. We note that the analogue method in the present setting may represent the recent climate worse than simply taking the average over the period of instrumental observations.

Our handling of uncertainty in terms of tolerance results in difficulties in implementing a distance measure like the Euclidean. A more formal definition of similarity should take into account the multivariate and correlated nature of uncertainty: in time and across proxies.

Our choice of elliptic tolerance regions may seem counterintuitive. Mainly, two related arguments are imaginable. First, the idea can be proposed that time and data are independent and a uniform rectangular selection criteria could be suggested. We address this already in the description of the method. We here concentrate on another argument. Following this second argument, our uncertainty about the value should not shrink at the border of our temporal uncertainty range but should become wider there, as we are less confident that the data value even is valid there. This also assumes an independence of dating and data and their uncertainties. However, our argument for the ellipse is the following. We regard our time-date point as sampled from a two-dimensional distribution. If we regard this to be a uniform distribution, we would also use a rectangular tolerance area. However, we regard the distribution as a two-dimensional Gaussian, which can be visualised as an ellipse in the two-dimensional plain. Thereby the probability density for a valid point reduces further away from the best estimate. If our analogue pool would well sample the climate space, we could weigh our time-data points by their likelihood within the two-dimensional Gaussian plain. Then values that are far off in either or both dimensions would be given less weight. However, as we have only a rather small candidate pool, we resort to a binary criterion of inclusion and exclusion.

Related to our handling of uncertainty is our approach of reconstructing data for those years when at least one proxy predictor is dated. This also may contribute to the wide range of the reconstructions by neglecting information in between these dates. Alternatively, one could pool the proxy dates into constant intervals of, for example, 100 years. The underlying assumptions here are as strong as in our procedure. We note that Jensen et al. (2018) use the published age models to interpolate their proxy records to consistent time steps. They compare their proxies to 10-year averages of the simulation pool. Incorporating, presumably Bayesian, age models maximises the available prior information used. Nevertheless, we decide against interpolation procedures, even based on Bayesian age models, assuming that this may result in overconfident reconstructions. For example, interpolation could suggest more certainty in reconstructed values where and when we have little or no proxy information (see, e.g. Figure 8i between approximately 9kyr and 11kyr BP).

Additional assumptions relate to characteristics of the considered proxy predictors. This includes our decision to generally compare the proxy predictors to centennial averages of the simulation output. Thereby, we do not allow for the fact that the proxy sensor might record extreme-like events. Similarly, we also do not consider the differing resolutions for each date and each location. Further, we compare the proxy predictors and the simulation pool in terms of temperatures instead of using surrogate proxies in proxy units from the simulation pool. Finally, the use of temperature for the surface and for an attributed

and calibrated season does not account for the sensor specific habitats and seasonal sensitivities or their changes (compare Jonkers and Kučera, 2017; Kretschmer et al., 2018). That is, while we make assumptions about, e.g., seasonality, these do not account for the possibility that the recorder changes its seasonality adaptively relative to environmental conditions. Our comparison, thus, is based on the assumption that the proxy inferred climate property and the proxy record relate reasonably well to the parameter of interest (annual surface temperature) and that, in turn, comparisons to the equivalent simulated output are valid. In doing that we rely on the previously published information about the considered proxy record. Similarly, our expansion of the temporal average reconstructions into 101-year time-series relies on the quality of the proxy data and on appropriate assumptions on the temporal representativeness of the data. The possibility for such a temporal downscaling is a unique feature of analogue search reconstructions from temporal averages and of comparable data assimilation techniques.

Possible improvements of the method would respect more explicitly the irregular resolution of the proxy records and the different resolutions between the records. Similarly, applications benefit if we can discriminate whether a proxy sensor records mean climatic conditions or extreme-like events. Including the proxy specific habitat and growth season also leads to a more appropriate comparison as does employing proxy forward models to make the comparison in proxy units.

Better understanding of the proxy systems and availability of the full simulation output data would allow for analogue searches that are more specific for each proxy series. It further would enable the use of locally calibrated process-based forward integrations by proxy system models. The advent of proxy system forward models in principle allows producing proxy parameter representations in the virtual environment of the simulations (Schmidt, 1999; Tolwinski-Ward et al., 2011; Thompson et al., 2011; Evans et al., 2013; Dee et al., 2015, 2016, 2017, 2018; Jones and Dee, 2018; Dolman and Laepple, 2018) but there are still gaps in the understanding on how the sensor recording of the biological, physical, chemical, or geological process reacts to the environment. Additionally, records may lack necessary information. While such applications are quickly developing (see Dee et al., 2016; Jones and Dee, 2018; Dolman and Laepple, 2018; Konecky et al., 2019), data assimilation of this kind of information is still not operational even for the Common Era with its potentially high resolution and potentially high quality proxies (Hakim et al., 2016; Tardif et al., 2019; Emile-Geay et al., 2017).

It is generally advisable to use consistent proxy parameters, a consistent recalibration, and a consistent calibration target. This should increase the probability of the proxy predictors constraining the pool of potential analogues (compare the results in section 3.2.1). Often such consistency is an implicit or explicit assumption (compare, e.g., Reschke et al., 2019). On the other hand, the analogue approach, in theory, should allow using different parameters and calibrations if the comparison is to the same target. Indeed, ideally, it should also compensate even a comparison of different parameters. This, however, depends on how much proxy records indeed constrain the ultimate target property for the reconstruction.

Our reconstruction is only for the approximate domain of the proxy predictors. However, it may be possible that a set of proxy predictors from, for example, Europe also provides information on larger scale climate variables. Further, we deal only with temperature reconstructions. However, climate is more than simply temperature. Indeed, if there is evidence that the proxy predictors are relevant constraints on other climate fields beyond, in this example, temperature, the pool of analogues can provide information on other climate variables.

However, reconstructing other variables for hydrology or climate dynamics depends on a sufficient number of proxy records that reliably represent these. That is, there are two conditions on the proxy records, they have to represent the variable and there have to be enough of them. In addition, we have to be confident that the simulation pool reliably represents the climate variable and its spatial distribution. Considering the number of available reliable proxies for, e.g., precipitation and the quality of simulations' representation of it, we would expect that reconstruction success using the analogue method may be worse for these other variables than for temperature (compare also Gómez-Navarro et al., 2015).

Regarding the temporal resolution, a test of our method suggests that, for a given assumed tolerance level, the analogue search is more successful in finding valid analogues if we consider higher resolution data and less successful if we reduce the resolution of the data. That is, the method performs slightly better in finding valid analogues when we use 51-year averaged simulation data than when we use 101-year averaged data, and it is even more successful in finding valid analogues using interannual data. While such an interannual analogue search may misinterpret what the proxy data represents, it may be a more truthful comparison considering the potential level of environmental noise in the proxy data relative to the targeted temperature signal.

Similarly, we find more valid analogues if we use less stringent criteria in our search for valid analogues. A single best analogue reconstruction also gives a more continuous reconstruction.

However, all approaches have in common that reconstruction medians as well as reconstruction ranges are relatively constant over time. The reconstructions show little variability and are lacking clear differences in climate between the late and the early Holocene.

A likely reason for the small variability in central estimates and the generally rather constant character of our reconstructions could be that the space of valid analogues is too unconstrained and the method labels too many candidates as valid analogues. However, also the single best approach shows such a behaviour. That is, while the reconstruction is undoubtedly only weakly constrained, even the best analogues differ little between subsequent dates. Part of this may be due to our choice to consider a rather large temporal range of influence of individual dated records. Our ellipses of tolerance may result in a strong influence of an unlikely value at a specific date. This could potentially be solved by considering explicitly the likelihood of a value at a date instead of simply taking a binary criterion. A less complex solution could be obtained by pooling proxy values in temporal windows, weighting them within these windows, and then performing a reconstruction considering specific ranges of tolerance.

Our aim here is to use the local proxy uncertainty to select analogues. There is a trade-off between considering the uncertainty of the proxies and constraining the number of analogues. That is, if we want to consider the uncertainty in the way we do, then we allow for weakly constrained analogue ranges. If we allow different levels of proxy uncertainty, we can choose only the best $M$ analogues. We, in turn, can limit the number of analogues or weigh them by particular criteria, e.g., based on their distance to individual proxies or their overall Euclidean distance.

Beyond these methodological aspects the size and character of the pool of analogue candidates influences the quality of the results. Indeed, the lacking sensitivity to differences in climate and the lacking variability in our results may be a sign of an insufficient pool size or an insufficient overlap between simulated climate and the environmental conditions described by the proxy records.

Our results suggest that a pool including mid-Holocene, Last Glacial Maximum, and transient deglacial simulations does not ensure finding valid analogues for the time-period of the deglaciation and the Holocene. An insufficient large pool of candidate analogues requires more tolerant assumptions on uncertainty to obtain valid analogues. Thereby, the analogues remain unconstrained. A small pool also allows for non-uniqueness of analogues. Additionally climatological inconsistencies become more likely if the range of simulated periods in the model pool is wide.

We do not use anomalies. If there was a large ensemble of simulations over our period of interest, the use of anomalies would be advisable. Similarly, if all proxy records had common modern age data, there might be a valid anomaly building process. However, we include simulations for time-slices with notable different climatologies, and proxy records begin at various modern dates. One solution could be a sliding climatology for the proxies, which is added again for the final reconstruction. We note that, if we want to apply proxy forward models based on the calibration between measured property and temperature we do not use anomalies either because calibration relations frequently need temperature on either the Celsius or Kelvin scales.

This section outlined a number of potential improvements of the approach. Some of these would increase the number of necessary computations. While the increase in costs is not prohibitive, we decided against including such procedures here. However, it appears particularly worthwhile to try to implement a workflow that combines feasible data science methods, some version of simple data assimilation, and a proxy system model framework like PRYSM (Dee et al., 2015, 2016, 2018; Jones and Dee, 2018) in future attempts of spatiotemporally resolved reconstructions if the interest is in a dynamical understanding of the climate variability over multimillennial timescales.

## 5 Summary and concluding remarks

The analogue method is a computationally cheap data assimilation approach. Here, we discuss a specific application for time uncertain, sparse, and irregularly sampled proxies. We focus on the North Atlantic sector and the time period from approximately 15kyr BP to the late 20th century.

The approach succeeds in providing reconstructions in a pseudoproxy setup for some past dates. Already this setup highlights two potential problems. The method may either fail to find valid analogues or provide a wide range of potential analogues, which do not necessarily include a target climate. These problems relate to assumptions on the uncertainty in the proxy input data.

The approach performs comparable for realistic proxy setups. However, then, the analogue search often fails to find valid analogues as none of our candidate fields comply with our criteria for a valid analogue. That is the method fails to provide a climate reconstruction because of a lack of valid analogues. In the present case, this particularly occurs over the late deglaciation and early Holocene.

Furthermore, our reconstructions by analogue are generally rather imprecise for the used proxies and a limited pool of simulation data. The range of potential analogue values can become very wide for a given date. Regional average reconstruction medians show little variation over time.

The analogue method is non-linear and considers the spatial covariances between the proxy records. While it lacks precision in our setup, it nevertheless provides us with spatial field estimates of past climate states that are consistent with the regional inter-relations as presented by the proxy predictors.

*Data availability.* We provide lists of valid analogues per date and experiment at https://osf.io/pj9eg. This allows identifying valid climate states for dates. We also provide files for area mean analogue ranges and medians.

The proxy data we use is available from the supplement of Marcott et al. (2013) at https://doi.org/10.1126/science.1228026 (see also https://science.sciencemag.org/content/suppl/2013/03/07/339.6124.1198.DC1, both links last accessed December 30, 2019). Primary data citations are Cacho et al. (2006), Grimalt and Calvo (2006), Came et al. (2007b), Castañeda et al. (2010b), Emeis et al. (2003b), Emeis et al. (2000a), Giunta and Emeis (2006), Larocque and Hall (2006), Grimalt and Marchal (2006), Rodrigues et al. (2010), Sarnthein et al. (2003b), Sundqvist et al. (2014a, see also Digerfeldt, 2010, Digerfeldt, 2009, and Sundqvist et al., 2014b), Sundqvist et al. (2014a, see also Voeltzel, 2010a, Voeltzel, 2010b, and Sundqvist et al., 2014b), Thornalley et al. (2009b), and Kim et al. (2004b).

Simulation data is available from a number of sources. Data from simulations for PMIP3 can be obtained from the Earth System Grid Federation, e.g., at the node https://esgf-data.dkrz.de/projects/esgf-dkrz/ (last accessed December 30, 2019). Last Millennium ensemble data and TraCE-21ka output are available at https://www.earthsystemgrid.org/ (last accessed December 30, 2019). Millennium COSMOS simulation data is best accessed via https://cera-www.dkrz.de/WDCC/ui/cerasearch/ (last accessed December 30, 2019). Quaternary QUEST data may be obtained via https://catalogue.ceda.ac.uk/uuid/a43dcfaccfae4824ab9ab2b572703e72 (last accessed December 30, 2019).

**Table A1.** Additional information for the used proxy records: Proxy ID, main reference, and reference for the data sets. For additional information see Table 1.

| Proxy ID | Original Publication | Data References |
|---|---|---|
| MD95-2043 | Cacho et al. (2001) | Cacho et al. (2006) |
| M39-008 | Cacho et al. (2001) | Cacho et al. (2006) |
| MD95-2011 | Calvo et al. (2002) | Grimalt and Calvo (2006) |
| ODP 984 | Came et al. (2007a) | Came et al. (2007b) |
| GeoB 7702-3 | Castañeda et al. (2010a) | Castañeda et al. (2010b) |
| IOW225517 | Emeis et al. (2003a) | Emeis et al. (2003b) |
| IOW225514 | Emeis et al. (2003a) | Emeis et al. (2003b) |
| M25/4-KL11 | Emeis et al. (2000b) | Emeis et al. (2000a) |
| AD91-17 | Giunta et al. (2001) | Giunta and Emeis (2006) |
| Lake 850 | Larocque and Hall (2004) | Larocque and Hall (2006) |
| Lake Nujulla | Larocque and Hall (2004) | Larocque and Hall (2006) |
| MD95-2015 | Marchal et al. (2002) | Grimalt and Marchal (2006) |
| D13882 | Rodrigues et al. (2009) | Rodrigues et al. (2010) |
| GIK23258-2 | Sarnthein et al. (2003a) | Sarnthein et al. (2003b) |
| Flarken Lake | Seppä et al. (2005) | Sundqvist et al. (2014a) |
| Tsuolbmajavri Lake | Seppä and Birks (2001) | Sundqvist et al. (2014a) |
| RAPID-12-1K | Thornalley et al. (2009a) | Thornalley et al. (2009b) |
| GeoB 5901-2 | Kim et al. (2004a) | Kim et al. (2004b) |

## Appendix A:  Additional information on the chosen proxies and the simulation pool

### A1    References for the chosen proxy records

Table A1 provides references to the original publications for the individual proxy records. The table further adds references to the datasets directly and thereby the repositories where the records are available.

### A2    Additional information on the simulation pool

Table A2 provides references for the various models from which we include simulations in the candidate pool. The table further gives links to the repositories where interested researchers can obtain the simulation data.

Tables A3 to A5 complement tables 2 and A2. They give the simulation IDs that allow finding the simulations more easily in the repositories.

**Table A2.** Additional information about the pool of simulation data: Model name, main reference, and link to the provider of the data. For additional information see Table 2.

| Model | References | Link |
| --- | --- | --- |
| CNRM-CM5 | Voldoire et al. (2013) | https://esgf-data.dkrz.de/ |
| COSMOS-ASO | Budich et al. (2010) | https://esgf-data.dkrz.de/ |
| CSIRO-Mk3L-1-2 | Phipps et al. (2011) | https://esgf-data.dkrz.de/ |
| GISS-E2-R | Schmidt et al. (2014b) | https://esgf-data.dkrz.de/ |
| HadCM3 | Collins et al. (2001) | https://esgf-data.dkrz.de/ |
| HadGEM2-CC | Jones et al. (2011) | https://esgf-data.dkrz.de/ |
| HadGEM2-ES | Jones et al. (2011) | https://esgf-data.dkrz.de/ |
| IPSL-CM5A-LR | Dufresne et al. (2013) | https://esgf-data.dkrz.de/ |
| MPI-ESM-P | Giorgetta et al. (2013) | https://esgf-data.dkrz.de/ |
| CESM1 | Otto-Bliesner et al. (2015) | https://www.earthsystemgrid.org/dataset/ucar.cgd.ccsm4.CESM_CAM5_LME.html |
| CCSM3 | Liu et al. (2009) | https://www.earthsystemgrid.org/project/trace.html |
| MPI-ESM-Cosmos | Jungclaus et al. (2010) | https://cera-www.dkrz.de/WDCC/ui/cerasearch/project?acronym=MILLENNIUM_COSMOS |
| FAMOUS-HadCM3 | Smith and Gregory (2012) | https://catalogue.ceda.ac.uk/uuid/a43dcfaccfae4824ab9ab2b572703e72 |

**Table A3.** Information on indidividual simulations: model, simulation, and period.

| Model | Simulation ID | Period |
|---|---|---|
| CNRM-CM5 | lgm_r1i1p1 | Last Glacial Maximum |
| CNRM-CM5 | midHolocene_r1i1p1 | Mid Holocene |
| COSMOS-ASO | lgm_r1i1p1 | Last Glacial Maximum |
| CSIRO-Mk3L-1-2 | midHolocene_r1i1p1 | Mid Holocene |
| GISS-E2-R | lgm_r1i1p150 | Last Glacial Maximum |
| GISS-E2-R | lgm_r1i1p151 | Last Glacial Maximum |
| GISS-E2-R | midHolocene_r1i1p1 | Mid Holocene |
| GISS-E2-R | past1000_r1i1p121 | Last Millennium |
| GISS-E2-R | past1000_r1i1p122 | Last Millennium |
| GISS-E2-R | past1000_r1i1p1221 | Last Millennium |
| GISS-E2-R | past1000_r1i1p123 | Last Millennium |
| GISS-E2-R | past1000_r1i1p124 | Last Millennium |
| GISS-E2-R | past1000_r1i1p125 | Last Millennium |
| GISS-E2-R | past1000_r1i1p126 | Last Millennium |
| GISS-E2-R | past1000_r1i1p127 | Last Millennium |
| GISS-E2-R | past1000_r1i1p128 | Last Millennium |
| HadCM3 | past1000_r1i1p1 | Last Millennium |
| HadGEM2-CC | midHolocene_r1i1p1 | Mid Holocene |
| HadGEM2-ES | midHolocene_r1i1p1 | Mid Holocene |
| IPSL-CM5A-LR | lgm_r1i1p1 | Last Glacial Maximum |
| IPSL-CM5A-LR | midHolocene_r1i1p1 | Mid Holocene |
| IPSL-CM5A-LR | past1000_r1i1p1 | Last Millennium |
| MPI-ESM-P | lgm_r1i1p1 | Last Glacial Maximum |
| MPI-ESM-P | lgm_r1i1p2 | Last Glacial Maximum |
| MPI-ESM-P | midHolocene_r1i1p1 | Mid Holocene |
| MPI-ESM-P | midHolocene_r1i1p2 | Mid Holocene |
| MPI-ESM-P | past1000_r1i1p1 | Last Millennium |

**Table A4.** List of simulations continued

| Model | Simulation ID | Period |
|-------|---------------|--------|
| CESM1 | 0850cntl.001.cam.h0 | pre-industrial control |
| CESM1 | 001.cam.h0 | Last Millennium |
| CESM1 | 002.cam.h0 | Last Millennium |
| CESM1 | 003.cam.h0 | Last Millennium |
| CESM1 | 004.cam.h0 | Last Millennium |
| CESM1 | 005.cam.h0 | Last Millennium |
| CESM1 | 006.cam.h0 | Last Millennium |
| CESM1 | 007.cam.h0 | Last Millennium |
| CESM1 | 008.cam.h0 | Last Millennium |
| CESM1 | 009.cam.h0 | Last Millennium |
| CESM1 | 010.cam.h0 | Last Millennium |
| CESM1 | 011.cam.h0 | Last Millennium |
| CESM1 | 012.cam.h0 | Last Millennium |
| CESM1 | 013.cam.h0 | Last Millennium |
| CESM1 | 850forcing.003.cam.h0 | Last Millennium |
| CESM1 | GHG.001.cam.h0 | Last Millennium |
| CESM1 | GHG.002.cam.h0 | Last Millennium |
| CESM1 | GHG.003.cam.h0 | Last Millennium |
| CESM1 | LULC_HurttPongratz.001.cam.h0 | Last Millennium |
| CESM1 | LULC_HurttPongratz.002.cam.h0 | Last Millennium |
| CESM1 | LULC_HurttPongratz.003.cam.h0 | Last Millennium |
| CESM1 | ORBITAL.001.cam.h0 | Last Millennium |
| CESM1 | ORBITAL.002.cam.h0 | Last Millennium |
| CESM1 | ORBITAL.003.cam.h0 | Last Millennium |
| CESM1 | OZONE_AER.001.cam.h0 | 1850CE-2005CE |
| CESM1 | SSI_VSK_L.001.cam.h0 | Last Millennium |
| CESM1 | SSI_VSK_L.003.cam.h0 | Last Millennium |
| CESM1 | SSI_VSK_L.004.cam.h0 | Last Millennium |
| CESM1 | SSI_VSK_L.005.cam.h0 | Last Millennium |
| CESM1 | VOLC_GRA.001.cam.h0 | Last Millennium |
| CESM1 | VOLC_GRA.002.cam.h0 | Last Millennium |
| CESM1 | VOLC_GRA.003.cam.h0 | Last Millennium |
| CESM1 | VOLC_GRA.004.cam.h0 | Last Millennium |
| CESM1 | VOLC_GRA.005.cam.h0 | Last Millennium |

**Table A5.** List of simulations continued.

| Model | Simulation ID | Period |
| --- | --- | --- |
| CCSM3 | trace | LGM to present |
| MPI-ESM-Cosmos | mil0001 | pre-industrial control |
| MPI-ESM-Cosmos | mil0006 | Last Millennium up 2005CE |
| MPI-ESM-Cosmos | mil0021 | Last Millennium to 2100CE |
| MPI-ESM-Cosmos | mil0025 | Last Millennium to 2100CE |
| MPI-ESM-Cosmos | mil0026 | Last Millennium to 2100CE |
| FAMOUS-HadCM3 (accelerated) | ALL-5G | Last Glacial Cycle |
| FAMOUS-HadCM3 (accelerated) | GHG | Last Glacial Cycle |
| FAMOUS-HadCM3 (accelerated) | ORB | Last Glacial Cycle |
| FAMOUS-HadCM3 (accelerated) | ALL-ZH | Last Glacial Cycle |
| FAMOUS-HadCM3 (accelerated) | ICE | Last Glacial Cycle |

*Author contributions.* OB designed and conducted the study and was the main author. Both authors discussed the methods, the results, and their implications.

*Competing interests.* The authors declare that they have no conflict of interest.

*Acknowledgements.* Funding for this research is by the German Federal Ministry of Education and Research (BMBF) within the Research for Sustainability initiative (FONA; www.fona.de) through the first and second phases of the PalMod project (FKZ: 01LP1509A, FKZ: 01LP1926B). Discussions with Marlene Klockmann and Sebastian Wagner helped to improve the manuscript. We thank two reviewers and the editor for their valuable comments.

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
