# Peer review of "Technical Note: Considerations on using uncertain proxies in the analogue method for spatiotemporal reconstructions of millennial-scale climate"

_Climate of the Past, 2019_

## Referee Comment (RC1) · Anonymous Referee #1 · 13 Feb 2020

I think the paper addresses an important topic and provides a useful extension of the Analogue Method, combining the data and dates uncertainties. However, I think the paper lacks clarity in the description of the method (some fundamental steps as the generation of the confidence ellipses are not properly explained) and I have concerns about the pseudo-proxy setup.

In particular, I think the assessment of the method' skill (of course possible under pseudo-proxy conditions) is flawed: the same run used as "truth" is used inside the Analogue Pool leading, therefore, to a potential overestimation of the skill. In addition to that, I can not comprehend why the authors selected a pseudo-proxy network design

(number of proxies, locations of proxies, period covered, uncertainties, etc.) that do not resemble at all the real-world case they later try to reconstruct. I recommend the authors to re-do the exercise generating a pseudo-proxy environment as close to the real-case as possible. Of course, later the here presented pseudo-proxy setup could be considered informative as how would the method perform if more proxies were considered, etc. but the generation of a closer to real situation is nonetheless essential and I suggest for it not to be bypassed.

General Comments

- The description of the method is not clear enough. How do you define the uncertainty ellipses? I don't see anywhere in the text the methodology followed to find such ellipses? Also, related to that, what's the difference between 90%, 99% or 99.99% uncertainty ellipses? Please, provide a clear methodology to follow to find them. What are the confidence intervals? Please, define.

- Why in the pseudoproxy setup you don't mimic the real-world conditions your are trying to reconstruct? I find it confusing that the pseudoproxy and real-world proxy locations and time-spans are not the same. As it is now, because the pseudoproxy and proxy cases differ, the results of the pseudoproxy analysis are not completely transferable to the real-world case. I suggest to generate a pseudoproxy network that is exactly the same as in the real-world case, and show the results in that case.

- The title of the manuscript talks about climate reconstructions. However, the manuscript deals only with surface temperature reconstructions. I suggest to modify the manuscript title to reflect this and to add some discussion on how the method could/could not be applied to reconstruct some other climate variables (particularly, how do you expect the results to change when reconstructing a more challenging variable as precipitation?).

- It would be interesting to compare with the results of having a fixed number of Analogues, for example 1 Analogue.

- How do results change if using a less years for the sliding window-mean? For example 50-years-means? There is a mention to interannual data in the discussion session, but not comparison plots are shown.

- For the pseudo-proxy setting: The selected reality is the simulation Trace21k. Most of the Analogues come also for this simulation. It would be fairer for assessing the method's skill if the chosen reality is excluded form the Analogues pool. How do the results change if done so? When allowing the same simulation as reality to enter the pool, results might be overly optimistic. For some of the simulations listed there are several runs available, in that case one run could be selected as reality and the other pool of Analogues.

Specific Comments

Abstract: - In the first paragraph authors talk about the last 21 kyr. However, in the second paragraph the target is reduced to the last 15kyr. Please, rephrase or explain failure in the target. - The authors could emphasize that in the present for the reconstruction method seems to be no better than a long-term mean. - These fields reveal that uncertainty are also large locally. Please, change for . . . uncertainty is also. . .

Introduction: - Please clarify the definition of nonillion

Section 2: - Here you sometimes use the word Analog instead of Analogue. Please, unify throughout the manuscript - Figure 1: Please, add latitude and longitude. Also introduce the acronyms P01 and E01, as so far they have not been introduced in the text. - Page 4: 'Our interest is in temperature', please clarify if it is surface, annual mean, etc. What is a temperature calibration? -Page 5: Please explain better the meaning of "at best centennial" Does this mean that there are no proxy records with resolution finer that 100 years? - Page 6: why not consider the same period for real and pseudo proxy setups?, how are ellipses of confidence constructed? Please, provide the appropriate ellipse equation for its construction. - Page 8: What is a credible interval? Please, define. - Page 10: The authors say: "randomly chosen pseudo age uncertainties". How

are those selected? Is the random process a Gaussian distribution? Which mean and variance? This needs more clarification. Figure 3: Isn't it easier to show the plots in the form of line-plots? Specially plot a is difficult to read, as it looks like a huge black block, differences are hard to distinguish. Figure 4: Please, put all the plots in the same scale

Section 3: - Page 14: The authors indicate very little variability in the reconstruction median over certain periods. This probably arises due to too many Analogues are selected in those periods. How could you constrain the Analogue selection?

Figure 5: Panel c: please add name like "warmer case", "colder case" and the respective locations (lon, lat). Panel d: add the subtitle "Regional average" Panel e: add the subtitle "Grid point: (lon, lat)" In panels d and e: I can't understand what the authors mean by "examples". Why some of the examples look like dots and some as dashed lines? Are the dots (dashes) associated to the warmer (colder) case shown in panel c?

It would be interesting to discuss the moments when the Target is outside the envelope (Figure 5a)

Figure 6: Please, add the units directly above the colorbar. Also, indicate the year that is being shown as Example.

Figure 7: Please put all the plots in the same scale.

Figure 8: Similar considerations as in Figure 5.

Figure 9: similar considerations as in Figure 6.

Page 24:

In the summary the authors say that the method succeeds in the pseudo-proxy setup. I think that sentence might be overestimating the skill of the method, as the authors used one model run (Trace21k) both as truth and as proxy pool. Please, remove the truth from the possible pool of Analogues to be able to properly analyse the method's skill.

For the real-case the authors say the reconstructions fail. How can you assure failure when you don't know the truth? I think the sentence should be re-phrased and the only thing that can be known for sure is the failure to find Analogues within the selected pool. I think that it needs to be made clearer that not knowing the truth in the real-case is exactly the reason for making pseudo-proxy analysis. Which leads, again, for the importance of the pseudo-proxy setup (design of the network, period covered, etc.) to be as similar as possible to the real-case.

―――――――――――――――――――

---

## Referee Comment (RC2) · Anonymous Referee #2 · 3 Mar 2020

General Comments:

The paper discusses an analogue method of paleoclimate reconstruction. In this method, the researcher starts with a set of paleoclimate records (here, temperature-sensitive records in or near Europe) and searches for similar climate states within a pool of climate simulation outputs. By finding modeled states with match the proxy records, this method can be used to estimate the state of the climate system at locations which do not have local data. This method has been used in previous research, so the main focus of this paper is on the treatment of temporal and magnitude uncertainty of the proxy records.

In general, the goal of the paper–to better account for uncertainty in a computationally cheap reconstruction method–is worthwhile, so the case study presented in this paper is welcome. However, the method doesn't seem to work very well, which seems to be a major shortcoming. While, in theory, this may be acceptable as a stepping stone to further research, I also have additional concerns about the design and presentation of the research. In particular: 1) descriptions of the paper's methodology are sometimes confusing, and would benefit from further refinement; 2) I have several concerns about the paper's methodology, which seem like they limit the success of finding analogues; a revised methodology may result in a more successful reconstruction and a more interesting paper; and 3) the figures could be improved. These points are expanded upon in the "Specific comments" section below. I feel like these are important points which should be addressed.

Specific comments:

1. In a method-heavy paper, extra care must be taken to ensure that the paper is intuitive. When reading the paper, however, I had a variety of questions about how the method worked and what factors were keeping it from working better. Several of these confusions are listed below:

- The discussion of ellipses, which represent uncertainty in time and magnitude, is somewhat confusing at first, and it took me some time to understand they were used within the methodology.

- The relevance of the 90% vs. 99% vs. 99.99% cutoffs is not clearly explained. It appears that they refer to percentiles of magnitude and time uncertainty, but how are they calculated?

- Some aspects of panels d and e in Figs. 5 and 8 are unclear. As far as I understand, these panels are showing the annual data underlying the selected 101-year means, but I'm not sure what I should take away from them. Can their purpose be better explained, or can they be revised to show the relevant points in a more intuitive manner? In

particular, I don't understand the lines marked as "examples". Also, it may help if the "examples" were solid lines rather than dotted/dashed.

In general, I would encourage the authors to read through the manuscript again with a focus on making explanations clearer and more intuitive.

2. I am concerned about several aspects of the methodology, which seem like they may prevent the method from finding good analogues. My main two concerns are described below, with the second point being the more important of the two. Unless I am misunderstanding something (see point #1 above), I would like to see these concerns discussed or, preferably, directly accounted for within the methodology.

2.1. Uncertainty Ellipse Edge-Effects:

The use of uncertainty ellipses, which have a hard cutoff, may prevent the method from finding good analogues. One example of this may be imagined at the left and right "edges" of the ellipses. At the left and right edges of the ellipse, the vertical extent of an ellipse (representing magnitude uncertainty) becomes very small, eventually reaching 0. If the method is looking for analogues near the edge of one of these ellipses, the range of an "acceptable" analogue would be very narrow, rejecting many potential candidates.

Let's take the scenario in section 2.2.4 as an example. The paper states that there is a hypothetical proxy value at 500 BP, with age uncertainties from 600 to 400 BP. This hypothetical uncertainty ellipse stretches between 600 and 400 BP, with its maximum magnitude uncertainty at 500 BP. If an analogue search is conducted at 500 BP, the method accepts all points within the full uncertainty range of the ellipse. However, if an analogue search is conducted at 401 BP, the uncertainty range of the ellipse (i.e. the height of the ellipse, similar to the ones visualized in Fig. 2b) would be much smaller, therefore rejecting many potential analogues. This seems counter-intuitive to me. Wouldn't it make more sense to broaden the magnitude uncertainty as you get farther from the central age date, since we are less sure that the data point is applicable

as we get farther from its original dated age?

This issue may only be a problem at the start or end of a proxy record, or near a very long gap, but I expect that it would become more and more of a problem as the method is applied to more proxies, which naturally have different start and end dates. Unless I'm misunderstanding the method, I think that a better handling of these "edge effects" would help the method find more valid matches. Perhaps rectangles could be used instead of ellipses, since I see no reason that magnitude uncertainty should be decreased near the edge of temporal uncertainties. If anything, I would expect a particular point to become less precise toward the edges, not more precise. Since altering the method to address this would likely be too much work, I think that this point should be at least be mentioned in the paper.

2.2. Potential for Outliers to Cause Method Failure:

The paper mentions that the method uses the absolute temperatures calibrated from proxies, rather than anomalies. The authors discuss the problems surrounding the choice of absolute values vs. anomalies, but I'm concerned that biases in the absolute value of a single record (or simply non-climate proxy variations) could cause the method to fail. Consider applying this methodology to a group of proxies where a single proxy has been accidentally calibrated to be too warm by 5 degrees C. An error like this could hypothetically cause every single potential analogue to fail for the entire length of the proxy, as it's possible that no modeled state would show a spike of temperature at that particular location compared to everywhere else in the region. This means that the method would fail even if every other proxy were a perfect recorder of climate.

If a single problematic proxy can cause the whole method to fail, this problem will only become more likely to occur as the method is applied to a larger and larger proxy database. As it is, the method has trouble finding analogues with even a small set of proxies (as little as 7 proxies for the E09 case). This seems like a fundamental problem with the method, limiting its future application. The authors try to widen the group of

successful analogues by using wider uncertainty bands, including/removing records, and using annual model states rather than 101-year means, but I don't think that any of these solutions fix the underlying problem, which I suspect is the use of a binary match/mismatch dichotomy with the uncertainty ellipses. Using strict match/mismatch criteria probably makes the method overly sensitive to mismatches with single proxies. The use of a skill metric, as used in other work, may help alleviate issues arising from a subset of problematic records. Alternately, perhaps analogues could be accepted even if a certain percentage of the proxies don't match, to account for biases and non-climate noise within the proxy data set.

To the authors' credit, much of the paper does discuss potential problems with the method, and also suggests ways that things could be improved in the future. Indeed, the paper appears to be an exploration of how to account for age/magnitude uncertainties, rather than the presentation of a finished methodology. However, the paper would be much more satisfying to read if some of these issues were implemented directly, hopefully leading to a more complete reconstruction than the one shown in Fig. 8.

If this is not possible, I would at the very least like to see the following: 1) More discussion of the methodological problems mentioned above. 2) A different title, which accurately reflects the fact that the paper's methodology is a work-in-progress rather than a finished method. As-is, the title makes it sound like this paper demonstrates a finished methodology, when it appear to be an exploration of uncertainties which may lead to a better method in the future. Because of this, a better title might be something like: "Considerations of proxy uncertainties within the analogue method of paleoclimate reconstruction".

3. In general, several of the figures could be improved. For example, the black and red colors in Fig. 3 are difficult to distinguish, and the lines in Figs. 5c and 8c are difficult to interpret, since they use similar thicknesses and opacities. Improving the figures may also help make the methodology more intuitive, as I commented about in point #1 above.

A few other minor questions/concerns: Why only use 101-year means, rather than means which vary site-by-site to better reflect the temporal characteristics of individual proxy records? Also, why does the pseudoproxy experiment only use summer means, as mentioned in line 30 on page 10? And why does the number of sites differ between the pseudoproxy experiment (Fig. 1a) and the real experiment (Fig. 1b)? I had other questions about methodological choices while reading the paper, but the major points discussed throughout the review above seemed like the most important.

A final technical note: some figures (especially Fig. 4) have so many lines that the paper is difficult to print (it gets stuck on a "flattening" step for a long time).

In summary, while the paper focuses on an interesting and useful approach to paleo-climate reconstruction, I think that several things need to be improved before it can be considered for publication. A fundamental problem is that this appears to be a method paper, but the method doesn't work very well. If the method cannot be improved, the concerns above should at least be addressed and the paper should get a new title which better reflects its contents.

Finally, despite all of my comments and concerns, I do think that this is an interesting and potentially useful method, and I hope that further progress is made in the future.

---

## Author Comment (AC1) · 2 Apr 2020

Dear editor, dear referees,

Hereby we want to thank the editor and the referees for their evaluation of our manuscript and their helpful comments. Below we provide a response to their remarks.

We note that a couple of our replies ask for guidance by the editor.

On behalf of the authors

Yours sincerely

Oliver Bothe

Referee 1

I think the paper addresses an important topic and provides a useful extension of the Analogue Method, combining the data and dates uncertainties. However, I think the paper lacks clarity in the description of the method (some fundamental steps as the generation of the confidence ellipses are not properly explained) and I have concerns about the pseudo-proxy setup.

*Response: We thank the referee for their generous evaluation of our manuscript. Below we address these points in more detail. In our revisions, we will particularly take care to clarify our method. Below we also address the concerns about the pseudo-proxy setup.*

In particular, I think the assessment of the method' skill (of course possible under pseudo-proxy conditions) is flawed: the same run used as "truth" is used inside the Analogue Pool leading, therefore, to a potential overestimation of the skill. In addition to that, I can not comprehend why the authors selected a pseudo-proxy network design (number of proxies, locations of proxies, period covered, uncertainties, etc.) that do not resemble at all the real-world case they later try to reconstruct. I recommend the authors to re-do the exercise generating a pseudo-proxy environment as close to the real-case as possible. Of course, later the here presented pseudo-proxy setup could be considered informative as how would the method perform if more proxies were considered, etc. but the generation of a closer to real situation is nonetheless essential and I suggest for it not to be bypassed.

*Response: There are two points to address here:*

*1. We will do/redo one experiment with a pseudoproxy setup as close to the real-world cases as possible.*

*2. We understand the concern of the referee, and agree that the setup is suboptimal. We are confident that the submitted manuscript already was careful not to overestimate*

*the skill of the method. However, we are going to be even more clear in stating how good the method may be.*

*Here, we want to point out why we use the setup as criticised by the referee: For one, at the time of our study, the Trace-21ka simulation was the only available simulation providing a continuous deglacial climate trajectory in annual resolution. Tests showed that in our chosen setup the method does only find analogues from the Trace-21ka simulation from which we also constructed the pseudo-proxies, as we wrote in the manuscript.*

*Thus, if we exclude Trace-21ka from the candidate pool the method fails completely, and if we don't use Trace-21ka for the pseudoproxy construction, we cannot use a simulation in interannual resolution. We will discuss this in our revisions. For this reason, for the time being, and unless the editor advises us to change this, we do not plan to change our setup. We allow that we could have used the QUEST FAMOUS simulations to provide pseudo-proxies in, presumably, centennial resolution.*

General Comments

- The description of the method is not clear enough. How do you define the uncertainty ellipses? I don't see anywhere in the text the methodology followed to find such ellipses. Also, related to that, what's the difference between 90%, 99% or 99.99% uncertainty ellipses? Please, provide a clear methodology to follow to find them. What are the confidence intervals? Please, define.

*Response: We will clarify this in our revisions, possibly including a figure if the text becomes too cluttered with technical descriptions.*

- Why in the pseudoproxy setup you don't mimic the real-world conditions your are trying to reconstruct? I find it confusing that the pseudoproxy and real-world proxy locations and time-spans are not the same. As it is now, because the pseudoproxy

and proxy cases differ, the results of the pseudoproxy analysis are not completely transferable to the real-world case. I suggest to generate a pseudoproxy network that is exactly the same as in the real-world case, and show the results in that case.

*Response: We will do/redo one pseudo-proxy setup with a setup as close to one of the real-world cases as possible.*

- The title of the manuscript talks about climate reconstructions. However, the manuscript deals only with surface temperature reconstructions. I suggest to modify the manuscript title to reflect this and to add some discussion on how the method could/could not be applied to reconstruct some other climate variables (particularly, how do you expect the results to change when reconstructing a more challenging variable as precipitation?).

*Response: We will consider the recommendations of both reviewers and change the title. We will also discuss the extension and the transferability of the method. However, already here we want to stress, one needs to remain sceptical that additional variables can be reconstructed. This would amount to assuming that precipitation is tightly constrained by temperature. While this may be true on a global scale, at regional scales precipitation is also modulated by shifts in storm tracks and changes in the atmospheric circulation in general. This can be spatially very heterogeneous.*

- It would be interesting to compare with the results of having a fixed number of Analogues, for example 1 Analogue.

*Response: Indeed this would be an interesting experiment. One may, however, ask whether it is really meaningful considering the large uncertainties of the proxies, as one would only test relative to the best estimate for each proxy. We consider to provide such a test, but so far do not see it as essential for this manuscript. Analyses for shorter time scales (compare recent works from Juan José Gómez-Navarro and colleagues in Climate Dynamics and Climate of the Past) have found that fewer analogues lead to higher variability in the reconstructions but also to lower skill. This is plausibly also the*

*case here, as these effects can be explained by statistical sampling. Thus, it would not be a novel contribution in this regard.*

- How do results change if using a less years for the sliding window-mean? For example 50-years-means? There is a mention to interannual data in the discussion session, but not comparison plots are shown.

*Response: We consider to provide a test with shorter or longer sliding-window means. We note that Appendix Figure A5a already provides a comparison to interannual data for experiment E01.*

- For the pseudo-proxy setting: The selected reality is the simulation Trace21k. Most of the Analogues come also for this simulation. It would be fairer for assessing the method's skill if the chosen reality is excluded from the analogues pool. How do the results change if done so? When allowing the same simulation as reality to enter the pool, results might be overly optimistic. For some of the simulations listed there are several runs available, in that case one run could be selected as reality and the other pool of Analogues.

*Response: See our response to a previous point: We understand the concern of the referee, and agree that the setup is suboptimal. We are confident that the submitted manuscript already was careful not to overestimate the skill of the method. However, we are going to be even more clear in stating how good the method may be.*

*Here, we want to point out, why we use the setup as criticised by the referee: For one, at the time of our study, the Trace-21ka simulation was the only available simulation providing a continuous deglacial climate trajectory in annual resolution. Tests showed that in our chosen setup the method does only find analogues from the Trace-21ka simulation from which we also constructed the pseudo-proxies as we write in the manuscript.*

*Thus, if we exclude Trace-21ka from the candidate pool the method fails completely,*

*and if we don't use Trace-21ka for the pseudoproxy construction, we cannot use a simulation in interannual resolution. We will discuss this in our revisions. For the time being, and unless the editor advises us to change this, we are not going to change our setup. We allow that we could have used the QUEST FAMOUS simulations to provide pseudo-proxies in, presumably, centennial resolution.*

Specific Comments

Abstract:

- In the first paragraph authors talk about the last 21 kyr. However, in the second paragraph the target is reduced to the last 15kyr. Please, rephrase or explain failure in the target.

*Response: We will be more clear about the temporal scope of the manuscript. The discrepancy is solely related to the different temporal extent of the pseudoproxy and real-world applications.*

- The authors could emphasize that in the present for the reconstruction method seems to be no better than a long-term mean.

*Response: We are not fully clear to what "in the present" refers here, but we will put this information into a revised manuscript.*

- These fields reveal that uncertainty are also large locally. Please, change for . . . uncertainty is also. . .

*Response: We will correct this.*

Introduction:

- Please clarify the definition of nonillion

*Response: We will do so.*

Section 2:

- Here you sometimes use the word Analog instead of Analogue. Please, unify throughout the manuscript

*Response: We are sorry for this oversight and we will do so.*

- Figure 1: Please, add latitude and longitude. Also introduce the acronyms P01 and E01, as so far they have not been introduced in the text.

*Response: We will do both.*

- Page 4: 'Our interest is in temperature', please clarify if it is surface, annual mean, etc. What is a temperature calibration?

*Response: We will clarify these questions.*

-Page 5: Please explain better the meaning of "at best centennial" Does this mean that there are no proxy records with resolution finer that 100 years?

*Response: We will clarify this.*

- Page 6: why not consider the same period for real and pseudo proxy setups?, how are ellipses of confidence constructed? Please, provide the appropriate ellipse equation for its construction.

*Response: We are able to extend the reconstruction period for the pseudoproxy approach back to the Last Glacial Maximum. We regard this an interesting exercise.*

*Therefore, we will keep the different periods. We acknowledge that there are formal arguments for synchronising the setups.*

*We will add a clearer explanation of how we construct the ellipses.*

- Page 8: What is a credible interval? Please, define.

*Response: We will clarify our terminology.*

- Page 10: The authors say: "randomly chosen pseudo age uncertainties". How are those selected? Is the random process a Gaussian distribution? Which mean and variance? This needs more clarification.

*Response: We will clarify this briefly.*

Figure 3: Isn't it easier to show the plots in the form of line-plots? Specially plot a is difficult to read, as it looks like a huge black block, differences are hard to distinguish.

*Response: We think the vertical lines better represent the discrete character of the approach but we will reconsider how to visualize the data.*

Figure 4: Please, put all the plots in the same scale

*Response: The reviewer's suggestion would make it close to impossible to identify changes in individual series. However, we will synchronise the absolute range of the temperature-axes for all panels.*

Section 3:

- Page 14: The authors indicate very little variability in the reconstruction median over certain periods. This probably arises due to too many Analogues are selected in those periods. How could you constrain the Analogue selection?

*Response: There is a trade-off between considering the uncertainty of the proxies*

*and constraining the number of analogues. We consider providing a discussion of this
aspect.*

Figure 5: Panel c: please add name like "warmer case", "colder case" and the respective locations (lon, lat). Panel d: add the subtitle "Regional average" Panel e: add the subtitle "Grid point: (lon, lat)" In panels d and e: I can't understand what the authors mean by "examples". Why some of the examples look like dots and some as dashed lines? Are the dots (dashes) associated to the warmer (colder) case shown in panel c? It would be interesting to discuss the moments when the Target is outside the envelope (Figure 5a)

*Response: We will try to improve the visualisation of our results, and we will provide
a clearer description of the results. We are not going to discuss in detail the specific
cases when the target is outside the envelope but we will thoroughly discuss reasons
why the target may generally fall outside the envelope.*

Figure 6: Please, add the units directly above the colorbar. Also, indicate the year that is being shown as Example.

*Response: We will clarify the Figure.*

Figure 7: Please put all the plots in the same scale.

*Response: As for Figure 4, the reviewer's suggestion would make it hard to identify
changes in individual series. However, we will synchronise the absolute range of the
temperature-axes for all panels.*

Figure 8: Similar considerations as in Figure 5. Figure 9: similar considerations as in Figure 6.

*Response: We will clarify all four Figures.*

Page 24: In the summary the authors say that the method succeeds in the pseudo-proxy setup. I think that sentence might be overestimating the skill of the method, as

the authors used one model run (Trace21k) both as truth and as proxy pool. Please, remove the truth from the possible pool of Analogues to be able to properly analyse the method's skill.

*Response: We cannot remove the truth from the pool of analogues as we then would not be able to apply the method due to the lack of suitable simulations as we wrote in the manuscript. We will more clearly discuss this.*

For the real-case the authors say the reconstructions fail. How can you assure failure when you don't know the truth? I think the sentence should be re-phrased and the only thing that can be known for sure is the failure to find Analogues within the selected pool. I think that it needs to be made clearer that not knowing the truth in the real-case is exactly the reason for making pseudo-proxy analysis. Which leads, again, for the importance of the pseudo-proxy setup (design of the network, period covered, etc.) to be as similar as possible to the real-case.

*Response: We will rephrase this to make it clear that a failure of the method is equivalent to a failure of finding analogues in the candidate pool.*

---

## Author Response (AR1)

Dear editor, dear referees,

Hereby we want to thank the editor and the referees for their evaluation of our manuscript and their helpful comments. Below we provide first a general list of changes to the manuscript and then our updated responses to their remarks.

We want to apologize that providing these corrections took us so long.

On behalf of the authors

Yours sincerely

Oliver Bothe

**List of changes**

- Title
  - Title changed
- Abstract
  - minor changes in text
- Introduction
  - minor changes
- Analog method, assumptions, and data
  - Adapted Figure 1 according to reviewer comments and to new proxy collection
  - Adapted Figure 1 caption
  - reframing in terms of tolerance ranges
  - clarifying our considerations on uncertainty
  - clarifying our method
  - adding ellipse equation
  - adding comments on additional experiments
  - minor changes in caption of Figure 2
  - moved Figure from appendix to this section as new Figure 3
  - different pseudoproxy setup
  - explained changes to pseudoproxy generation
  - updated former Figure 3 (now Figure 4) to show all experiments
  - clarification on QUEST-FAMOUS data
  - minor further changes
- Results
  - Figures changed to improve them
  - Changes due to use of different pseudoproxy setup in text and figures
  - added results for different proxy setups from appendix and relevant Figure
  - added results for different methodological choices
- Discussion
  - Changes to better reflect shortcomings of our approach
- Summary and concluding remarks
  - Changed for new version
- Appendix
  - Largely moved to main text

**Editor**

Comments to the Author: Dear authors,

Thank you for submitting your reply to the comments of the two reviewers. Both find your manuscript of interest, but suggest major revisions. Please submit a revised version according to the changes you proposed.

*Response: We thank the editor for this evaluation and guidance.*

Regarding your specific questions about 1) estimating skill from non-independent pseudoproxies: I agree with the reviewer that this is an important point and encourage you to include, as you suggested yourselves, an experiment with independent pseudoproxies based on the QUEST-FAMOUS simulations. Perhaps this would also help to include the information that is now in the appendices in the main text?

*Response: We thank the editor for this guidance. We now changed the pseudoproxy test to a setting closer to the real conditions. We do not discuss anymore the other test.*

*We also move most of Appendix A to the main text and remove other parts of Appendix A.*

2) title: I agree with the reviewer that a title that better reflects that this is work in progress would be good. In doing so, climate does not need to be changed to temperature.

*Response: We thank the editor for this guidance. We changed the title.*

The wording used to describe the ellipses is somewhat confusing, as is also clear from the questions posed by the second reviewer. Your explanation "The ellipses do not represent the uncertainty ranges in the value of the proxies, but rather the confidence with which we claim to know the value of the proxy at that time." explains the problem well and I think that besides an improved description of the method and further discussion, a change in the wording would also help to clarify what these ellipses really mean. Perhaps "confidence ellipse" would be a good alternative?

*Response: We aim to be as clear as possible for us in describing the approach.*

*We first reframed everything in terms of confidence ellipses but then thought it might be even clearer to represent the ellipses as tolerance ranges.*

And finally something very minor, in Fig A2 there's an "o" missing in first time chironomids are mentioned.

*Response: Thank you for spotting this. We corrected this.*

With kind regards, Lukas Jonkers

**Referee 1**

I think the paper addresses an important topic and provides a useful extension of the Analogue Method, combining the data and dates uncertainties. However, I think the paper lacks clarity in the description of the method (some fundamental steps as the generation of the confidence ellipses are not properly explained) and I have concerns about the pseudoproxy setup.

*Response: We thank the referee for their generous evaluation of our manuscript. Below we address these points in more detail. In our revisions, we will particularly take care to clarify our method. Below we also address the concerns about the pseudoproxy setup.*

In particular, I think the assessment of the method' skill (of course possible under pseudoproxy conditions) is flawed: the same run used as "truth" is used inside the Analogue Pool leading, therefore, to a potential overestimation of the skill. In addition to that, I can not comprehend why the authors selected a pseudoproxy network design (number of proxies, locations of proxies, period covered, uncertainties, etc.) that do not resemble at all the real-world case they later try to reconstruct. I recommend the authors to re-do the exercise generating a pseudoproxy environment as close to the real-case as possible. Of course, later the here presented pseudoproxy setup could be considered informative as how would the method perform if more proxies were considered, etc. but the generation of a closer to real situation is nonetheless essential and I suggest for it not to be bypassed.

*Response: There are two points to address here:*

*1. We redo our pseudoproxy test with a setup closer to the real-world case. See changes in our section on methods and data.*

*2. We changed the target of our pseudoproxy setup. We try to be careful not to overestimate the skill of the method.*

*We now use QUEST-FAMOUS simulation data for the pseudoproxy setup, which, however, does not have interannual resolution.*

*Here, we want to again point out why we used the setup as criticised by the referee: For one, at the time of our study, the Trace-21ka simulation was the only available simulation providing a continuous deglacial climate trajectory in annual resolution. We assumed that an interannual setup is the most reliable approach. Tests showed that in our chosen setup the method does only find analogues from the Trace-21ka simulation from which we also constructed the pseudoproxies, as we wrote in the manuscript.*

*Thus, if we exclude Trace-21ka from the candidate pool the method fails completely, and excluding the Trace-21ka simulation from the pseudoproxy construction prevents using a simulation with interannual resolution for the construction of pseudoproxies.*

*Following the review-comments we now use one of the QUEST-FAMOUS simulations. The FAMOUS-HadCM3 simulations for QUEST use accelerated forcings. That is, the last glacial cycle of approximately 120,000 years of climate forcing was simulated in approximately 12,000 simulation years. Thus, the annual simulation data is only representative of ten years of climate evolution.*

**General Comments**

- The description of the method is not clear enough. How do you define the uncertainty ellipses? I don't see anywhere in the text the methodology followed to find such ellipses. Also, related to that, what's the difference between 90%, 99% or 99.99% uncertainty ellipses? Please, provide a clear methodology to follow to find them. What are the confidence intervals? Please, define.

*Response: We rewrote the methods section as well as other parts of the manuscript to clarify these points.*

- Why in the pseudoproxy setup you don't mimic the real-world conditions your are trying to reconstruct? I find it confusing that the pseudoproxy and real-world proxy locations and time-spans are not the same. As it is now, because the pseudoproxy and proxy cases differ, the results of the pseudoproxy analysis are not completely transferable to the real-world case. I suggest to generate a pseudoproxy network that is exactly the same as in the real-world case, and show the results in that case.

*Response: We redo the pseudoproxy setup with a setup closer to the real-world case.*

- The title of the manuscript talks about climate reconstructions. However, the manuscript deals only with surface temperature reconstructions. I suggest to modify the manuscript title to reflect this and to add some discussion on how the method could/could not be applied to reconstruct some other climate variables (particularly, how do you expect the results to change when reconstructing a more challenging variable as precipitation?).

*Response: We modified the title.*

*We shortly note in the discussions that reconstruction success for other variables is less likely than for temperature.*

- It would be interesting to compare with the results of having a fixed number of Analogues, for example 1 Analogue.

*Response: Indeed this would be an interesting experiment. One may, however, ask whether it is really meaningful considering the large uncertainties of the proxies, as one would only test relative to the best estimate for each proxy.*

*We shortly discuss such a test in our results section now.*

*We want to repeat that analyses for shorter time scales (compare recent works from Gómez-Navarro and colleagues in Climate Dynamics and Climate of the Past) have found that fewer analogues lead to higher variability in the reconstructions but also to lower skill. This is plausibly also the case here, as these effects can be explained by statistical sampling.*

- How do results change if using a less years for the sliding window-mean? For example 50-years-means? There is a mention to interannual data in the discussion session, but not comparison plots are shown.

*Response: We now shortly discuss test-cases with interannual data, 51-year averages and 501-year averages.*

- For the pseudoproxy setting: The selected reality is the simulation Trace21k. Most of the Analogues come also for this simulation. It would be fairer for assessing the method's skill if the chosen reality is excluded from the analogues pool. How do the results change if done so? When allowing the same simulation as reality to enter the pool, results might be overly optimistic. For some of the simulations listed there are several runs available, in that case one run could be selected as reality and the other pool of Analogues.

*Response: See our response to a previous point: We understand the concern of the referee, and agree that the setup is suboptimal. We reconsidered our writing and hope the new version is sufficiently careful not to overestimate the skill of the method.*

*We now use a different setup. We try to be as clear as possible about the potential quality of the method.*

*Here, we want to point out why we used the setup as criticised by the referee: For one, at the time of our study, the Trace-21ka simulation was the only available simulation providing a continuous deglacial climate trajectory in annual resolution. Tests showed that in our chosen setup the method does only find analogues from the Trace-21ka simulation from which we also constructed the pseudoproxies as we write in the manuscript.*

*Thus, if we exclude Trace-21ka from the candidate pool the method fails completely, and if we don't use Trace-21ka for the pseudoproxy construction, we cannot use a simulation with interannual resolution.*

*We now use the QUEST FAMOUS simulations to provide pseudoproxies.*

*Regarding the suggestion of a single model ensemble: we want to emphasize that there is not really a suitable single model ensemble of simulations over the periods of interest available to test the analogue method on the timescales of interest.*

**Specific Comments**

\* Abstract:

- In the first paragraph authors talk about the last 21 kyr. However, in the second paragraph the target is reduced to the last 15kyr. Please, rephrase or explain failure in the target.

*Response: We will be more clear about the temporal scope of the manuscript. The discrepancy was solely related to the different temporal extent of the pseudoproxy and real-world applications. To be specific, the period of the pseudoproxy setup was limited by the length of the available simulation while the period of the real world setup was limited by the length of the proxy records.*

*Our changed pseudoproxy setup effectively leads to more comparable periods.*

- The authors could emphasize that in the present for the reconstruction method seems to be no better than a long-term mean.

*Response: We are not fully clear to what "in the present" refers here. We now note "that the analogue method in the present setting may represent the recent climate worse than simply taking the average over the period of instrumental observations."*

- These fields reveal that uncertainty are also large locally. Please, change for . . . uncertainty is also. . .

*Response: We change this to "uncertainties are also".*

\* Introduction:

- Please clarify the definition of nonillion

*Response: We do clarify in our revisions that nonillion refers to $10^{30}$.*

\* Section 2:

- Here you sometimes use the word Analog instead of Analogue. Please, unify throughout the manuscript

*Response: We are sorry for this oversight. We changed instances of "analog" to "analogue".*

- Figure 1: Please, add latitude and longitude. Also introduce the acronyms P01 and E01, as so far they have not been introduced in the text.

*Response: We added these. Figure 1 now includes axes for longitude and latitude. The caption now introduces P01 and E01.*

- Page 4: 'Our interest is in temperature', please clarify if it is surface, annual mean, etc. What is a temperature calibration?

*Response: We clarify this now.*

-Page 5: Please explain better the meaning of "at best centennial" Does this mean that there are no proxy records with resolution finer that 100 years?

*Response: We clarify this.*

- Page 6: why not consider the same period for real and pseudo proxy setups?, how are ellipses of confidence constructed? Please, provide the appropriate ellipse equation for its construction.

*Response: Considering our previous pseudoproxy setup we were able to extend the reconstruction period for the pseudoproxy approach back to the Last Glacial Maximum. We regarded this an interesting exercise. Considering the new pseudoproxy setup we also are able to extend it but concentrate our discussions on the shorter period.*

*We clarified the explanation of how we construct the ellipses and added the equation.*

- Page 8: What is a credible interval? Please, define.

*Response: We clarified our terminology. We now reframe the uncertainties in terms of tolerance ranges to accept analogues.*

- Page 10: The authors say: "randomly chosen pseudo age uncertainties". How are those selected? Is the random process a Gaussian distribution? Which mean and variance? This needs more clarification.

*Response: Bothe et al. (2019) include a switch in their script for pseudoproxy calculation. One may use a Gaussian distribution and the parameters of this distribution. A second option is to use an uncertainty dependent on the assumed random smoothing of the pseudoproxy. A third option calculates the uncertainty assuming a constant smoothing of the pseudoproxy record length and a random Gaussian offset. We use this setup but scale it down to reduce the width of the uncertainties.*

Figure 3: Isn't it easier to show the plots in the form of line-plots? Specially plot a is difficult to read, as it looks like a huge black block, differences are hard to distinguish.

*Response: We think the vertical lines better represent the discrete character of the approach but we changed the visualisation to line-steps.*

Figure 4: Please, put all the plots in the same scale

*Response: The reviewer's suggestion would make it harder to identify changes in individual series. We now use a common absolute range of the temperature-axes for all panels.*

**\*** Section 3:

- Page 14: The authors indicate very little variability in the reconstruction median over certain periods. This probably arises due to too many Analogues are selected in those periods. How could you constrain the Analogue selection?

*Response: There is a trade-off between considering the uncertainty of the proxies and constraining the number of analogues. That is, if we want to consider the uncertainty in the way we do, then we allow for weakly constrained analogue ranges. If we allow different levels of proxy uncertainty, we can choose only the best $M$ analogues. We then can limit the number of analogues by another criterion based on their distance to individual proxies or their overall Euclidean distance.*

*Indeed, a likely explanation for the little variability in central estimates and the generally rather constant character of our reconstructions could be that the space of valid analogues is too unconstrained and too many analogues are considered valid. However, also the single-best analogue approach shows such a behaviour. That is, while the reconstruction is undoubtedly badly constrained, even the best analogues differ little between subsequent dates. Part of this may be due to our choice to consider a rather large temporal range of influence of individual dated records. Our ellipses of tolerance may result in a strong influence of an unlikely value at a specific date. This could potentially be solved by explicitly considering the likelihood of a value at a particular date instead of simply taking a binary criterion. A less complex solution could be obtained by pooling proxy values in temporal windows, weighting them*

within these windows, and then performing a reconstruction considering certain ranges of tolerance to accept analogues.

*We add this discussion to the manuscript.*

Figure 5: Panel c: please add name like "warmer case", "colder case" and the respective locations (lon, lat). Panel d: add the subtitle "Regional average" Panel e: add the subtitle "Grid point: (lon, lat)" In panels d and e: I can't understand what the authors mean by "examples". Why some of the examples look like dots and some as dashed lines? Are the dots (dashes) associated to the warmer (colder) case shown in panel c? It would be interesting to discuss the moments when the Target is outside the envelope (Figure 5a)

*Response: We will try to improve the visualisation of our results, and we will provide a clearer description of the results. We tried to follow the reviewer suggestions on Figure 5. As the pseudoproxy setup changed the discussion of cases outside the envelope would have to change as well. That is, even in the perfect model setup of the preprint, the simulation data and the pseudoproxy differed. Then we could not expect the analogue reconstruction to always include the original target.*

Figure 6: Please, add the units directly above the colorbar. Also, indicate the year that is being shown as Example.

*Response: We added the units and mention the year in the Figure caption.*

Figure 7: Please put all the plots in the same scale.

*Response: As for Figure 4, the reviewer's suggestion would make it hard to identify changes in individual series. Now, we use a common absolute range of the temperature-axes for all panels.*

Figure 8: Similar considerations as in Figure 5. Figure 9: similar considerations as in Figure 6.

*Response: We clarified all four Figures. We adapted Figure 9 following the suggestions on Figure 6. Note, as we added further Figures, the Figure-numbering changed.*

Page 24: In the summary the authors say that the method succeeds in the pseudoproxy setup. I think that sentence might be overestimating the skill of the method, as the authors used one model run (Trace21k) both as truth and as proxy pool. Please, remove the truth from the possible pool of Analogues to be able to properly analyse the method's skill.

*Response: While we would prefer using an interannual input for the pseudoproxies we changed the setup so that the truth is not any longer in the pool of analogues. We are confident that our previous statement was careful enough and think the current manuscript is clearer.*

For the real-case the authors say the reconstructions fail. How can you assure failure when you don't know the truth? I think the sentence should be re-phrased and the only thing that can be known for sure is the failure to find Analogues within the selected pool. I think that it needs to be made clearer that not knowing the truth in the real-case is exactly the reason for making pseudoproxy analysis. Which leads, again, for the importance of the pseudoproxy setup (design of the network, period covered, etc.) to be as similar as possible to the real-case.

*Response: We rephrase this to highlight that a failure of the method is equivalent to a failure of finding analogues in the candidate pool.*

**Referee 2**

**General Comments:**

The paper discusses an analogue method of paleoclimate reconstruction. In this method, the researcher starts with a set of paleoclimate records (here, temperatur-esensitive records in or near Europe) and searches for similar climate states within a pool of climate simulation outputs. By finding modeled states with match the proxy records, this method can be used to estimate the state of the climate system at locations which do not have local data. This method has been used in previous research, so the main focus of this paper is on the treatment of temporal and magnitude uncertainty of the proxy records.

In general, the goal of the paper–to better account for uncertainty in a computationally cheap reconstruction method–is worthwhile, so the case study presented in this paper is welcome. However, the method doesn't seem to work very well, which seems to be a major shortcoming. While, in theory, this may be acceptable as a stepping stone to further research, I also have additional concerns about the design and presentation of the research. In particular:

*Response: We thank the referee for the positive reading of our manuscript. We would particularly thank them for highlighting the manuscript's value as a stepping stone.*

1) descriptions of the paper's methodology are sometimes confusing, and would benefit from further refinement;

*Response: We hope to have clarified the methodology sufficiently. We did not include an additional figure to illustrate it.*

2) I have several concerns about the paper's methodology, which seem like they limit the success of finding analogues; a revised methodology may result in a more successful reconstruction and a more interesting paper;

*Response: We thank the referee for raising the possibilities to improve the manuscript. We address the comments below.*

and 3) the figures could be improved. These points are expanded upon in the "Specific comments" section below. I feel like these are important points which should be addressed.

*Response: We tried to improve the visualisations.*

**Specific comments:**

1. In a method-heavy paper, extra care must be taken to ensure that the paper is intuitive. When reading the paper, however, I had a variety of questions about how the method worked and what factors were keeping it from working better. Several of these confusions are listed below:

*Response: We thank the referee for their detailed criticisms.*

- The discussion of ellipses, which represent uncertainty in time and magnitude, is somewhat confusing at first, and it took me some time to understand they were used within the methodology.

*Response: We extend our description of the ellipses and try to be clearer in our terminology throughout the manuscript. We decided against a figure to specifically explain their role.*

- The relevance of the 90% vs. 99% vs. 99.99% cutoffs is not clearly explained. It appears that they refer to percentiles of magnitude and time uncertainty, but how are they calculated?

*Response: This is part of the calculation of the uncertainty ellipses. We clarify theses aspects in the revised manuscript.*

- Some aspects of panels d and e in Figs. 5 and 8 are unclear. As far as I understand, these panels are showing the annual data underlying the selected 101-year means, but I'm not sure

what I should take away from them. Can their purpose be better explained, or can they be revised to show the relevant points in a more intuitive manner?

*Response: We describe the purpose of these panels in more detail.*

In particular, I don't understand the lines marked as "examples". Also, it may help if the "examples" were solid lines rather than dotted/dashed. In general, I would encourage the authors to read through the manuscript again with a focus on making explanations clearer and more intuitive.

*Response: We tried to clarify the panels and their descriptions.*

2. I am concerned about several aspects of the methodology, which seem like they may prevent the method from finding good analogues. My main two concerns are described below, with the second point being the more important of the two. Unless I am misunderstanding something (see point #1 above), I would like to see these concerns discussed or, preferably, directly accounted for within the methodology.

*Response: We thank the referee that they detail their concerns so carefully.*

2.1. Uncertainty Ellipse Edge-Effects:

The use of uncertainty ellipses, which have a hard cutoff, may prevent the method from finding good analogues. One example of this may be imagined at the left and right "edges" of the ellipses. At the left and right edges of the ellipse, the vertical extent of an ellipse (representing magnitude uncertainty) becomes very small, eventually reaching 0. If the method is looking for analogues near the edge of one of these ellipses, the range of an "acceptable" analogue would be very narrow, rejecting many potential candidates.

*Response: The referee is correct in this description. The ellipse describes a two dimensional interval in which we search. Thus, at this edge, there is little probability of finding a valid analogue considering the age uncertainty and the data uncertainty. An alternative to this approach would be to assume that both uncertainties affect the selection independently and, in turn, taking a rectangle. Even then, we would have edge effects though of a different kind. Here, we use a two dimensional Gaussian to represent the effect of proxy-uncertainty and dating-uncertainty on our tolerance to accept an analogue. Therefore, our current edge effect is not a bug but a feature. We want the data to allow for less analogues in either direction. We will try to clarify this.*

Let's take the scenario in section 2.2.4 as an example. The paper states that there is a hypothetical proxy value at 500 BP, with age uncertainties from 600 to 400 BP. This hypothetical uncertainty ellipse stretches between 600 and 400 BP, with its maximum magnitude uncertainty at 500 BP. If an analogue search is conducted at 500 BP, the method accepts all points within the full uncertainty range of the ellipse. However, if an analogue search is conducted at 401 BP, the uncertainty range of the ellipse (i.e. the height of the ellipse, similar to the ones visualized in Fig. 2b) would be much smaller, therefore rejecting many potential analogues. This seems counter-intuitive to me. Wouldn't it make more sense to broaden the magnitude uncertainty as you get farther from the central age date, since we are less sure that the data point is applicable as we get farther from its original dated age?

*Response: Wouldn't we, in this alternative scenario, then overemphasize the ranges far away from the original dated age?*

*Apparently we were not clear enough in explaining how to interpret the ellipses. The ellipses do not represent the uncertainty ranges in the value of the proxies, but rather the tolerance with which we accept analogues. An analogue that may be numerically close to the target should not be as easily accepted for dates that are far off the median proxy date as they are accepted for dates that are temporally closer to the proxy median age. Essentially the ellipses define a weighting scheme (although with binary weights) according to that tolerance level. If we adopt the scheme suggested by the reviewer, we would select many analogues that appear to match the proxy for dates at the edges of the dating-uncertainty interval, where actually we are very unsure that the proxy is delivering any useful information about the*

*climate at that point in time. In that situation we do require the analogue to be numerically very close to the proxy, otherwise we would reject it. In contrast, we are laxer for dates that are temporally close to the proxy age.*

*We hope that our revisions make these points more clearly.*

This issue may only be a problem at the start or end of a proxy record, or near a very long gap, but I expect that it would become more and more of a problem as the method is applied to more proxies, which naturally have different start and end dates. Unless I'm misunderstanding the method, I think that a better handling of these "edge effects" would help the method find more valid matches. Perhaps rectangles could be used instead of ellipses, since I see no reason that magnitude uncertainty should be decreased near the edge of temporal uncertainties. If anything, I would expect a particular point to become less precise toward the edges, not more precise. Since altering the method to address this would likely be too much work, I think that this point should be at least be mentioned in the paper.

*Response: We try to clarify the description of our method and our assumption on why to use an ellipse and not a rectangle. A rectangular tolerance region would lead to accepting analogues for dates far off the most probable date with the same tolerance as for dates that are temporally closer to the most probable age. This would be fine if we consider the dating as having uniform probability over the dating uncertainty range, which is not plausible. We shortly discuss the effect of a rectangular tolerance range.*

2.2. Potential for Outliers to Cause Method Failure:

The paper mentions that the method uses the absolute temperatures calibrated from proxies, rather than anomalies. The authors discuss the problems surrounding the choice of absolute values vs. anomalies, but I'm concerned that biases in the absolute value of a single record (or simply non-climate proxy variations) could cause the method to fail. Consider applying this methodology to a group of proxies where a single proxy has been accidentally calibrated to be too warm by 5 degrees C. An error like this could hypothetically cause every single potential analogue to fail for the entire length of the proxy, as it's possible that no modeled state would show a spike of temperature at that particular location compared to everywhere else in the region. This means that the method would fail even if every other proxy were a perfect recorder of climate.

*Response: We understand the concern of the referee. By considering the uncertainty of the record we would hope to be able to compensate for such an error at least partially. This should be independent of whether it is a systematic bias in the record or whether only a single measurement is erroneous. However, we cannot exclude that such biases lead to a failure of the method.*

If a single problematic proxy can cause the whole method to fail, this problem will only become more likely to occur as the method is applied to a larger and larger proxy database. As it is, the method has trouble finding analogues with even a small set of proxies (as little as 7 proxies for the E09 case). This seems like a fundamental problem with the method, limiting its future application. The authors try to widen the group of successful analogues by using wider uncertainty bands, including/removing records, and using annual model states rather than 101-year means, but I don't think that any of these solutions fix the underlying problem, which I suspect is the use of a binary match/mismatch dichotomy with the uncertainty ellipses. Using strict match/mismatch criteria probably makes the method overly sensitive to mismatches with single proxies. The use of a skill metric, as used in other work, may help alleviate issues arising from a subset of problematic records. Alternately, perhaps analogues could be accepted even if a certain percentage of the proxies don't match, to account for biases and non-climate noise within the proxy data set.

*Response: We would again like to argue that including the uncertainty of the records should compensate for this problem. Problems with the reliability of the proxies affect any reconstruction method. One can assume that the method compensates for them or one can accept that unreliable proxies reduce our ability to make reliable estimates about past climates.*

*We think the failure of finding analogues is rather due to the insufficient pool of analogues and less due to problems with the reliability of the proxies.*

*We shortly discuss experiments where we allow that it is enough if N-1, N-2, or 75% of all proxy records are matched.*

To the authors' credit, much of the paper does discuss potential problems with the method, and also suggests ways that things could be improved in the future. Indeed, the paper appears to be an exploration of how to account for age/magnitude uncertainties, rather than the presentation of a finished methodology. However, the paper would be much more satisfying to read if some of these issues were implemented directly, hopefully leading to a more complete reconstruction than the one shown in Fig. 8.

*Response: We do not follow up on our suggestion in the initial response to rewrite the manuscript as an exploration of how to handle the uncertainties in the analogue method.*

*We hope that our rewriting sufficiently addresses the referee's points.*

*We, still, do want to emphasize that failure of a method may primarily signal that our data (cf. our proxy information or our simulation pool or both) are insufficient to inform us about a problem at hand. We do not claim here that this is the case with our paper, we just want to emphasize that completeness of a reconstruction is not an information about the quality of a method, a paper, or the reconstruction.*

If this is not possible, I would at the very least like to see the following: 1) More discussion of the methodological problems mentioned above. 2) A different title, which accurately reflects the fact that the paper's methodology is a work-in-progress rather than a finished method. As-is, the title makes it sound like this paper demonstrates a finished methodology, when it appear to be an exploration of uncertainties which may lead to a better method in the future. Because of this, a better title might be something like: "Considerations of proxy uncertainties within the analogue method of paleoclimate reconstruction".

*Response: We hope that our revisions sufficiently discuss the problems described by the referees and those already mentioned in our submitted manuscript.*

*We adapt the title.*

3. In general, several of the figures could be improved. For example, the black and red colors in Fig. 3 are difficult to distinguish, and the lines in Figs. 5c and 8c are difficult to interpret, since they use similar thicknesses and opacities. Improving the figures may also help make the methodology more intuitive, as I commented about in point #1 above.

*Response: We reconsider all our visualisations. Particularly, we hope to have addressed the comments with respect to Figures 3, 5, and 8.*

A few other minor questions/concerns: Why only use 101-year means, rather than means which vary site-by-site to better reflect the temporal characteristics of individual proxy records? Also, why does the pseudoproxy experiment only use summer means, as mentioned in line 30 on page 10? And why does the number of sites differ between the pseudoproxy experiment (Fig. 1a) and the real experiment (Fig. 1b)? I had other questions about methodological choices while reading the paper, but the major points discussed throughout the review above seemed like the most important.

*Response: The referee is correct. Ideally one should use site-specific means and adapt these for each individual measurement. The information to do so is not necessarily available - as stated in the manuscript. We considered this at one point but did fail to achieve a computationally effective implementation at that point.*

*We used summer means as we made the assumption that this is a representative season for the proxy locations. We change this now to annual means as we also changed the simulation from which we compute the pseudoproxies.*

*We also now use the same number of locations in the pseudoproxy experiment as in the main real-world experiment. We decided against using the seasonal representations from the*

*real-world case.*

A final technical note: some figures (especially Fig. 4) have so many lines that the paper is difficult to print (it gets stuck on a "flattening" step for a long time).

*Response: We have to prevent this happening. We will reconsider all our visualisations and the output format for these cases. To our knowledge this should not happen anymore.*

In summary, while the paper focuses on an interesting and useful approach to paleoclimate reconstruction, I think that several things need to be improved before it can be considered for publication. A fundamental problem is that this appears to be a method paper, but the method doesn't work very well. If the method cannot be improved, the concerns above should at least be addressed and the paper should get a new title which better reflects its contents.

*Response: We thank the referee for their fair evaluation of our manuscript. We, however, want to express our surprise that technical notes should only deal with well working methods. We will try to account for all the reviewer's suggestions.*

Finally, despite all of my comments and concerns, I do think that this is an interesting and potentially useful method, and I hope that further progress is made in the future.

*Response: We want to thank the referee once more for their generous evaluation.*

[revised manuscript text omitted]

---

## Referee Report (RR1)

**General Comments**

I think the manuscript has been substantially improved, especially with the clarification of the method and the inclusion of a better designed Pseudo Proxy Experiment. However, I still have some concerns regarding: (1) the clarity of the method's explanation (which should be central for a technical note), (2) some unsustained conclusions and (3) the repetition of several sentences along the text.

I indicate lines and pages as in the file with tracked changes.

%%

(1): Method's explanation

1. In this version of the text the method is much better explained. However, I still can't find anywhere a description of what is meant by 90%, 99%, etc. uncertainty ellipses.

The first appeareance of the 90%, 99%, etc. in the text is in the following sentence (line 33, page 7):

"We can do so at different levels of proxy-time uncertainty, e.g., 90% or 99%, comparable to common expressions of uncertainty intervals."

At other time it says: "To do so, we consider the proxy values valid at all dates within their a 90% dating uncertainty, then identify the range of these values, and take the mid-point of the range as the proxy value for this date." Which, to my point of view makes no sense.

I think a clear definition of x% ellipse uncertainty need to be provided, as it is a main part of the methodology.

2.
Line 23, page 17
"Valid analogues are those simulation fields that are within the resultant tolerance envelopes for all pseudoproxy locations available for a date."

I find this sentence of extreme importance for the methodology. However, it only appears in the Results section. I think it should be moved to the description of the method, where it is much needed. The term "valid analogues" appears several times in the text but I think in this sentence is the first time that a proper definition is given, thus its relevance.

%%

(2): Unsustained conclusions

3. Line 11, page 29

"Generally, the reconstruction success appears to be better for proxy setups that only include UK0
records (Figure 11b,c)"

How can you say is „better" when you don't know the truth? If you want to make a conclusion
like this you should do this experiment in PPE setup. There is no base for such a statement.

Line 31, page 33

"That is, the method performs slightly better using 51-year averaged simulation data than using 101-year averaged data, and it performs even better using interannual data."

Again, I find this conclusions dangerous and erroneous, not derived from the present analysis. how do you know is better when you did experiments only when not knowing the truth (not PPE setup)?

I think this conclusion doesn't follow from your results and is inappropriate.

From you analysis you can only say that using 51 means leads to finding more possible analogues, if this. Whether this leads to a better reconstruction is not shown in this paper.

%%

(3): Repetition of sentences:

Several time in the text I found (almost identical) sentences repeated. Please, read the new text carefully and avoid this type of problems. Information should be more organized and not unnecessary repeated.

Here only 3 examples, but there are many more:

- Line 23, page 6:

We construct the pseudoproxies following the procedure of Bothe et al. (2019a, more specifically their ensemble approach)

Line 7, page 14:

We calculate the pseudoproxies following the procedure of Bothe et al. (2019a, more specifically their ensemble approach) but omit their effective dating uncertainty error term.

- Line 19, page 6:

The data is available in monthly resolution for the full simulation period for air temperature in 1.5 meter height, and as snaphots every ten simulation years for surface temperature

Line 18, page 14:

Data is available in simulation monthly resolution for air temperature in 1.5 meter height but only as snaphots every ten simulation years for surface temperature.

- Line 32, page 6

Figure 1a shows the 17 pseudoproxy locations and allows to identify their slight offset to the real proxy locations (Figure 1b) due to the discrete character of the simulation data

Line 1, page 16

Figure 1a shows the 17 pseudoproxy locations. These are close to the realistic proxy locations.

%%%%%%%%%%%%%%%%%%%%%%%%%%%%%%%%%%%%%%%%%%%%

**Specific Comments**

Figure 1: It is really hard to notice any differences between panels (a) and (b). Please, plot in only 1 panel and overlay a grid.

Figure 3: Include 1 more column for P01 setup, in a different colour.

Figure 6 and 9: Panel (a) is in Kelvin while all the other panels and text are in °C. Please, unify.

Also, the captions are really unorganized and difficult to follow. The first describe all the panels and then go on to describe them all again.

Figure 7 and 10: Add latitude and Longitude.

You use „QUEST-Famous" and „ QUEST Famous". Please, select only one denomination.

Typo: snahots (appears twice in the text)

---

## Referee Report (RR2)

Paper review

Title: Technical Note: Considerations on using uncertain proxies in the analogue method for spatiotemporal reconstructions of millennial-scale climate

Authors: Oliver Bothe and Eduardo Zorita

The paper describes an analogue method of deriving spatially-complete temperature fields from sparse proxy records by finding modeled climate fields which match the proxy data within uncertainty bounds. The paper first tests this method using a pseudoproxy framework and then does several experiments using real proxy data. The authors also consider alternate experimental designs, such as using different proxy records and different criteria for finding valid matches.

Overall, I think that this is worthwhile research, and it is always good to explore methodologies which can supplement our incomplete perspective on past climate from proxy records. Additionally, the paper is much improved from the first version. The authors do a good job discussing potential pitfalls or improvements to the method, which is useful since future improvements would benefit the methodology. The primary shortcoming of the paper is the fact that the method doesn't work as well as one would hope. Instead of finding insightful analogues for past climate, the method tends to either find too many analogues (resulting in a very broad estimate) or none at all. The authors acknowledge this shortcoming, which is good to see, and the title accurately reflects the paper's "work-in-progress" nature. As it is, I think this is a useful paper, with the understanding that the method presented does not appear to represent a final product but rather a stepping stone toward more successful methodologies in the future. Many of my concerns with the first draft, such as the somewhat confusing description of the methodology and unclear elements of the figures, have been improved. However, I still do not agree with the authors use of uncertainty ellipses. My comments and suggestions are discussed in more detail below.

General comments:

My general comments concern the overall design of the methodology. While the method described in the paper has clear limitations, I think that this is appropriately discussed to a large degree in the paper. The authors have tested, or at least discussed, some ways of improving the method in the future, which is commendable. However, I want to return to two of my comments from my first review:

1. First, the pass/fail dichotomy of the analogue search seems very strict. If one (or more) proxy has a considerable bias, this single record could cause the method to fail. This can be seen in the author's choice to exclude a proxy on p.11, stating that "we find (not shown) that including this record puts very strong constraints on the analogue candidates and can reduce the chance of finding valid analogues." The excluded proxy is "the alkenone unsaturation ratios of Bendle and Rosell-Melé (2007". This limitation could be mitigated if proxy uncertainty values were varied on a proxy-by-proxy basis (for example, by assigning a large uncertainty to certainty records individually, rather than increasing the uncertainty percentiles for all proxies simultaneously), but it may be difficult to find reasonable uncertainty values for each proxy. The paper decides to use consistent values across proxy records. To the paper's credit, the authors explore alternate experimental designs where analogues are found valid even if they fail to match a certain number/percentage of proxies. This is good. I'm not sure if anything else needs

to be done regarding this point in the paper, but I would like the authors to continue considering this point if they pursue this research more in the future.

2. Secondly, I still feel that the shape of the uncertainty ellipses may lead to difficulties finding valid analogues. As I see it, the main problem with the use of ellipses is that the analogue search becomes stricter, rather than more lenient, for searches farther from the original date of a given datapoint. This presents a problem at the ends of records, and will only get worse as more proxies are included. For example, consider Figure 2b. As a thought experiment, imagine that the data point shown at 7.8 kYa BP is the oldest point in this proxy record. When searching for analogues at 7.8 kYa, any value from ~22-30 degrees C would be deemed acceptable. However, when searching for analogues at ~8.5 kYa (i.e. just inside the edge of the uncertainty ellipse), only a very small range of values around 25 degrees C would be considered acceptable. Why would the analogue search be stricter for a period where the data point is less likely to be valid? Shouldn't it be more lenient, since we don't know if the datapoint is even relevant at that age? Furthermore, if we look for analogues at ~8.6 kYa (i.e. just outside of that uncertainty ellipse), any temperature would be deemed valid for this location (i.e. this data point provides no constraint on the analogues chosen for that date). This shift from somewhat strict (near the original age point) to extremely strict (at the border of the ellipse) to not-a-factor-at-all (outside of the ellipse) doesn't make sense to me. Wouldn't it make more sense for a data point to have less impact on the analogue selection for ages when it is less likely to be relevant? In the authors' response, they ask "*Wouldn't we, in this alternative scenario, then overemphasize the ranges far away from the original dated age?*" Yes, but that seems appropriate. The method would result in larger uncertainty bounds for times when the proxies provide less of a constraint. I'm not sure how this would be implemented mathematically in the methodology (perhaps through the use of rectangular uncertainty bands, as explored in the paper, to the authors' credit), but the current methodology is unsatisfying to me. One idea would be to have proxy uncertainties increase as one gets farther from central age of each proxy data point, and then require that the analogues be within the bounds of all proxy data points. The authors explain their reasoning for their choices in their response to my first review, but I disagree with their rationale. It is possible, however, that I am overlooking something.

I fear that the problems in both of the points above will become worse as more proxy records added. Users of the methodology could deal with this by excluding certain proxies (as mentioned in the example earlier), but this seems time intensive and potentially somewhat arbitrary. Regarding my second point above, it would probably take a lot of time to change the method entirely, but I would like to see a little more discussion (perhaps an extra paragraph) about this point in the paper.

Specific comments

- Page 1, line 19. By "climate state" do you mean the spatial climate field?
- Figure 1 caption. Perhaps mention that the pseudoproxy locations are slightly offset from the actual proxy locations. I know that this is mentioned in the paper's text, but including a brief statement in the Fig. 1 caption should help prevent people from wondering why it appears that the same panel is shown twice.
- Page 6, lines 27-28. The statement "Using anomalies circumvents this issue" makes it sound like you are going to use anomalies, when you don't. Consider rephrasing to make this clearer.
- Page 6, line 32: "Dansgaard-Oeschger (DO)" should be "Dansgaard-Oeschger (DO) events".

- Page 7, line 5. I would change "original temperature units" to "absolute temperature units", since "original" makes it sound like you would be using the measured proxy units.
- Page 8. Regarding the 90% and 99% uncertainty values (and later 99.9% and 99.99%), are temporal uncertainty values the same for each proxy?
- Page 9, line 17-18. The phrase "the envelope does not necessarily cover all years within the period of interest" doesn't make much sense to me. Can you rephrase?
- Page 9, lines 32-33: The paper states "…increase the probability to find a valid analogue at a certain date". But what if widening the time uncertainties made new proxies relevant to some ages? Couldn't that actual reduce the likelihood of finding an analogue (see page 24, lines 24-26)?
- Page 12, lines 14-15. The paper says "Regarding proxy uncertainty, we try to identify as complete uncertainty estimates as possible from Marcott et al. (2013) and their references. For the sake of simplicity, we decided to assume an uncertainty of sigma = 1K for all proxies." If you use a common number for all proxies, why did you look at individual proxy values? Is 1K close to the mean or median value?
- Page 15, lines 1-2: I'm not sure exactly what this sentence means: "The latter modification also avoids that individual data points have an overly strong influence within our envelopes of tolerance." Can you rephrase?
- Page 15, line 4: I'm not sure what "black lines" refers to in this sentence.
- Page 15, line 5: Remove the word "below".
- Figure 6, panel c: Perhaps rearrange the labels so that "QUEST FAMOUS target" and "pseudoproxy" are listed next to each other (since the pseudoproxy is generated from the QUEST model output, right?). Also, put the "Analogue median" and "Analogue range" labels next to each other since they're also related.
- Figure 6, panel d: The "Valid analogue examples" label only has one color, but the examples are shown in two different colors. Could you add the second color next to the label?
- Figure 6. Panels d and e took me a while to understand. If possible, could you try to explain it in a more intuitive way?
- Page 21, line 11: Is there any particular reason that age 8000 was chosen for the examples (or age ~2430 for the examples in figure 9)?
- Page 21, line 14. The paper says "Their local range at no point exceeds 4 degree Celsius". However, it looks like you are referring to the deviation from the median, rather than the total max-min range. Can you check this number and rephrase if necessary? Please do this for the "20 degrees Celsius" value a few lines later as well.
- Figure 9, panels d and e: In these two panels, the two examples look like time-shifted versions of each other. Why is this? Is it because the same 101-year mean was chosen for each date?
- Figure 11 caption: The caption says "All panels include the median, 90% interval, and full range for the reconstructions". However, only one range is shown. Is the displayed range the 90% interval or the full range? Please fix this line, both here and in the caption to Fig. 12.
- Page 26, line 5: I would remove the comma after "Please".
- Page 26, line 6: Consider changing "further" to "also".
- Figure 12 caption. I would change "are valid if they fail" to "are valid even if they fail".
- Page 28, line 12: "warmer" than what?
- Page 28, lines 23-25: Consider discussing why the "single best analogue" differs from the default experimental setup (by large amounts for some ages). Is it because the "best" analogue may fail at one or more locations but still produce a better overall fit than other analogues? Since the "single best" methodology sounds similar to previous analogue methods, it would be interesting

to explore why it diverges from your main experiment, and whether your method provides benefits over the "single best" analogue.

- Page 30, line 2. I don't understand the use of "e.g." in this sentence. Please rephrase.
- Page 30, line 11. What does "seasonal sensitivities" in this sentence refer to? I thought that you already considered seasonality in your proxy-to-model comparisons (as stated on page 12, lines 10-12). Or, if all proxy comparisons are made to annual-mean model data, than this needs to be corrected at several points in the paper (e.g. page 12 and Table 1 "season used").
- Page 32, line 30. The method succeeds only for some ages.
- Page 34, line 8. What does "central" mean in this sentence?

Despite the quantity of these comments, I thought that the paper provides useful considerations and tests of the analogue method. Although I still disagree with the shape of the uncertainty ellipses, I think this is useful and interesting research.

---

## Author Response (AR3)

Dear editor, dear referees,

We hope our response finds you well.

We thank you for your careful and encouraging evaluation of our manuscript. Before we provide a detailed response to your comments, we have to highlight one relevant change in the revised manuscript.

During reconsidering the single-best analogue reconstruction test in response to referee #2's question on the discrepancy between the single best analogue and the tolerance area analogue reconstructions, we noted a bug in the preparation of the simulation data.

The preprocessing of the model data into the input for the analogue search resulted in the introduction of invalid values (Not A Number) for one proxy location in one model grid which was used in three simulations. Specifically, this affected the data from simulations with MIROC-ESM (Sueyoshi et al., 2013: Set-up of the PMIP3 paleoclimate experiments conducted using an Earth system model, MIROC-ESM, Geosci. Model Dev., doi:10.5194/gmd-6-819-2013).

This resulted in the fact that simulation data from these three simulations could never be selected as valid analogues in our tolerance area approach but were included in the single best analogue reconstruction and chosen as analogues there because the distance algorithm discounted the locations without numerical information. That is, the single best reconstruction was effectively calculated using a reduced number of locations.

Because the simulations were never part of any of our main reconstruction tasks, we choose the following solution to this issue. We remove from the manuscript the simulations for which data was erroneously prepared, and we redo the single best reconstruction. We acknowledge that this post-hoc modification of the experiment description is not optimal.

We invite you, the referees and the editor, to comment on this issue. We also invite you to provide recommendations whether the post-hoc change in experimental design should be mentioned in the manuscript.

Please see the tracked changes or the manuscript file for the details of all changes. Below we also list the major changes to the manuscript.

On behalf of the authors

Yours sincerely

Oliver Bothe

**List of changes**

- Experiment design
  - As noted above in our letter to the editor and the referees we had to adapt the single-best experimental setup to account for a bug. Subsequently discussions of the results had to be adpated as well.
- Abstract
  - Minor change
- Introduction
  - Minor clarifications
- Analog method, assumptions, and data
  - Clarifications and modifications on text and figures in response to the referees' comments and suggestions.
- Results
  - Adapted description of single-best results to the changed setup.
  - Clarification of figure captions in response and beyond response to the referees' comments.
  - Minor modifications in response to the referees' comments.
- Discussion
  - Additions in response to the referees' comments.
- Appendix
  - Minor modification.

**Editor**

Dear authors,

Thank you for submitting a revised version of your manuscript, which has now been evaluated by the reviewers. Both reviewers agree on the significance of your contribution. However, they also suggest some additional changes to improve clarity. The second reviewer also remains doubtful about the methodology, but they think the approach you suggest is worthwhile exploring and they are hence favourable to sharing it with the community. Nevertheless, I do encourage to take their concerns seriously and address them in a revised manuscript. The first reviewer also asks for additional clarification and some rethinking of the conclusions.

Best regards, Lukas Jonkers

*Response: Dear editor, thank you for your comments. Below we respond to the reviewers' comments and try to address their concerns. We particularly aim to consider the points highlighted in your comments.*

**Referee 1**

General Comments

I think the manuscript has been substantially improved, especially with the clarification of the method and the inclusion of a better designed Pseudo Proxy Experiment. However, I still have some concerns regarding: (1) the clarity of the method's explanation (which should be central for a technical note), (2) some unsustained conclusions and (3) the repetition of several sentences along the text.

*Response: Thank you for this positive evaluation. Below we try to address all your points in detail.*

I indicate lines and pages as in the file with tracked changes.

(1): Method's explanation 1. In this version of the text the method is much better explained. However, I still can't find anywhere a description of what is meant by 90%, 99%, etc. uncertainty ellipses.

The first appeareance of the 90%, 99%, etc. in the text is in the following sentence (line 33, page 7):

"We can do so at different levels of proxy-time uncertainty, e.g., 90% or 99%, comparable to common expressions of uncertainty intervals."

At other time it says: "To do so, we consider the proxy values valid at all dates within their a 90% dating uncertainty, then identify the range of these values, and take the mid-point of the range as the proxy value for this date." Which, to my point of view makes no sense.

I think a clear definition of x% ellipse uncertainty need to be provided, as it is a main part of the methodology.

*Response: We understand the referee's concern. We address this now. We provide a quick derivation of our percentage tolerance view from classical confidence intervals and the chosen language there.*

*We also recognize that we use percentages differently, in so far as we also use it for the coverage of the reconstructions. We do not think we can avoid this double usage.*

*However, we agree that we have to change the occurrence mentioned by the referee. First, the sentence is apparently grammatically incorrect. Second, we change the sentence to: "...we identify the range of possible proxy values within a certain dating uncertainty, and take the mid-point of that range as the proxy value for this date." We stress again here that this procedure is only used because we (i) consider proxies that are irregularly spaced in time but (ii) want to identify a single value for a date that (iii) allows us to compute a Euclidean distance.*

2.

Line 23, page 17

"Valid analogues are those simulation fields that are within the resultant tolerance envelopes for all pseudoproxy locations available for a date."

I find this sentence of extreme importance for the methodology. However, it only appears in the Results section. I think it should be moved to the description of the method, where it is much needed. The term "valid analogues" appears several times in the text but I think in this sentence is the first time that a proper definition is given, thus its relevance.

*Response: We account for the referee's comment by trying to add this information more clearly at more locations. We want to stress however, that it was explicitly mentioned in our method section.*

(2): Unsustained conclusions 3. Line 11, page 29 "Generally, the reconstruction success appears to be better for proxy setups that only include UK0 records (Figure 11b,c)"

How can you say is „better" when you don't know the truth? If you want to make a conclusion like this you should do this experiment in PPE setup. There is no base for such a statement.

*Response: We appreciate the referee's concern and will clarify this. However, we feel forced to emphasize that our conclusion is not unsupported. By saying the success appears to be better we want to say that it appears that the reconstruction is more complete. We do not want to say anything about the quality of the resultant reconstruction in its relation to reality. We wrote so in the subsequent sentence. We change this now to: Generally, the method appears to provide more complete reconstructions among our proxy setups for those that only include $U_{37}^{K'}$ records (Figure 11b,c). That is, such consistent sets of proxies provide a more continuous reconstruction for both local tolerance assumptions.*

Line 31, page 33

"That is, the method performs slightly better using 51-year averaged simulation data than using 101-year averaged data, and it performs even better using interannual data."

Again, I find this conclusions dangerous and erroneous, not derived from the present analysis. how do you know is better when you did experiments only when not knowing the truth (not PPE setup)?

I think this conclusion doesn't follow from your results and is inappropriate.

From you analysis you can only say that using 51 means leads to finding more possible analogues, if this. Whether this leads to a better reconstruction is not shown in this paper.

*Response: The referee is correct in their last paragraph that this is what we can conclude. And this is exactly what we are stating in the paragraph.*

*We agree with the referee that we have to clarify this. However, again we want to stress that the whole paragraph from which the quote is selected deals with the reconstruction success in terms of finding analogues at all and not in terms of proximity to the truth. We indeed think the paragraph was written clearly already, but we modify it slightly.*

(3): Repetition of sentences:

Several time in the text I found (almost identical) sentences repeated. Please, read the new text carefully and avoid this type of problems. Information should be more organized and not unnecessary repeated.

*Response: We acknowledge the referee's concerns. We put effort into removing such duplications. We hope to have succeeded.*

Here only 3 examples, but there are many more:

- Line 23, page 6:

We construct the pseudoproxies following the procedure of Bothe et al. (2019a, more specifically their ensemble approach)

Line 7, page 14:

We calculate the pseudoproxies following the procedure of Bothe et al. (2019a, more specifically their ensemble approach) but omit their effective dating uncertainty error term.

- Line 19, page 6:

The data is available in monthly resolution for the full simulation period for air temperature in 1.5 meter height, and as snaphots every ten simulation years for surface temperature

Line 18, page 14:

Data is available in simulation monthly resolution for air temperature in 1.5 meter height but only as snaphots every ten simulation years for surface temperature.

- Line 32, page 6

Figure 1a shows the 17 pseudoproxy locations and allows to identify their slight offset to the real proxy locations (Figure 1b) due to the discrete character of the simulation data

Line 1, page 16

Figure 1a shows the 17 pseudoproxy locations. These are close to the realistic proxy locations.

*Response: We addressed the mentioned occurrences and hope to have addressed further problematic duplications.*

**Specific Comments**

Figure 1: It is really hard to notice any differences between panels (a) and (b). Please, plot in only 1 panel and overlay a grid.

*Response: We regard overlaying a grid as not helpful as it adds an unnecessary element. We now show the information from both panels in one panel.*

Figure 3: Include 1 more column for P01 setup, in a different colour.

*Response: We do so.*

Figure 6 and 9: Panel (a) is in Kelvin while all the other panels and text are in °C. Please, unify.

*Response: We do so.*

Also, the captions are really unorganized and difficult to follow. The first describe all the panels and then go on to describe them all again.

*Response: We tried to reduce redundancy.*

Figure 7 and 10: Add latitude and Longitude.

*Response: We do so.*

You use „QUEST-Famous" and „ QUEST Famous". Please, select only one denomination.

*Response: We thank the referee for noting this, we change the one occurrence of QUEST-FAMOUS to QUEST FAMOUS*

Typo: snahots (appears twice in the text)

*Response: We thank the referee for noting this.*

**Referee 2**

**General Comments:**

Paper review

Title: Technical Note: Considerations on using uncertain proxies in the analogue method for spatiotemporal reconstructions of millennial-scale climate

Authors: Oliver Bothe and Eduardo Zorita

The paper describes an analogue method of deriving spatially-complete temperature fields from sparse proxy records by finding modeled climate fields which match the proxy data within uncertainty bounds. The paper first tests this method using a pseudoproxy framework and then does several experiments using real proxy data. The authors also consider alternate experimental designs, such as using different proxy records and different criteria for finding valid matches.

Overall, I think that this is worthwhile research, and it is always good to explore methodologies which can supplement our incomplete perspective on past climate from proxy records. Additionally, the paper is much improved from the first version. The authors do a good job discussing potential pitfalls or improvements to the method, which is useful since future improvements would benefit the methodology. The primary shortcoming of the paper is the fact that the method doesn't work as well as one would hope. Instead of finding insightful analogues for past climate, the method tends to either find too many analogues (resulting in a very broad estimate) or none at all. The authors acknowledge this shortcoming, which is good to see, and the title accurately reflects the paper's "work-in-progress" nature. As it is, I think this is a useful paper, with the understanding that the method presented does not appear to represent a final product but rather a stepping stone toward more successful methodologies in the future. Many of my concerns with the first draft, such as the somewhat confusing description of the methodology and unclear elements of the figures, have been improved. However, I still do not agree with the authors use of uncertainty ellipses. My comments and suggestions are discussed in more detail below.

*Response: We thank the referee for their detailed assessment. Below, we respond to individual points.*

General comments:

My general comments concern the overall design of the methodology. While the method described in the paper has clear limitations, I think that this is appropriately discussed to a large degree in the paper. The authors have tested, or at least discussed, some ways of improving the method in the future, which is commendable. However, I want to return to two of my comments from my first review:

1. First, the pass/fail dichotomy of the analogue search seems very strict. If one (or more) proxy has a considerable bias, this single record could cause the method to fail. This can be seen in the author's choice to exclude a proxy on p.11, stating that "we find (not shown) that including this record puts very strong constraints on the analogue candidates and can reduce the chance of finding valid analogues." The excluded proxy is "the alkenone unsaturation ratios of Bendle and Rosell-Melé (2007)". This limitation could be mitigated if proxy uncertainty values were varied on a proxy-by-proxy basis (for example, by assigning a large uncertainty to certainty records individually, rather than increasing the uncertainty percentiles for all proxies simultaneously), but it may be difficult to find reasonable uncertainty values for each proxy. The paper decides to use consistent values across proxy records. To the paper's credit, the authors explore alternate experimental designs where analogues are found valid even if they fail to match a certain number/percentage of proxies. This is good. I'm not sure if anything else needs to be done regarding this point in the paper, but I would like the authors to continue considering this point if they pursue this research more in the future.

*Response: We thank the referee for this comment.*

*We agree that future considerations have to include a thorough review of criteria when analogues are considered to be valid under uncertainty. That is, the method will benefit if we can find robust criteria that not only allow for uncertainty in the input proxy data but even explicit wrongness of this data.*

*As a side note: we are not sure that allowing for different tolerance thresholds among multiple proxies is a promising strategy as it may provoke criticisms of overly subjective assumptions. However, that does not imply that one should not try it.*

*In view of the referee's comment, for now, we do not add additional comments on this topic in the manuscript.*

2. Secondly, I still feel that the shape of the uncertainty ellipses may lead to difficulties finding valid analogues. As I see it, the main problem with the use of ellipses is that the analogue search becomes stricter, rather than more lenient, for searches farther from the original date of a given datapoint. This presents a problem at the ends of records, and will only get worse as more proxies are included. For example, consider Figure 2b. As a thought experiment, imagine that the data point shown at 7.8 kYa BP is the oldest point in this proxy record. When searching for analogues at 7.8 kYa, any value from 22-30 degrees C would be deemed acceptable. However, when searching for analogues at 8.5 kYa (i.e. just inside the edge of the uncertainty ellipse), only a very small range of values around 25 degrees C would be considered acceptable. Why would the analogue search be stricter for a period where the data point is less likely to be valid? Shouldn't it be more lenient, since we don't know if the datapoint is even relevant at that age? Furthermore, if we look for analogues at 8.6 kYa (i.e. just outside of that uncertainty ellipse), any temperature would be deemed valid for this location (i.e. this data point provides no constraint on the analogues chosen for that date). This shift from somewhat strict (near the original age point) to extremely strict (at the border of the ellipse) to not-a-factor-at-all (outside of the ellipse) doesn't make sense to me. Wouldn't it make more sense for a data point to have less impact on the analogue selection for ages when it is less likely to be relevant? In the authors' response, they ask "Wouldn't we, in this alternative scenario, then overemphasize the ranges far away from the original dated age?" Yes, but that seems appropriate. The method would result in larger uncertainty bounds for times when the proxies provide less of a constraint. I'm not sure how this would be implemented mathematically in the methodology (perhaps through the use of rectangular uncertainty bands, as explored in the paper, to the authors' credit), but the current methodology is unsatisfying to me. One idea would be to have proxy uncertainties increase as one gets farther from central age of each proxy data point, and then require that the analogues be within the bounds of all proxy data points. The authors explain their reasoning for their choices in their response to my first review, but I disagree with their rationale. It is possible, however, that I am overlooking something.

*Response: We acknowledge the referee's view.*

*Our argument for the ellipse is the following. We regard our time-date point as sampled from a two-dimensional distribution. If we regard this to be a uniform distribution, we would use a rectangular tolerance area. However, we regard the distribution as a two-dimensional Gaussian, which can be visualised as an ellipse in the two-dimensional plane. Thereby the probability density for a valid point reduces further away from the best estimate. If our analogue pool would well sample the climate space, we could weigh our time-data points by their likelihood within the two-dimensional Gaussian plane. Then values that are far off in either or both dimensions would be given less weight. However, as we have only a rather small candidate pool, we resort to a binary criterion of inclusion and exclusion.*

I fear that the problems in both of the points above will become worse as more proxy records added. Users of the methodology could deal with this by excluding certain proxies (as mentioned in the example earlier), but this seems time intensive and potentially somewhat arbitrary. Regarding my second point above, it would probably take a lot of time to change the method entirely, but I would like to see a little more discussion (perhaps an extra paragraph) about this point in the paper.

*Response: Once more we thank the referee for their thoughtful comments. We will add discussion on the second point.*

Specific comments

- Page 1, line 19. By "climate state" do you mean the spatial climate field?

*Response: We rephrase this now in this way.*

- Figure 1 caption. Perhaps mention that the pseudoproxy locations are slightly offset from the actual proxy locations. I know that this is mentioned in the paper's text, but including a brief statement in the Fig. 1 caption should help prevent people from wondering why it appears that the same panel is shown twice.

*Response: In response to referee #1, we now replot the Figure, and in response to referee #2 we also add a note.*

- Page 6, lines 27-28. The statement "Using anomalies circumvents this issue" makes it sound like you are going to use anomalies, when you don't. Consider rephrasing to make this clearer.

*Response: We thank the referee for pointing this out, our revisions try to make this clearer.*

- Page 6, line 32: "Dansgaard-Oeschger (DO)" should be "Dansgaard-Oeschger (DO) events".

*Response: We thank the referee.*

- Page 7, line 5. I would change "original temperature units" to "absolute temperature units", since "original" makes it sound like you would be using the measured proxy units.

*Response: We follow the advise.*

- Page 8. Regarding the 90% and 99% uncertainty values (and later 99.9% and 99.99%), are temporal uncertainty values the same for each proxy?

*Response: No. For the real proxies, we utilize the one data standard deviations provided in Marcott et al. (2013) for each proxy. Similarly, the pseudoproxy generation provides variable uncertainties. We clarify this.*

- Page 9, line 17-18. The phrase "the envelope does not necessarily cover all years within the period of interest" doesn't make much sense to me. Can you rephrase?

*Response: We try to do so, but also try to explain it here: the envelope stretches along the time dimension. The periods of interest is a continuous series of years from, e.g., 15,000 BP until today. The envelope, however, may be not continuous because of sample gaps.*

- Page 9, lines 32-33: The paper states ". . . increase the probability to find a valid analogue at a certain date". But what if widening the time uncertainties made new proxies relevant to some ages? Couldn't that actual reduce the likelihood of finding an analogue (see page 24, lines 24- 26)?

*Response: Exactly. We hoped to cover this case with "usually", but will be more explicit now.*

- Page 12, lines 14-15. The paper says "Regarding proxy uncertainty, we try to identify as complete uncertainty estimates as possible from Marcott et al. (2013) and their references. For the sake of simplicity, we decided to assume an uncertainty of sigma = 1K for all proxies." If you use a common number for all proxies, why did you look at individual proxy values? Is 1K close to the mean or median value?

*Response: The reason is that simply some references do not provide enough information to obtain a full uncertainty estimate. Our aim was to obtain full estimates for all proxies.*

*We modify the paragraph.*

- Page 15, lines 1-2: I'm not sure exactly what this sentence means: "The latter modification also avoids that individual data points have an overly strong influence within our envelopes of tolerance." Can you rephrase?

*Response: We try to clarify this.*

- Page 15, line 4: I'm not sure what "black lines" refers to in this sentence.

*Response: Sorry, we remove this now.*

- Page 15, line 5: Remove the word "below".

*Response: We do so.*

- Figure 6, panel c: Perhaps rearrange the labels so that "QUEST FAMOUS target" and "pseudoproxy" are listed next to each other (since the pseudoproxy is generated from the QUEST model output, right?). Also, put the "Analogue median" and "Analogue range" labels next to each other since they're also related.

*Response: We do so.*

- Figure 6, panel d: The "Valid analogue examples" label only has one color, but the examples are shown in two different colors. Could you add the second color next to the label?

*Response: We hope this is clearer now.*

- Figure 6. Panels d and e took me a while to understand. If possible, could you try to explain it in a more intuitive way?

*Response: We hope our modifications do now clearly describe the panels.*

- Page 21, line 11: Is there any particular reason that age 8000 was chosen for the examples (or age 2430 for the examples in figure 9)?

*Response: These years were chosen adhoc.*

- Page 21, line 14. The paper says "Their local range at no point exceeds 4 degree Celsius". However, it looks like you are referring to the deviation from the median, rather than the total max-min range. Can you check this number and rephrase if necessary? Please do this for the "20 degrees Celsius" value a few lines later as well.

*Response: The quoted numbers indeed already refer to the full local range.*

- Figure 9, panels d and e: In these two panels, the two examples look like time-shifted versions of each other. Why is this? Is it because the same 101-year mean was chosen for each date?

*Response: We thank the referee for this question. We now clarify in the text: "For the chosen year, there is only a small number of analogues, which form a sequence of consecutive simulated years from one simulation. Therefore, the two examples in panels (d) and (e) are simply time shifted sequences."*

- Figure 11 caption: The caption says "All panels include the median, 90% interval, and full range for the reconstructions". However, only one range is shown. Is the displayed range the 90% interval or the full range? Please fix this line, both here and in the caption to Fig. 12.

*Response: It is the full range. We apologize to the referees and the editor for this oversight. We correct both captions.*

- Page 26, line 5: I would remove the comma after "Please".

*Response: We do so.*

- Page 26, line 6: Consider changing "further" to "also".

*Response: We do so.*

- Figure 12 caption. I would change "are valid if they fail" to "are valid even if they fail".

*Response: We follow the referee's suggestion.*

- Page 28, line 12: "warmer" than what?

*Response: We clarify that we mean "warmer" compared to other setups.*

- Page 28, lines 23-25: Consider discussing why the "single best analogue" differs from the default experimental setup (by large amounts for some ages). Is it because the "best" analogue may fail at one or more locations but still produce a better overall fit than other analogues? Since the "single best" methodology sounds similar to previous analogue methods, it would be interesting to explore why it diverges from your main experiment, and whether your method provides benefits over the "single best" analogue.

*Response: We explore this now. While doing so we encountered a bug in our procedure that we also mention in our initial response to the editor and the referees. Therefore also the results for the single best reconstruction change.*

- Page 30, line 2. I don't understand the use of "e.g." in this sentence. Please rephrase.

*Response: We remove it.*

- Page 30, line 11. What does "seasonal sensitivities" in this sentence refer to? I thought that you already considered seasonality in your proxy-to-model comparisons (as stated on page 12, lines 10-12). Or, if all proxy comparisons are made to annual-mean model data, than this needs to be corrected at several points in the paper (e.g. page 12 and Table 1 "season used").

*Response: We clarify this. Seasonal sensitivities at this point do not mean the assumed seasonal sensitivities of our reconstruction process or a prior calibration but the true seasonal sensitivities of the recorders and how they may change over time. This includes that our assumptions may be too crude. We hope the following quote from Jonkers and Kučera (2017, https://doi.org/10.5194/cp-13-573-2017) for planktonic foraminifera clarifies our point: "seasonality changes with temperature in a way that minimises the environmental change that individual species experience".*

- Page 32, line 30. The method succeeds only for some ages.

*Response: We clarify this.*

- Page 34, line 8. What does "central" mean in this sentence?

*Response: Thank you. We mean by "central" here the part of the simulation name as listed in the respository that identifies which simulation was used. We clarify this by removing "central" and combining the two sentences.*

Despite the quantity of these comments, I thought that the paper provides useful considerations and tests of the analogue method. Although I still disagree with the shape of the uncertainty ellipses, I think this is useful and interesting research.

*Response: We thank the referee for this positive recommendation.*

---

## Author Response (AR4)

Dear editor, dear referee,

We hope our response finds you well.

We thank you for your comments. Below we address the suggested corrections.

On behalf of the authors

Yours sincerely

Oliver Bothe

**List of changes**

Minor corrections in line with editor and referee comments.

**Editor**

Dear authors,

your manuscript has now been evaluated by one reviewer and they are satisfied with the changes and the way you handled the bug you identified. I am therefore happy to accept your manuscript pending some very minor technical corrections. Please see the report by the reviewer and the annotated pdf.

Kind regards, Lukas Jonkers

*Response: Dear editor, thank you for your annotations. We hope to have caught all. We changed them according to your suggestion.*

**Referee 1**

I am satisfied with the latest version of the Manuscript. I think the quality of the study greatly improved since the first submission, both in terms of analysis and presentation.

*Response: We thank the referee.*

Regarding the "pre-processing bug" reported by the authors I think the proposed solution (excluding the affected data from the study) is reasonable and I see no reason to demand further changes in this direction.

*Response: Again we thank the referee.*

I just noticed a couple of minor technical corrections which I mention below.

Technical comments:

1. Similar sentence repeated in consecutive paragraphs: Page 8, line 31-32 of tracked changes file: "Then, our tolerance construct takes the shape of a two-dimensional Gaussian. This implies that our tolerance areas are ellipses." Page 9, lines 6-7: "As we take this distribution to be a two-dimensional Gaussian tolerance areas can be visualised as ellipses, the tolerance area becomes an ellipse"

*Response: We thank the referee for highlighting this and correct it. We remove the sentence "As we ... ellipse" and also change in the then following sentence "these" to "the".*

2. Both "single-best" and "single best" nomenclature is used along the text, please unify.

*Response: We thank the referee for highlighting this and correct it. We will use "single best" throughout.*

[revised manuscript text omitted]